# THE WASSERSTEIN BELIEVER
## LEARNING BELIEF UPDATES FOR PARTIALLY OBSERVABLE ENVIRONMENTS THROUGH RELIABLE LATENT SPACE MODELS

**Raphael Avalos**[1] *    **Florent Delgrange**[1,2] *
**Ann Nowé**[1] †    **Guillermo A. Pérez**[2,3] †    **Diederik M. Roijers**[1,4] †
[1] AI Lab, Vrije Universiteit Brussel (Belgium)    [2] University of Antwerp (Belgium)
[3] Flanders Make (Belgium)    [4] City of Amsterdam (The Netherlands)
{raphael.avalos, florent.delgrange}@vub.be

## ABSTRACT

Partially Observable Markov Decision Processes (POMDPs) are used to model environments where the state cannot be perceived, necessitating reasoning based on past observations and actions. However, remembering the full history is generally intractable due to the exponential growth in the history space. Maintaining a probability distribution that models the belief over the current state can be used as a sufficient statistic of the history, but its computation requires access to the model of the environment and is often intractable. While SOTA algorithms use Recurrent Neural Networks to compress the observation-action history aiming to learn a sufficient statistic, they lack guarantees of success and can lead to sub-optimal policies. To overcome this, we propose the Wasserstein Belief Updater, an RL algorithm that learns a latent model of the POMDP and an approximation of the belief update under the assumption that the state is observable during training. Our approach comes with theoretical guarantees on the quality of our approximation ensuring that our latent beliefs allow for learning the optimal value function.

## 1 INTRODUCTION

*Partially Observable Markov Decision Processes* (POMDPs) define a powerful framework for modeling decision-making in uncertain environments where the state is not fully observable. These problems are common in many real-world applications, such as robotics (Lauri et al., 2023), and recommendation systems (Wu et al., 2021). In contrast to *Markov Decision Processes* (MDPs), in a POMDP the agent perceives an imperfect observation of the state that does not suffice as conditioning signal for an optimal policy. As such, optimal policies must take the entire interaction history into account. As the space of possible histories scales exponentially in the length of the episode, using histories to condition policies is generally intractable. An alternative is the notion of *belief*, which is defined as a probability distribution over states based on the agent history. Beliefs are a sufficient statistic of the history (Kaelbling et al., 1998) but the computation of their closed-form expression require the access to a model of the environment and is in general intractable.

To overcome those challenges, SOTA algorithms compress the history into a fixed-size vector with the help of *Recurrent Neural Networks* (RNNs) (Hausknecht & Stone, 2015). Nonetheless, this may lead to information loss, resulting in suboptimal policies. To improve the likelihood of obtaining a sufficient statistic, RNNs can be combined with regularization techniques, including generative models (Chen et al., 2022; Hafner et al., 2019; 2021), particle filtering (Igl et al., 2018; Ma et al., 2020), and predicting distant observations (Gregor et al., 2018; 2019). However, *none of these techniques guarantee that the representation of histories induced by RNNs is suitable for optimizing the return*. Additionally, many algorithms assume that beliefs are simple distributions (e.g., Gaussian), which limits their applicability (Gregor et al., 2018; Lee et al., 2020; Hafner et al., 2021).

In this paper, we propose *Wasserstein Belief Updater* (WBU), a model-based reinforcement learning (RL) algorithm for POMDPs that allows learning the belief space over the unobservable states. Specifically, WBU learns an approximation of the belief update rule through a latent space model

---

* Both authors contributed equally to this research, alphabetic order.    † Equal supervision.

whose behaviors (expressed as expected returns) are close to those of the original environment. Furthermore, we show that WBU is guaranteed to induce a suitable representation of the history to optimize the return. WBU consists of three components that are learned in a round-robin fashion: the model, the belief learner, and the policy (Fig. 1). Harnessing only histories to learn a model whose dynamics can be provably linked to the original unobservable environment poses a considerable challenge. Therefore, in the same spirit as the *Centralized Training with Decentralized Execution* paradigm in *multi-agent* RL (MARL) (Oliehoek et al., 2008; Avalos et al., 2022), where leveraging additional information such as the true state of the environment is a common practice, *we assume that the POMDP states can be accessed during training*. While this might seem restrictive at first sight, this assumption is typically met in simulation-based training and can be applied in real-world settings such as robotics, where extra sensors can be used during training in a laboratory setting.

Our core contribution is the *development of a sound framework equipped with theoretical guarantees in the context of RL within partial observability*. While SOTA algorithms primarily concentrate on enhancing the overall return — potentially resulting in substantial performance gains — we contend that performance is not the exclusive goal and that possessing guarantees is equally important, as the balance between these two aspects varies based on the specific application. By tackling POMDPs with a formal approach, we offer theoretical guarantees that other methods cannot provide: we ensure that *our latent model is able to replicate the dynamics of the original, partially observable environment*, which further *yields a belief representation suitable for learning the value function*.

We learn the latent model of the POMDP via a *Wasserstein auto-encoded MDP* (WAE-MDP, Delgrange et al. 2022), which embeds bisimulation metrics — intuitively leading to our guarantees. In parallel, we maintain a belief distribution over the latent state space via a *belief update network*: we minimize its *Wasserstein distance* to the exact belief update rule, through a tractable variational proxy. To allow for complex belief distributions, we use *normalizing flows* (Kobyzev et al., 2021). In contrast to SOTA algorithms, the beliefs are only optimized towards accurately replicating its update rule. While we call recursively the belief network to maintain the belief distribution, we do not back-propagate through time and thus implement it as a simple feed forward network. The policy is learned on the latent belief space using a vector integrating the parameters of the belief distribution. Our experimental results are promising and show the ability of our algorithm to learn to encode the history into a representation useful to learn a policy, *without using RNNs*.

**Other related work.** DVRL (Igl et al., 2018) extends A2C (Mnih et al., 2016) combined with RNNs (R-A2C) with auxiliary losses aiming to learn beliefs via a variational autoencoder and particle filtering, but it lacks guarantees. DVRL further assumes independent normal distributions for beliefs, limiting its applicability. FORBES (Chen et al., 2022) use normalizing flows but learn policies conditioned on latent states, which is suboptimal as the state distribution is approximated with a single sample. Some works focus on specific POMDP types, like compact image representations (e.g., visual motor tasks, e.g., Lee et al. 2020) or states masked with Gaussian noise (Wang & Tan, 2021). While accessing states is common in MARL, it is less common in single-agent but has been explored in kernel-POMDPs (Nishiyama et al., 2012), and more recently for sample efficient learning (Lee et al., 2023). Leveraging additional information available during training (not necessarily the states) has also been explored by Lambrechts et al. (2023), but RNNs remain crucial while no abstraction nor representation quality guarantee are provided. Finally, other works (Gelada et al., 2019; Delgrange et al., 2022) study similar value difference bounds to ours and connect them to bisimulation theory (Larsen & Skou, 1989; Givan et al., 2003), but in fully observable environments.

## 2 BACKGROUND

### 2.1 PROBABILITY DISTRIBUTIONS AND DISCREPANCY MEASURES

We write $\Sigma(\mathcal{X})$ for the set of all *measurable* subsets of a complete, separable space $\mathcal{X}$, $\Delta(\mathcal{X})$ for the set of measures on $\mathcal{X}$, and $\delta_a \in \Delta(\mathcal{X})$ for the *Dirac measure* with impulse $a \in \mathcal{X}$. Let $P, Q \in \Delta(\mathcal{X})$ be measures with densities $p, q$, the divergence between $P$ and $Q$ can be measured via:

- the solution of the *optimal transport* problem (OT), defined as $\mathcal{W}_c(P, Q) = \inf_\lambda \mathbb{E}_{x,y \sim \lambda} c(x, y)$, which is the *minimum cost of changing $P$ into $Q$* (Villani, 2009), where $c \colon \mathcal{X} \times \mathcal{X} \to [0, \infty)$ is a cost function and the infimum is taken over the set of all *couplings* of $P$ and $Q$. When $c$ is equal to a distance metric $d$ over $\mathcal{X}$, $\mathcal{W}_d$ is the *Wasserstein distance* between the two distributions.

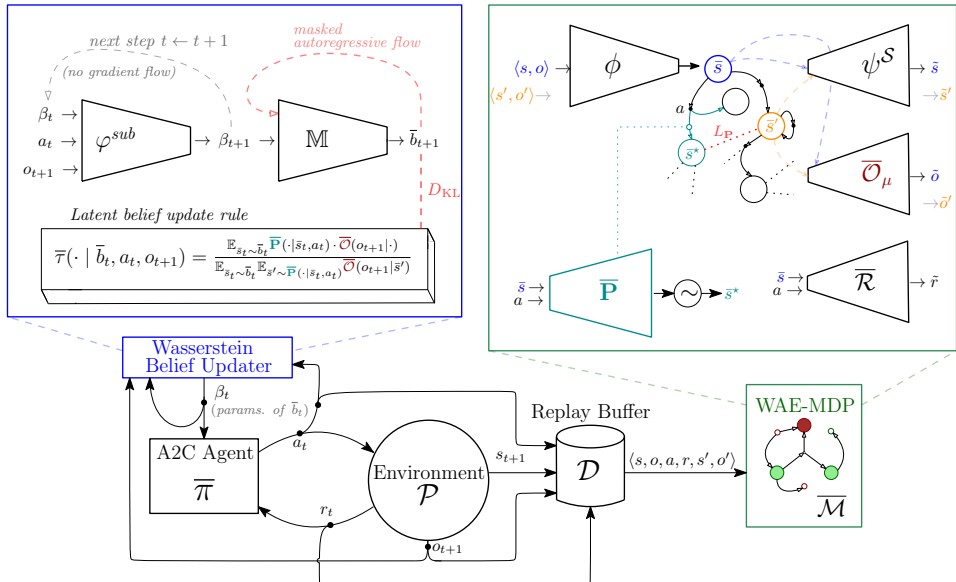

Figure 1: *WBU framework*. The WAE-MDP is presented in Sect. 3, and WBU in Sect. 4. Learning the different components is done in a round-robin fashion. The WAE-MDP learns from data collected by the RL agent and stored in a Replay Buffer. WBU uses the transition function $\overline{\mathbf{P}}$ and observation decoder $\overline{\mathcal{O}}$ of the WAE-MDP to learn to approximate the belief update rule. The agent learns a policy conditioned on the resulting *sub-belief* $\beta_t$ (i.e., the parameters of the latent belief $\bar{b}_t$).

- the *total variation distance* (TV), defined as $d_{TV}(P, Q) = \sup_{A \in \Sigma(\mathcal{X})} |P(A) - Q(A)|$. If $\mathcal{X}$ is equipped with the discrete metric $\mathbf{1}_{\neq}$, TV coincides with the Wasserstein metric.

- the *Kullback-Leibler* (KL) divergence, defined as $D_{\mathrm{KL}}(P, Q) = \mathbb{E}_{x \sim P} \left[ \log \left( {p(x)}/{q(x)} \right) \right]$.

## 2.2 DECISION MAKING UNDER UNCERTAINTY

**Markov Decision Processes** (MDPs) are tuples $\mathcal{M} = \langle \mathcal{S}, \mathcal{A}, \mathbf{P}, \mathcal{R}, s_I, \gamma \rangle$ where $\mathcal{S}$ is a set of *states*; $\mathcal{A}$, a set of *actions*; $\mathbf{P} \colon \mathcal{S} \times \mathcal{A} \to \Delta(\mathcal{S})$, a *probability transition function* that maps the current state and action to a *distribution* over the next states; $\mathcal{R} \colon \mathcal{S} \times \mathcal{A} \to \mathbb{R}$, a *reward function*; $s_I \in \mathcal{S}$, the *initial state*; and $\gamma \in [0, 1)$ a discount factor. We refer to MDPs with continuous state or action spaces as *continuous MDPs*. In that case, we assume $\mathcal{S}$ and $\mathcal{A}$ are complete separable metric spaces equipped with a Borel $\sigma$-algebra. An agent interacting in $\mathcal{M}$ produces *trajectories*, i.e., sequences of states and actions $\langle s_{0:T}, a_{0:T-1} \rangle$ where $s_0 = s_I$ and $s_{t+1} \sim \mathbf{P}(\cdot \mid s_t, a_t)$ for $t < T$.

**Policies and probability measure.** A *stationary policy* $\pi \colon \mathcal{S} \to \Delta(\mathcal{A})$ prescribes which action to choose at each step of the interaction. A policy $\pi$ and $\mathcal{M}$ induce a unique probability measure $\mathbb{P}^{\mathcal{M}}_\pi$ on the Borel $\sigma$-algebra over (measurable) infinite trajectories (Puterman, 1994). The typical goal of an RL agent is to learn a policy that maximizes the *expected return*, given by $\mathbb{E}^{\mathcal{M}}_\pi \left[ \sum_{t=0}^{\infty} \gamma^t \cdot \mathcal{R}(s_t, a_t) \right]$, by interacting with $\mathcal{M}$. We omit the superscript when the context is clear.

**Partially Observable MDPs** (POMDPs) are tuples $\mathcal{P} = \langle \mathcal{M}, \Omega, \mathcal{O} \rangle$ where $\mathcal{M}$ is an MDP with state space $\mathcal{S}$ and action space $\mathcal{A}$; $\Omega$ is a set of *observations*; and $\mathcal{O} \colon \mathcal{S} \times \mathcal{A} \to \Delta(\Omega)$ is an *observation function* that defines the distribution of observations that may occur when the MDP $\mathcal{M}$ transitions to a state upon the execution of a particular action. An agent in $\mathcal{P}$ actually interacts in $\mathcal{M}$, but *without directly observing the states* of $\mathcal{M}$: instead, the agent perceives observations, which yields *histories*, i.e., sequences of actions and observations $\langle a_{0:T-1}, o_{1:T} \rangle$ that can be associated to an (unobservable) trajectory $\langle s_{0:T}, a_{0:T-1} \rangle$ in $\mathcal{M}$, where $o_{t+1} \sim \mathcal{O}(\cdot \mid s_{t+1}, a_t)$ for all $t < T$.

**Beliefs.** Unlike in MDPs, where it is sufficient to take decisions based on the current state (Puterman, 1994), policies solely based on the current observation of $\mathcal{P}$ are not sufficient to optimize the return. Intuitively, due to the partial observability, the agent must take into account full histories in order

to make an informed decision on its next action. Alternatively, the agent can maintain a *belief* $b_t \in \Delta(\mathcal{S}) = \mathcal{B}$ over the current state of $\mathcal{M}$ (Åström, 1965). Given the next observation $o_{t+1}$, the next belief $b_{t+1}$ is computed according to the *belief update function* $\tau \colon \mathcal{B} \times \mathcal{A} \times \Omega \to \mathcal{B}$, where $\tau(b_t, a_t, o_{t+1}) = b_{t+1}$ iff the belief over any next state $s_{t+1} \in \mathcal{S}$ has for density

$$b_{t+1}(s_{t+1}) = \frac{\mathbb{E}_{s_t \sim b_t} \mathbf{P}(s_{t+1} \mid s_t, a_t) \cdot \mathcal{O}(o_{t+1} \mid s_{t+1}, a_t)}{\mathbb{E}_{s_t \sim b_t} \mathbb{E}_{s' \sim \mathbf{P}(\cdot \mid s_t, a_t)} \mathcal{O}(o_{t+1} \mid s', a_t)}. \tag{1}$$

Each belief $b_{t+1}$ constructed this way is a *sufficient statistic* for the history $\langle a_{0:t}, o_{1:t+1} \rangle$ to optimize the return (Kaelbling et al., 1998). We write $\tau^*(a_{0:t}, o_{1:t+1}) = \tau(\cdot, a_t, o_{t+1}) \circ \cdots \circ \tau(\delta_{s_I}, a_0, o_1) = b_{t+1}$ for the recursive application of $\tau$ along the history. The belief update rule derived from $\tau$ allows to formulate $\mathcal{P}$ as a continuous *belief MDP* $\mathcal{M}_{\mathcal{B}} = \langle \mathcal{B}, \mathcal{A}, \mathbf{P}_{\mathcal{B}}, \mathcal{R}_{\mathcal{B}}, b_I, \gamma \rangle$, where $\mathbf{P}_{\mathcal{B}}(b' \mid b, a) = \mathbb{E}_{s \sim b} \mathbb{E}_{s' \sim \mathbf{P}(\cdot \mid s, a)} \mathbb{E}_{o' \sim \mathcal{O}(\cdot \mid s', a)} \delta_{\tau(b, a, o')}(b')$; $\mathcal{R}_{\mathcal{B}}(b, a) = \mathbb{E}_{s \sim b} \mathcal{R}(s, a)$; and $b_I = \delta_{s_I}$. As for all MDPs, $\mathcal{M}_{\mathcal{B}}$ and any stationary policy for $\mathcal{M}_{\mathcal{B}}$ (thus conditioned on beliefs) induce a well-defined probability space over trajectories of $\mathcal{M}_{\mathcal{B}}$, which allows optimizing the expected return in $\mathcal{P}$.

## 2.3 Latent Space Modeling

**Latent MDPs.** Given the original (continuous or very large, possibly unknown) environment $\mathcal{M}$, a *latent space model* is another (tractable, explicit) MDP $\overline{\mathcal{M}} = \langle \overline{\mathcal{S}}, \mathcal{A}, \overline{\mathbf{P}}, \overline{\mathcal{R}}, \bar{s}_I, \gamma \rangle$ with state space linked to the original one via a *state embedding function*: $\phi \colon \mathcal{S} \to \overline{\mathcal{S}}$.

**Wasserstein Auto-encoded MDPs** (WAE-MDPs, Delgrange et al. 2023) are latent space models that are trained based on the OT from trajectories resulting from the execution of the RL agent policy in the real environment $\mathcal{M}$, to that reconstructed from the latent model $\overline{\mathcal{M}}$. The optimization process relies on a temperature $\lambda \in (0, 1)$ that controls the continuity of the latent space learned, the zero-temperature limit corresponding to a discrete latent state space (see Appendix E for a discussion). This procedure guarantees $\overline{\mathcal{M}}$ to be probably approximately *bisimilarly close* (Larsen & Skou, 1989; Givan et al., 2003; Delgrange et al., 2022) to $\mathcal{M}$ as $\lambda \to 0$: in a nutshell, *bisimulation metrics* imply the closeness of the two models in terms of probability measures and expected return (Desharnais et al., 2004; Ferns et al., 2011). Specifically, a WAE-MDP learns the following components:

| a *state embedding function* | $\phi \colon \mathcal{S} \to \overline{\mathcal{S}}$ | a *latent transition function* | $\overline{\mathbf{P}} \colon \overline{\mathcal{S}} \times \mathcal{A} \to \Delta(\overline{\mathcal{S}})$ | |
|---|---|---|---|---|
| a *latent reward function* | $\overline{\mathcal{R}} \colon \overline{\mathcal{S}} \times \mathcal{A} \to \mathbb{R}$ | a *state decoder* | $\psi \colon \overline{\mathcal{S}} \to \mathcal{S}.$ | (2) |

## 3 Learning the Dynamics

The agent is assumed to operate within a POMDP. In an RL setting, the former have no explicit access to the environment dynamics: instead, it reinforces its behaviors through interactions and experiences without directly accessing the transition, reward, and observation functions of the environment. To provide the aforementioned guarantees, we henceforth adhere the following assumption.

**Assumption 1** (Access to the state during training). *In addition to the observation, the agent is able to observe the true state of the environment, but only during the training phase.*

*Remark* 1. Seemingly restrictive, Assumption 1 can actually be met in a broad range of training scenarios, in particular those relying on simulators, where one could merely consider the RAM as the state. Otherwise, additional sensors with higher fidelity could be considered to obtain the state. Other applicable scenarios are model-based design or model-predictive control, where a model is accessible during training, and situations where accessing the state is costly (Bulychev et al., 2012).

Concretely, when the RL agent interacts in a POMDP $\mathcal{P} = \langle \mathcal{M}, \Omega, \mathcal{O} \rangle$ with underlying MDP $\mathcal{M} = \langle \mathcal{S}, \mathcal{A}, \mathbf{P}, \mathcal{R}, s_I, \gamma \rangle$, *we leverage this access to allow the agent to learn the dynamics of the environment*, i.e., those of $\mathcal{M}$, as well as those related to the observation function $\mathcal{O}$. To do so, we learn an internal, explicit representation of the experiences gathered, through a latent space model. *We then use this model as a teacher for the agent to make it learn how to perform its belief updates.* Hence, acquiring an accurate environment model is crucial to learn a reliable belief update function. In Sect. 3.2, we further demonstrate that the resulting model is guaranteed to closely replicate the original environment behaviors. The trick we use to learn such a model is to reason on an equivalent POMDP, where the underlying MDP is refined to encode all the crucial dynamics.

### 3.1 THE LATENT POMDP ENCODING

We enable learning the dynamics of $\mathcal{P}$ via a WAE-MDP by considering the POMDP $\mathcal{P}^\uparrow = \langle \mathcal{M}_\Omega, \Omega, \mathcal{O}^\uparrow \rangle$, where (i) the state space of the underlying MDP is refined to encode the observations: $\mathcal{M}_\Omega = \langle \mathcal{S}_\Omega, \mathcal{A}, \mathbf{P}_\Omega, \mathcal{R}_\Omega, \langle s_I, o_I \rangle, \gamma \rangle$ with $\mathcal{S}_\Omega = \mathcal{S} \times \Omega$, $\mathbf{P}_\Omega(s', o' \mid s, o, a) = \mathbf{P}(s' \mid s, a) \cdot \mathcal{O}(o' \mid s', a)$, $\mathcal{R}_\Omega(\langle s, o \rangle, a) = \mathcal{R}(s, a)$, and $o_I$ is an observation from $\Omega$ linked to the initial state $s_I$; (ii) the observation function $\mathcal{O}^\uparrow \colon \mathcal{S}_\Omega \to \Omega$ is the *deterministic* projection of the refined state on the observation space, with $\mathcal{O}^\uparrow(\langle s, o \rangle) = o$. The POMDPs $\mathcal{P}$ and $\mathcal{P}^\uparrow$ are equivalent (Chatterjee et al., 2016): $\mathcal{P}^\uparrow$ captures the stochasticity of $\mathcal{O}$ in its transition function through the refinement of the state space, further *yielding a deterministic observation function*, only dependent on refined states.

Henceforth, we reduce the problem of learning a latent space model of $\mathcal{P}$ to learning a WAE-MDP from $\mathcal{M}_\Omega$. Precisely, we learn a latent MDP $\overline{\mathcal{M}} = \langle \overline{\mathcal{S}}, \mathcal{A}, \overline{\mathbf{P}}, \overline{\mathcal{R}}, \bar{s}_I, \gamma \rangle$ linked to $\mathcal{M}_\Omega$ via the embedding $\phi \colon \mathcal{S}_\Omega \to \overline{\mathcal{S}}$. The latent MDP $\overline{\mathcal{M}}$ encodes the observation dynamics through $\overline{\mathbf{P}}$ and enables to learn the deterministic observation function $\mathcal{O}^\uparrow$ through the decoder $\psi \colon \overline{\mathcal{S}} \to \mathcal{S}_\Omega$, by decomposing the latter in two networks $\psi^{\mathcal{S}} \colon \overline{\mathcal{S}} \to \mathcal{S}$ and $\overline{\mathcal{O}}_\mu \colon \overline{\mathcal{S}} \to \Omega$, further yielding $\psi(\bar{s}) = \langle \psi^{\mathcal{S}}(\bar{s}), \overline{\mathcal{O}}_\mu(\bar{s}) \rangle$ for all $\bar{s} \in \overline{\mathcal{S}}$. This way, the WAE-MDP learns all the components of $\mathcal{P}^\uparrow$, the latter being equivalent to $\mathcal{P}$. With this model, we construct a *latent POMDP* $\overline{\mathcal{P}} = \langle \overline{\mathcal{M}}, \Omega, \overline{\mathcal{O}} \rangle$, where $\overline{\mathcal{O}}$ is a Dirac measure with impulse $\overline{\mathcal{O}}_\mu$. Figure 6 in Appendix F presents a visualization of the relationship between the different models.

As with any POMDP, the belief update function $\bar{\tau}$ of $\overline{\mathcal{P}}$ allows to reason on the belief space to optimize the return. The belief update procedure is illustrated in Appendix B. Formally, assuming the latent belief at time step $t \geqslant 0$ is $\bar{b}_t \in \Delta(\overline{\mathcal{S}}) = \mathcal{B}$, $a_t$ is executed, and then $o_{t+1}$ observed, $\bar{b}_t$ is updated according to $\bar{\tau}(\bar{b}_t, a_t, o_{t+1}) = \bar{b}_{t+1}$ iff, for any (unobservable) next state $\bar{s}_{t+1} \in \overline{\mathcal{S}}$,

$$\bar{b}_{t+1}(\bar{s}_{t+1}) = \frac{\mathbb{E}_{\bar{s}_t \sim \bar{b}_t} \overline{\mathbf{P}}(\bar{s}_{t+1} \mid \bar{s}_t, a_t) \cdot \overline{\mathcal{O}}(o_{t+1} \mid \bar{s}_{t+1})}{\mathbb{E}_{\bar{s}_t \sim \bar{b}_t} \mathbb{E}_{\bar{s}' \sim \overline{\mathbf{P}}(\cdot \mid \bar{s}_t, \bar{a}_t)} \overline{\mathcal{O}}(o_{t+1} \mid \bar{s}')}. \tag{3}$$

**Latent policies.** Given *any* history $h \in (\mathcal{A} \cdot \Omega)^*$, running a latent policy $\bar{\pi} \colon \overline{\mathcal{B}} \to \Delta(\mathcal{A})$ in $\mathcal{P}$ is possible by converting $h$ into a latent belief $\bar{\tau}^*(h) = \bar{b}$ and executing the action prescribed by $\bar{\pi}(\cdot \mid \bar{b})$. Training $\overline{\mathcal{M}}$ grants access to the dynamics required to update the belief through its closed form (Eq. 3). However, integrating over the full latent space remains computationally intractable.

> As a solution, we propose to leverage the access to the dynamics of $\overline{\mathcal{M}}$ to learn a *latent belief encoder* $\varphi \colon \overline{\mathcal{B}} \times \mathcal{A} \times \Omega \to \overline{\mathcal{B}}$ that approximates the belief update function by minimizing $D(\bar{\tau}^*(h), \varphi^*(h))$ for *some* discrepancy $D$ and $h \in (\mathcal{A} \cdot \Omega)^*$ drawn from *some* distribution. The belief encoder $\varphi$ thus enables to learn a policy $\bar{\pi}$ conditioned on latent beliefs to optimize the return in $\mathcal{P}$: *given the current history $h$, the next action to play is given by $a \sim \bar{\pi}(\cdot \mid \varphi^*(h))$.*[1]

Two key questions arise: (i) does the WAE-MDP encoding induces a latent POMDP with behaviors close to $\mathcal{P}$? (ii) is the history representation induced by $\varphi$ suitable for optimizing the expected return in $\mathcal{P}$? Our guarantees hinge on the history distribution and chosen discrepancy. The next section details our theoretical analysis of the required distributions and losses to answer (i) and (ii).

### 3.2 LOSSES AND THEORETICAL GUARANTEES

To yield the guarantees, we specifically target the *episodic RL process* setting for drawing histories.

**Assumption 2** (Episodic RL process). *The environment $\mathcal{P}$ embeds a special* reset state *so that (i) under any policy, the environment is almost surely eventually reset; (ii) when reset, the environment transitions to the initial state; and (iii) the reset state is observable.*

**Lemma 3.1.** *There exists a well defined stationary distribution $\mathcal{H}_{\bar{\pi}} \in \Delta((\mathcal{A} \cdot \Omega)^*)$ over histories likely to be seen at the limit of the interaction when $\bar{\pi}$ is executed in $\mathcal{P}$* (proof in Appendix D).

*Remark* 2. Assumption 2 is present in a vast majority of RL scenarios, where it is common practice to reset the environment and start anew from an initial state when the agent succeeds, fails, or after a finite number of time steps (Brockman et al., 2016; Pardo et al., 2018). A notable exception is in the domain of continual RL. However, the existence of a stationary distribution (Lem. 3.1) is often assumed in such scenarios (see, e.g., Huang 2020), which allows to relax the episodic assumption.

**Local losses.** The objective function of the WAE-MDP incorporates *local losses* (Gelada et al., 2019) that minimize the expected distance between the original and latent reward and transition functions:

$$L_{\mathcal{R}} = \mathop{\mathbb{E}}_{s,o,a \sim \mathcal{H}_{\bar{\pi}}} \left| \mathcal{R}(s,a) - \overline{\mathcal{R}}(\phi(s,o),a) \right|, \quad L_{\mathbf{P}} = \mathop{\mathbb{E}}_{s,o,a \sim \mathcal{H}_{\bar{\pi}}} \mathcal{W}_{\bar{d}} \left( \phi \mathbf{P}_{\Omega}(\cdot \mid s,o,a), \overline{\mathbf{P}}(\cdot \mid \phi(s,o),a) \right);$$

and both are optimized *locally*, i.e., under $\mathcal{H}_{\bar{\pi}}$, where $s,o,a \sim \mathcal{H}_{\bar{\pi}}$ is a shorthand for (i) $h \sim \mathcal{H}_{\bar{\pi}}$ so that $o$ is the last observation of $h$, (ii) $s \sim \tau^*(h)$, and (iii) $a \sim \bar{\pi}(\cdot \mid \varphi^*(h))$.[1] Furthermore, $\phi \mathbf{P}(\cdot \mid s,a)$ is the distribution of transitioning to $s' \sim \mathbf{P}(\cdot \mid s,a)$, then embedding it to the latent space $\bar{s}' = \phi(s')$, and $\bar{d}$ is a metric on $\overline{\mathcal{S}}$. In practice, the ability of observing states during learning enables the optimization of those local losses without the need of explicitly storing histories. Instead, we simply store the transitions of $\mathcal{M}_{\Omega}$ encountered while executing $\bar{\pi}$. We also introduce an *observation loss* which allows learning $\overline{\mathcal{O}}$:

$$L_{\mathcal{O}} = \mathop{\mathbb{E}}_{s,o,a \sim \mathcal{H}_{\bar{\pi}}} \mathop{\mathbb{E}}_{s' \sim \mathbf{P}(\cdot \mid s,a)} d_{TV} \left( \mathcal{O}(\cdot \mid s',a), \mathop{\mathbb{E}}_{o' \sim \mathcal{O}(\cdot \mid s',a)} \overline{\mathcal{O}} (\cdot \mid \phi(s',o')) \right). \tag{4}$$

**Belief losses.** We set $D$ as the Wasserstein distance between the true latent belief update and our belief encoder. In addition, we introduce the following *reward and transition regularizers* to reconcile the behaviors obtained in the fully observable latent model $\overline{\mathcal{M}}$ and the partially observable one $\overline{\mathcal{P}}$:

$$L_{\bar{\tau}} = \mathop{\mathbb{E}}_{h \sim \mathcal{H}_{\bar{\pi}}} \mathcal{W}_{\bar{d}} \left( \bar{\tau}^*(h), \varphi^*(h) \right), \quad L_{\overline{\mathcal{R}}}^{\varphi} = \mathop{\mathbb{E}}_{h,s,o,a \sim \mathcal{H}_{\bar{\pi}}} \mathop{\mathbb{E}}_{\bar{s} \sim \varphi^*(h)} \left| \overline{\mathcal{R}}(\phi(s,o),a) - \overline{\mathcal{R}}(\bar{s},a) \right|,$$

$$L_{\overline{\mathbf{P}}}^{\varphi} = \mathop{\mathbb{E}}_{h,s,o,a \sim \mathcal{H}_{\bar{\pi}}} \mathop{\mathbb{E}}_{\bar{s} \sim \varphi^*(h)} \mathcal{W}_{\bar{d}} \left( \overline{\mathbf{P}}(\cdot \mid \phi(s,o),a), \overline{\mathbf{P}}(\cdot \mid \bar{s},a) \right). \tag{5}$$

$L_{\overline{\mathcal{R}}}^{\varphi}$ and $L_{\overline{\mathbf{P}}}^{\varphi}$ aim at regularizing $\varphi$ and minimize the gap between the rewards (resp. transition probabilities) that are expected when drawing states from the current belief compared to those actually observed. Again, the ability to observe the states during training enables optimizing those losses without explicitly requiring the states to execute the policy. The belief loss and the related two regularizers can be optimized *on-policy*, i.e., coupled with the optimization of the latent policy $\bar{\pi}$ that is used to generate the episodes.

**Value difference bounds.** We provide guarantees concerning the *agent behaviors in* $\mathcal{P}$, when the policies are *conditioned on latent beliefs*. To do so, we formalize the behaviors of the agent through *value functions*. For a specific policy $\pi$, the value of a history is the expected return that would result from continuing to follow the policy from the latest point reached in that history: $V_{\pi}(h) = \mathbb{E}_{\pi} \left[ \sum_{t=0}^{\infty} \gamma^t r_t \mid b_I = \bar{\tau}^*(h) \right]$. Similarly, we write $\overline{V}_{\bar{\pi}}$ for the values of the latent policy $\bar{\pi}$ in $\overline{\mathcal{P}}$.

Suppose the agent uses a latent policy whose inputs are produced by $\varphi$, we claim that when the losses are minimized to zero, then (i) the latent model almost surely mimics the original environment, and (ii) our belief representation captures the value function. Precisely, let $\mathcal{L} = L_{\mathcal{R}} + L_{\overline{\mathcal{R}}}^{\varphi} + \overline{\mathcal{R}}^{\star} L_{\bar{\tau}} + \gamma K_{\overline{V}} \cdot (L_{\mathbf{P}} + L_{\overline{\mathbf{P}}}^{\varphi} + L_{\bar{\tau}} + L_{\mathcal{O}})$, with $\overline{\mathcal{R}}^{\star} = \left\| \overline{\mathcal{R}} \right\|_{\infty}$ and $K_{\overline{V}} = \overline{\mathcal{R}}^{\star}/1-\gamma$, then:

**Theorem 3.2** (Model quality). *For any latent policy $\bar{\pi} \colon \overline{\mathcal{B}} \to \Delta(\mathcal{A})$, the values of $\mathcal{P}$ and $\overline{\mathcal{P}}$ are bounded by the local and belief losses in average when, in both models, the actions are produced by $\bar{\pi}$, which is conditioned on the latent belief induced by $\varphi$, i.e., $a \sim \bar{\pi}(\cdot \mid \varphi^*(h))$:*

$$\mathop{\mathbb{E}}_{h \sim \mathcal{H}_{\bar{\pi}}} \left| V_{\bar{\pi}}(h) - \overline{V}_{\bar{\pi}}(h) \right| \leqslant \frac{\mathcal{L}}{1 - \gamma}. \tag{6}$$

**Theorem 3.3** (Representation quality). *Let $\bar{\pi}^{\star}$ be an optimal policy of $\overline{\mathcal{P}}$, then for any $\epsilon > 0$ and $n \geqslant 0$, there is a $K \geqslant 0$ so that for any histories $h_1, h_2$ of length at most $n$ that are measurable under $\mathcal{P}$ and $\overline{\mathcal{P}}$ with $\varphi^*(h_1) = \bar{b}_1$ and $\varphi^*(h_2) = \bar{b}_2$, the representation induced by $\varphi$ yields:*

$$\left| V_{\bar{\pi}^{\star}}(h_1) - V_{\bar{\pi}^{\star}}(h_2) \right| \leqslant K \mathcal{W}_{\bar{d}} \left( \bar{b}_1, \bar{b}_2 \right) + \epsilon + \frac{K L_{\bar{\tau}} + \mathcal{L}}{1 - \gamma} \left( \frac{1}{\mathcal{H}_{\bar{\pi}^{\star}}(h_1)} + \frac{1}{\mathcal{H}_{\bar{\pi}^{\star}}(h_2)} \right). \tag{7}$$

While Thm. 3.2 asserts that training a WAE-MDP as a latent space model of the environment results in similar behaviors (i.e., close expected returns) compared to the original environment when they are measured under the agent policy — which justifies the usage of $\overline{\mathcal{P}}$ as model of the environment — Thm. 3.3 states that our learned update procedure yields a belief representation which is well-suited to optimize the policy: execution traces leading to close latent beliefs (via our learned updater $\varphi$) are guaranteed to yield close expected returns as well (proofs in Appendix F).

---

[1] Analogous to $\tau^*$, we define $\varphi^*$ as the recursive application of $\varphi$ along histories.

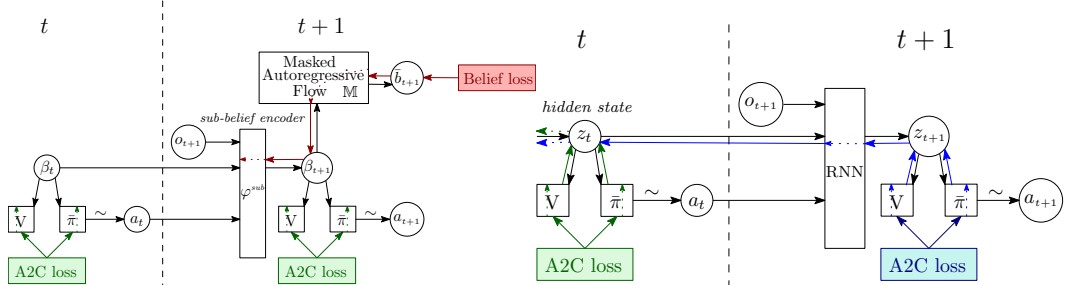

Figure 2: WBU (*left*) learns to encode the history into a sub-belief solely by optimizing the belief loss. The policy, being conditioned on the sub-belief, is learned via A2C and does not back-propagate through the sub-belief encoder. The R-A2C agent (*right*) uses BPTT: the RNN leverages gradients from future time-steps to improve its compression of the history for learning a policy and value function. In both plots, the colored arrows represent the gradient flows of the different losses.

## 4   LEARNING TO BELIEVE

In the following, we assume that we have access to the latent model learned by the WAE-MDP.

**Architecture.** Our latent belief encoder $\varphi$ aims at generalizing to *any* POMDP. Therefore, *we do not make any assumption about the underlying belief distribution*. To accommodate complex belief distributions, we use a *Masked Auto-Regressive Flows* (MAF) (Papamakarios et al., 2017), a type of normalizing flow built on the auto-regressive property. Precisely, to fit with the WAE-MDP framework and leverage the guarantees presented in Sect. 3.2, we use the MAF of Delgrange et al. (2023) that learns relaxed multivariate latent distributions.

The *sub-belief* $\beta_t$ is the vector that embeds the parameters of the belief distribution, which is converted into a belief via the MAF $\mathbb{M}(\beta_t) = \bar{b}_t$. We use a *sub-belief encoder* $\varphi^{\mathrm{sub}}$ to recursively update $\beta_t$ via $\varphi^{\mathrm{sub}}(\beta_t, a_t, o_{t+1}) = \beta_{t+1}$, so that $\varphi(\bar{b}_t, a_t, o_{t+1}) = \mathbb{M} \circ \varphi^{\mathrm{sub}}(\beta_t, a_t, o_{t+1})$. RNNs are trained via back-propagation through time (BPTT), which is challenging (Pascanu et al., 2013). In contrast, albeit sub-beliefs are updated recursively in the same spirit as RNN hidden states, *we do not need to use BPTT and use a simple feed-forward network for $\varphi^{sub}$*, as illustrated in Fig. 2. In R-A2C, RNN hidden states serve as compact representations of histories for the policy. Since values of time-steps closer to the end of an episode are easier to learn, the gradients of future time-steps tend to be more accurate; thus BPTT helps learning. This is in stark contrast to learning the belief update rule: the beliefs of early time-steps are easier to infer, so BPTT is unnecessary and might even be armful. Additionally, viewing the belief update function as the transition function of $\mathcal{M}_{\mathcal{B}}$, disabling BPTT aligns with the literature on model-based RL for learning Markovian transition functions (Gelada et al., 2019; François-Lavet et al., 2019). See Appendix G for a more detailed discussion on BPTT.

**Training.** We aim to train $\varphi^{\mathrm{sub}}$ and $\mathbb{M}$ to approximate the update rule by minimizing the Wasserstein between the belief update rule $\bar{\tau}$ of the latent POMDP, and the belief encoder $\varphi$ (Eq. 5), to leverage the theoretical learning guarantees of Thm. 3.2 and 3.3. However, Wasserstein optimization is known to be challenging, often requiring the use of additional networks, Lipschitz constraints, and a min-max optimization procedure (Arjovsky et al., 2017), similar to how WAE-MDPs are trained. Also, sampling from both distributions is necessary for optimizing Wasserstein and, while sampling from our belief approximation is straightforward, sampling from the update rule (Eq. 3) is non-trivial.

As an alternative to the Wasserstein optimization, we minimize the KL divergence between the two distributions. While this $D_{\mathrm{KL}}$ proxy is easier to optimize and only requires sampling from one of the two distributions (in our case, the belief encoder), it bounds the Wasserstein distance by the Pinsker's inequality (Borwein & Lewis, 2005) in the zero-temperature limit of the WAE-MDP (cf. Appendix E for a discussion).

**On-policy KL divergence.** Using $D_{\mathrm{KL}}$ as a proxy for $\mathcal{W}_{\bar{d}}$ allows to narrow the gap between $\varphi$ and $\bar{\tau}$. We train $\varphi$ *on-policy*, with the same samples as used for $\bar{\pi}$, which aids learning despite gradients are not allowed to flow between the networks. At any time-step $t \geqslant 0$, given the current belief $\bar{b}_t$,

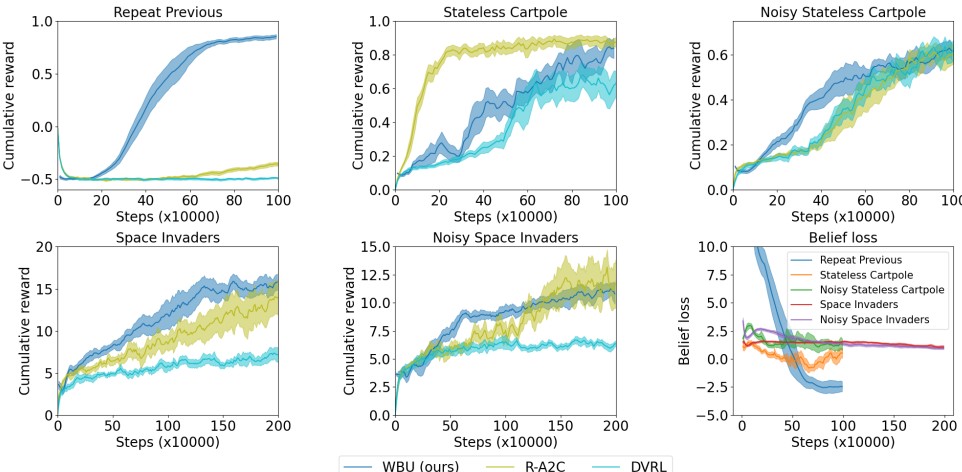

Figure 3: Evolution of the (i) undiscounted cumulative return for WBU, R-A2C and DVRL, and (ii) estimated belief loss during learning for WBU (mean and standard error). We report $5$ instances of each algorithm. Appendix I details the hyperparameter search performed.

the action $a_t$ played by the agent, and the next perceived observation $o_{t+1}$, the belief proxy loss is:

$$D_{\mathrm{KL}}\left(\varphi\left(\bar{b}_t, a_t, o_{t+1}\right) \parallel \bar{\tau}\left(\bar{b}_t, a_t, o_{t+1}\right)\right) = \log\left(\underset{\bar{s} \sim \bar{b}_t}{\mathbb{E}} \underset{\bar{s}' \sim \overline{\mathbf{P}}(\cdot \mid \bar{s}, a_t)}{\mathbb{E}} \overline{\mathcal{O}}\left(o_{t+1} \mid \bar{s}'\right)\right) +$$

$$\underset{\bar{s}_{t+1} \sim \varphi\left(\bar{b}_t, a_t, o_{t+1}\right)}{\mathbb{E}} \left[\log \varphi\left(\bar{s}_{t+1} \mid \bar{b}_t, a_t, o_{t+1}\right) - \log \underset{\bar{s} \sim \bar{b}_t}{\mathbb{E}} \overline{\mathbf{P}}\left(\bar{s}_{t+1} \mid \bar{s}, a_t\right) - \log \overline{\mathcal{O}}\left(o_{t+1} \mid \bar{s}_{t+1}\right)\right]. \quad (8)$$

Eq. 8 consists of $4$ terms: a normalization factor, negative entropy of $\varphi$, belief update conformity with the latent MDP's state transition function, and filtration of latent states unrelated to $o_{t+1}$.

*Remark* 3 (Latent observations). The WAE-MDP learns from the augmented POMDP $\mathcal{P}^\uparrow$ (Sect. 3.1), which is equivalent to $\mathcal{P}$ and possesses a deterministic observation function. The WAE-MDP learns such a deterministic mapping through $\overline{\mathcal{O}}_\mu$. However, when deterministic, the observation terms of Eq. 3 and 8 are Dirac, which prevents learning: deterministically filtering out states from the next belief that do not share the next observation is sample inefficient and may further require constructing the full belief distribution, which is usually intractable. In practice, to alleviate this, we model the observation function as a normal distribution: $\overline{\mathcal{O}}(\cdot \mid \bar{s}) = \mathcal{N}(\overline{\mathcal{O}}_\mu(\bar{s}), \sigma^2)$ where the variance can, e.g., be learned via $L_{\mathcal{O}}$. Notice that we can enforce annealing the variance to zero to recover a Dirac.

**Policy learning** is enabled by inputting the sub-belief into the policy, while the optimization of the belief encoder parameters by the RL agent is not allowed. Our method is applicable to *any* on-policy algorithm, and we employ A2C in our experiments. We provide the final algorithm in Appendix H.

## 5 EXPERIMENTS

To evaluate our approach, we identify three types of POMDPs: those requiring *long-term memory*, those where *features of the state space are hidden* (and may be inferred from short-term memory), and those with *noisy* observations. Notably, we stress that long-term memory is crucial in POMDPs, whereas short-term memory could be mitigated by stacking frames (e.g., Mnih et al. 2015). We compare our agent to R-A2C and DVRL (Fig. 3), trained in environments from POPGYM (Morad et al., 2023) and our own partially observable version of MINATAR (Young & Tian, 2019).

**Memorization.** The REPEATPREVIOUS environment involves shuffling two decks of cards at the start of each episode and presenting the agent with a card at each time step. The goal is to identify the suit of the card seen 8 time steps earlier. *Our algorithm stands out as the sole method demonstrating mid- to long-term memorization capabilities.* Unlike other methods, notably DVRL which also attempts to learn a belief distribution, WBU provably acquires a suitable representation of the history by learning to maintain a sufficient statistic, thereby explaining its ability to retain past information.

**Hidden features.** We employ a cart pole scenario (STATELESSCARTPOLE) where velocity components of the system are hidden. R-A2C excels rapidly here, capitalizing on short-term memory to infer velocities from the preceding observation, while DVRL is overtaken. Still, WBU eventually reaches R-A2C final performance. We also explore the SPACEINVADERS environment, where the agent takes command of a cannon with the objective of shooting at groups of moving aliens. In the observation, we intentionally concealed the direction of alien movement and confounded friendly and enemy fires. In this more challenging setting, WBU excels by earning the highest rewards.

**Noise.** We explore two types of noise. First, we introduce *Gaussian noise* to the observations of STATELESSCARTPOLE. Second, for SPACEINVADERS, *binary noise* is injected via a radar-like mask obscuring the position of each alien with high probability. Hence, the agent must infer their positions based on previous observations. By leveraging its ability to maintain a belief over the noiseless (latent) state space, WBU demonstrates its resilience to noise and swiftly provides superior solutions, whereas R-A2C eventually achieves comparable performance but with more variance.

**Belief representation.** Ideally, close policy inputs should lead to close values, which would ease its optimization. Thm. 3.3 provides such a representation guarantee and ensures that the representation induced by $\varphi$ captures the value function. To support this, we performed a t-SNE (van der Maaten & Hinton, 2008) on our belief representation at the late stage of training, which projects latent beliefs on a 2D space (Fig. 4). Interestingly, *latent beliefs clustered together have indeed close values*, in line with Thm. 3.3. We reported the belief loss throughout the training phases (Fig. 3). Importantly, unlike other baselines, *our approach distinctly separates the optimization of $\varphi$ and $\bar{\pi}$*. Consequently, *the policy optimization does not influence the representation* which is solely learned via $\varphi$. The decrease in this loss thus relates to improved representation quality for RL.

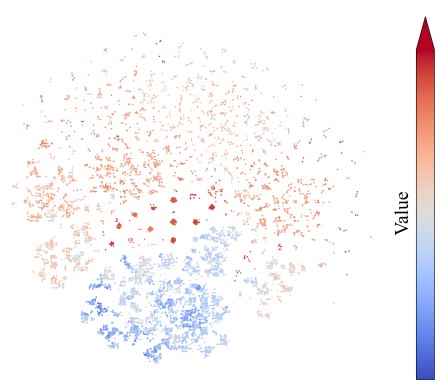

Figure 4: Two-dimensional t-SNE of our belief representation for SPACEINVADERS

## 6 CONCLUSION

WBU provides a principled approach that approximates directly the belief update for POMDPs, in contrast to SOTA methods that uses the RL objective and regularization to attempt to turn the history into a sufficient statistic. By learning the belief and its update rule, we provide strong guarantees on the quality of the belief, its ability to condition the optimal value function, and ultimately, the effectiveness of our algorithm. Our theoretical analysis and experimental results demonstrate the potential of our approach. Overall, WBU provides a promising new direction for RL in POMDPs, with potential applications in a wide range of settings where decision-making is complicated by uncertainty and partial observability, or when guarantees on the agent behaviors are required.

**Future work.** The theory we developed is not limited to the algorithm proposed in our paper. It opens diverse avenues for future work, e.g., on formally verifiable policies for POMDPs, by leveraging the guarantees presented in our framework. We also leave the study and the adaptation of our framework under relaxed assumptions (e.g., in settings akin to the work of Lambrechts et al. 2022) to future work. In addition, Thm 3.2 enables policy optimization through planning in the learned model, as demonstrated in successful model-based RL methods (e.g., Hafner et al. 2021). Scaling to high-dimensional observations (e.g., images) may potentially be computation-intensive due to observation filtering. For this further challenge, we suggest either to modify the WAE-MDP framework by using a stochastic decoder (see Tolstikhin et al. 2018, e.g., via PixelCNN, van den Oord et al. 2016), or learning a lower-dimensional latent observation space synced with the policy with a normalized or (relaxed) discrete prior (e.g., via a WAE-GAN, Tolstikhin et al. 2018). Finally, incorporating bisimulation metrics (Desharnais et al., 2004; Ferns et al., 2011) will strengthen guarantees for belief learning, even though bisimulation is challenging in POMDPs (Castro et al., 2009).

ACKNOWLEDGEMENTS

This research was supported by funding from the Flemish Government under the "Onderzoeksprogramma Artificiële Intelligentie (AI) Vlaanderen" program and was supported by the DESCARTES iBOF project. R. Avalos is supported by the Research Foundation – Flanders (FWO), under grant number 11F5721N. G.A. Perez is also supported by the Belgian FWO "SAILor" project (G030020N). We thank Mathieu Reymond, Denis Steckelmacher, and Mustafa Mert Çelikok for their valuable feedback.

REPRODUCIBILITY STATEMENT

We referenced in the main text the parts of the Appendix presenting the proofs of our Lemma (Appendix D) and Theorems (Appendix F). We also provide the pseudo-code of our algorithm (Appendix H) as well as extra details required to compute our losses (Appendix B, E, and F.2). The code is available at `https://github.com/raphaelavalos/wbu`. Additionally, we provide the details of our hyperparameter search (Appendix I).

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

# APPENDIX

## A   ADDITIONAL RELATED WORK

In the same spirit as in our work, the observability assumption of Lee et al. (2023) aligns perfectly with ours, but is leveraged in a different way, to provide results oriented towards sample complexity in the tabular case, as well as regret bounds when particular types of function approximators are used. Several complexity results and regrets bounds were developed for sub-classes of POMDPS in finite horizon, such as when the observability is well-behaved (weakly revealed POMDPs, Liu et al., 2022), or when the sequential decision making problem admits low rank structure (Liu et al., 2022, Huang et al., 2024). On the contrary, our results hold for general POMDPs; precisely, our constraints rather rely on the training procedure and not on the nature of the POMDP. Finally, we note that RQL-AIS (SeyedSalehi et al., 2023) presents an analysis of recurrent Q-learning for the tabular setting which also includes value difference bounds where their latent state is the (discrete) hidden state of an RNN. While theoretically interesting, their bounds are generally intractable as they rely on global metrics requiring them to be minimized over the whole history and action space. This limitation is made clear in their algorithm as they use the local version of those metrics as regularization losses. In our case, because we focus on on-policy metrics (local losses) our theoretical guarantees hold in practice.

**Bisimulation.** One of our result involves abstraction quality guarantees (Theorem 3.2), formalized as the closeness of the value function of the original and latent models. In spirit, this result can be related to *bisimulation*. Bisimulation is a behavioral *equivalence relation* between states and models whose properties include trajectory and value *equivalence* (Larsen & Skou, 1991; Givan et al., 2003). However, due to their "all or nothing" nature, bisimulation relations are often too restrictive; intuitively, states that share close rewards and dynamics but exhibit slight numerical differences are treated as being completely different. Bisimulation pseudometrics (de Alfaro et al., 2003; Desharnais et al., 2004) generalize bisimulation by assigning a distance between states and quantifying their bisimilarity instead of yielding binary signals indicating whether they are bisimilar or not. Specifically, real-valued (discounted) quantitative properties (de Alfaro et al., 2003; Chatterjee et al., 2010) as well as expected returns (Ferns et al., 2011) are guaranteed to be Lipschitz continuous w.r.t. the related bisimulation distance (compare with the *almost-Lipschitzness* of our Theorem 3.3).

Works on representation learning for RL (e.g., Gelada et al., 2019; van der Pol et al., 2020; Castro et al., 2021; Zhang et al., 2021; Zang et al., 2022; Delgrange et al., 2023) use bisimulation metrics to optimize the representation quality offered by a latent space, and aim at learning representations which capture the bisimulation distance, so that two states close in the representation provide close and relevant information for learning the policy. Precisely, *such works seek the Lipschitz continuity of the optimal value function with regards to the distance between pairs of points in the latent space*.

Note that while our work builds upon the WAE-MDP framework (Delgrange et al., 2023), which learns a latent model based on bisimulation metrics, our work does not learn any bisimulation-like metric in the underlying latent space of the POMDP. Instead, *we prioritize what truly matters in the context of representation for RL: the Lipschitz-continuity of the optimal value function*. In particular, the attentive reader may notice that, in contrast to usual representation bounds in MDPs (e.g., Gelada et al., 2019; Delgrange et al., 2022), our bound involves the almost Lipschitzness the value function, as well as various additional losses linked to the belief update and observation functions, which underlines the challenge of providing such guarantees. Indeed, bisimulation in the context of POMDPs represents a major challenge (e.g., Castro et al., 2009): intuitively, in the context of POMDPs, the bisimulation equivalence is not defined on the state space but rather on the belief space induced by the different components of the POMDP. Notably, even if the POMDP is finite, i.e., state, action, and observation spaces are finite, the belief space can be represented as the usual (uncountable space) simplex, which prevents the application of bisimulation algorithms developed in the context of countable MDPs.

## B   THE BELIEF UPDATE RULE

Let $\overline{\mathcal{P}} = \langle \overline{\mathcal{M}}, \Omega, \overline{\mathcal{O}} \rangle$ be a latent POMDP with underlying MDP $\overline{\mathcal{M}} = \langle \overline{\mathcal{S}}, \mathcal{A}, \overline{\mathbf{P}}, \overline{\mathcal{R}}, \bar{s}_I, \gamma \rangle$. At step $t \geqslant 0$, assume that the current latent belief is $\bar{b}_t \in \overline{\mathcal{B}}$. Then, when $a_t \in \mathcal{A}$ is executed and $o_{t+1} \in \Omega$

is observed, $\bar{b}_t$ is updated according to $\bar{\tau}(\bar{b}_t, a_t, o_{t+1})$ as follows, so that for any (believed) state $\bar{s}_{t+1} \in \bar{\mathcal{S}}$,

$$\bar{b}_{t+1}(\bar{s}_{t+1}) = \frac{\mathbb{E}_{\bar{s}_t \sim \bar{b}_t} \, \overline{\mathbf{P}}(\bar{s}_{t+1} \mid \bar{s}_t, a_t) \cdot \overline{\mathcal{O}}(o_{t+1} \mid \bar{s}_{t+1})}{\mathbb{E}_{\bar{s}_t \sim \bar{b}_t} \, \mathbb{E}_{\bar{s}' \sim \overline{\mathbf{P}}(\cdot \mid \bar{s}_t, \bar{a}_t)} \, \overline{\mathcal{O}}(o_{t+1} \mid \bar{s}')}. \tag{9}$$

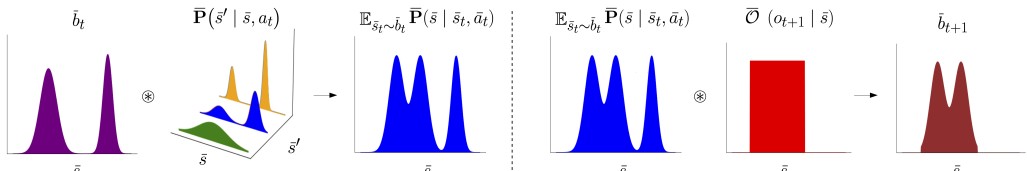

Figure 5: The belief update rule: (*left*) transformation of the current belief $\bar{b}_t$ with the transition probability function $\overline{\mathbf{P}}$, evaluated on the current action $a_t$, into the next state probability density; (*right*) filtering out the next states that could not have produced the next observation $o_{t+1}$.

The belief update rule $\bar{\tau}$ is divided in two steps (cf. Fig. 5). First, the current belief distribution $\bar{b}_t$ is used to marginalize the latent transition function $\overline{\mathbf{P}}$ over the believed latent states, to further infer the distribution over the possible next states. This first part corresponds to computing the probability of going to the next believed state $\bar{s}_{t+1}$ from the possible states $\bar{s}_t$ witnessed by the current belief $\bar{b}_t$. Second, the next observation $o_{t+1}$ is used to filter the resulting density based on the observation function: believed latent states that do not share the observation $o_{t+1}$ perceived are filtered out according to $\overline{\mathcal{O}}(o_{t+1} \mid \bar{s}_{t+1})$. It is worth noting that the latent model is learned from $\mathcal{P}^\uparrow$, whose observation function is deterministic. Without modelling the latent observation function $\overline{\mathcal{O}}$ as a normal distribution, the second part of the belief update would need to eliminate all next states with different observations — which is not gradient descent friendly. The third operation (not present in Fig. 5) normalizes the output of the observation filtering to obtain a probability density.

## C  DIRAC MEASURES

In this work, we consider the *Dirac delta function* $\delta$ as a *measure*. Specifically, this means that for any complete, separable space $\mathcal{X}$ and point $a \in \mathcal{X}$, the *Dirac measure* with impulse $a$ is $\delta_a \in \Delta(\mathcal{X})$ and satisfies $\delta_a(A) = 1$ if $a \in A$ and $\delta_a(A) = 0$ otherwise, for any $A \in \Sigma(\mathcal{X})$. Interesting properties of the Dirac measure include $\delta_a = \lim_{\sigma \to 0} \mathcal{N}(a, \sigma^2)$, where $\mathcal{N}(a, \sigma^2)$ is the normal distribution with mean $a$ and variance $\sigma^2$, and $\int_{\mathcal{X}} \delta_a(x) f(x) \, dx = f(a)$ for any compactly supported function $f$.

## D  PROOF OF LEMMA 3.1: STATIONARITY OVER HISTORIES

Let us formally restate the Lemma:

**Lemma D.1.** *Let $\mathcal{P}$ be an episodic POMDP with action space $\mathcal{A}$ and observation space $\Omega$, and $\bar{\pi}$ be a latent policy conditioned on the beliefs of a latent POMDP $\overline{\mathcal{P}}$ that shares the action and observation spaces of $\mathcal{P}$ while being lifted in $\mathcal{P}$ via the belief encoder, i.e., $a \sim \bar{\pi}(\cdot \mid \varphi^*(h))$ for any $h \in (\mathcal{A} \cdot \Omega)^*$. Then, there is a well defined stationary distribution $\mathcal{H}_{\bar{\pi}} \in \Delta((\mathcal{A} \cdot \Omega)^*)$ over histories drawn at the limit from the interaction of the RL agent with $\mathcal{P}$, when it operates under the latent policy $\bar{\pi}$.*

*Proof sketch.* Build a *history unfolding* as the MDP whose state space consists of all histories and keeps track of the current history of $\mathcal{P}$ at any time of the interaction. The resulting MDP remains episodic since it is equivalent to $\mathcal{P}$: the former mimics the behaviors of the latter under $\bar{\pi}$. All episodic processes are *ergodic* (e.g., Huang 2020), which guarantees the existence of such a distribution. $\qquad\square$

We dedicate this Section to formally detailing and proving every claim of this proof sketch. Before going further, we formally define the notion of *episodic process*, and we further introduce the notions of *memory-based policies*, *Markov Chains*, and *limiting distributions in Markov Chains*.

## D.1 PRELIMINARIES

Recall by Assumption 2 that the environment $\mathcal{P}$ is an episodic process. We *formally* recall the notion of *episodic process*:

**Definition D.2** (Episodic RL process). *The RL procedure is* episodic *iff the environment $\mathcal{P}$ embeds a special* reset state $s_{\mathsf{reset}} \in \mathcal{S}$ *so that*

 *(i) under any policy $\pi$, the environment is almost surely eventually reset:*
$$\mathbb{P}^{\mathcal{M}}_{\pi}(\{\, s_{0:\infty}, a_{0:\infty} \mid \exists t > 0, s_t = s_{\mathsf{reset}} \,\}) = 1;$$

 *(ii) when reset, the environment transitions to the initial state: $\mathbf{P}(s_I \mid s_{\mathsf{reset}}, a) > 0$ and $\mathbf{P}(\mathcal{S}\backslash\{\, s_I, s_{\mathsf{reset}} \,\} \mid s_{\mathsf{reset}}, a) = 0$ for all $a \in \mathcal{A}$, and the distribution is the same for all actions, i.e., for all $a, a' \in \mathcal{A}$, $\mathbf{P}(\cdot \mid s_{\mathsf{reset}}, a) = \mathbf{P}(\cdot \mid s_{\mathsf{reset}}, a')$;*

 *(iii) the reset state is observable: there is an observation $o^{\star} \in \Omega$ so that $\mathcal{O}(o^{\star} \mid s', a) = 0$ when $s' \neq s_{\mathsf{reset}}$, and $\mathcal{O}(\cdot \mid s_{\mathsf{reset}}, a) = \delta_{o^{\star}}$ for $a \in \mathcal{A}$.*

*An* episode *is a history $\langle a_{0:T-1}, o_{1:T} \rangle$ where $\mathcal{O}(o_1 \mid s_I, a_0) > 0$ and $o_T = o^{\star}$.*

**Encoding memory through Mealy machines.** *Policies* are building blocks to define the probability space of any MDP. To deal with policies whose decisions are based on unrolling histories, we formally define the notion of *memory* in policies.

**Definition D.3** (Policy as Mealy Machine). *Given an MDP $\mathcal{M} = \langle \mathcal{S}, \mathcal{A}, \mathbf{P}, \mathcal{R}, s_I, \gamma \rangle$, any policy for $\mathcal{M}$ can be encoded as a stochastic Mealy machine $\pi = \langle Q, \pi_{\alpha}, \pi_{\mu}, q_I \rangle$ as follows: $Q$ is a set of memory states; $\pi_{\alpha} \colon \mathcal{S} \times Q \to \Delta(\mathcal{A})$ is the next action function; $\pi_{\mu} \colon \mathcal{S} \times Q \times \mathcal{A} \times \mathcal{S} \to \Delta(Q)$ is the memory update function; and $q_I$ is the initial memory state.* [2]

*Example* 1 (Stationary policy). A stationary policy $\pi$ can be encoded as any Mealy machine $\pi$ with memory space $Q$ where $|Q| = 1$.

*Example* 2 (Latent policy). Let $\mathcal{P} = \langle \mathcal{M}, \Omega, \mathcal{O} \rangle$ with underlying MDP $\mathcal{M} = \langle \mathcal{S}, \mathcal{A}, \mathbf{P}, \mathcal{R}, s_I, \gamma \rangle$ and the latent space model $\overline{\mathcal{P}}$ with initial state $\bar{s}_I$ be the POMDPs of Lemma D.1. Then, any latent (stationary) policy $\bar{\pi} \colon \overline{\mathcal{B}} \to \Delta(\mathcal{A})$ conditioned on the belief space $\overline{\mathcal{B}}$ of $\overline{\mathcal{P}}$ can be executed in the belief MDP $\mathcal{M}_{\mathcal{B}}$ of $\mathcal{P}$ via the Mealy machine $\bar{\pi}' = \langle \overline{\mathcal{B}}, \bar{\pi}_{\alpha}, \bar{\pi}_{\mu}, \delta_{\bar{s}_I} \rangle$, keeping track in its memory of the current latent belief $\bar{b} \in \overline{\mathcal{B}}$ inferred by our belief encoder $\varphi$. This enables the agent to take its decisions solely based on the latter: $\bar{\pi}_{\alpha}(\cdot \mid b, \bar{b}) = \bar{\pi}(\cdot \mid \bar{b})$. When the belief MDP transitions to the next belief $b'$, the memory is then updated according to the observation dynamics:

$$\bar{\pi}_{\mu}(\bar{b}' \mid b, \bar{b}, a, b') = \frac{\mathbb{E}_{s \sim b} \mathbb{E}_{s' \sim \mathbf{P}(\cdot|s,a)} \mathbb{E}_{o' \sim \mathcal{O}(\cdot|s',a)} \delta_{\varphi(\bar{b}, a, o')}(\bar{b}') \cdot \delta_{\tau(b, a, o')}(b')}{\mathbb{E}_{s \sim b} \mathbb{E}_{s' \sim \mathbf{P}(\cdot|s,a)} \mathbb{E}_{o' \sim \mathcal{O}(\cdot|s',a)} \delta_{\tau(b, a, o')}(b')} \quad \text{if } b' \neq \delta_{s_{\mathsf{reset}}},$$

$$\bar{\pi}_{\mu}(\cdot \mid b, \bar{b}, a, \delta_{s_{\mathsf{reset}}}) = \delta_{\bar{s}_{\mathsf{reset}}} \qquad\qquad \text{otherwise (to fulfil the episodic constraint).}$$

Note that $\bar{\pi}_{\mu}$ is simply obtained by applying the usual conditional probability rule: $\bar{\pi}_{\mu}(\bar{b}' \mid b, \bar{b}, a, b') = {\Pr(b', \bar{b}'|b, \bar{b}, a)}/{\Pr(b'|b, \bar{b}, a)}$, where $\Pr(b', \bar{b}' \mid b, \bar{b}, a) = \mathbb{E}_{s \sim b} \mathbb{E}_{s' \sim \mathbf{P}(\cdot|s,a)} \mathbb{E}_{o' \sim \mathcal{O}(\cdot|s',a)} \delta_{\varphi(\bar{b}, a, o')}(\bar{b}') \cdot \delta_{\tau(b, a, o')}(b')$ and $\Pr(b' \mid b, \bar{b}, a) = \mathbf{P}_{\mathcal{B}}(b' \mid b, a)$ since the next *original* belief state is independent of the current *latent* belief state.

**Definition D.4** (Markov Chain). *A* Markov Chain *(MC) is an MDP whose action space $\mathcal{A}$ consists of a singleton, i.e., $|\mathcal{A}| = 1$. Any MDP $\mathcal{M} = \langle \mathcal{S}, \mathcal{A}, \mathbf{P}, \mathcal{R}, s_I, \gamma \rangle$ and memory-based policy $\pi = \langle Q, \pi_{\alpha}, \pi_{\mu}, q_I \rangle$ induces a Markov Chain*
$$\mathcal{M}^{\pi} = \langle \mathcal{S} \times Q, \mathbf{P}_{\pi}, \mathcal{R}_{\pi}, \langle s_I, q_I \rangle, \gamma \rangle,$$
*where:*

---

[2] for convenience, we use the next state as input of the memory update function in addition to the current state of the MDP. This is non-standard but we contend that this change is harmless. Indeed, one may construct an MDP with a dummy state $\langle s, a, s' \rangle$ which is entered each time the system transitions from $s$ to $s'$ when $a$ is chosen, before going to $s'$. Then, it suffices to change the discount to $\sqrt{\gamma}$ to compensate for the addition of this extra transition.

- *the state space consists of the product of the original state space and the memory of $\pi$;*

- *the transition function embeds the next action and the policy update functions from the policy, i.e.,*

$$\mathbf{P}_\pi\big(\langle s', q'\rangle \mid \langle s, q\rangle\big) = \mathop{\mathbb{E}}_{a \sim \pi_\alpha(\cdot \mid s, q)} \pi_\mu\big(q' \mid s, q, a, s'\big) \cdot \mathbf{P}\big(s' \mid s, a\big), \text{ and}$$

- *the rewards are averaged over the possible actions produced by the next action function, i.e.,* $\mathcal{R}_\pi(\langle s, q\rangle) = \mathbb{E}_{a \sim \pi_\alpha(\cdot \mid s, q)} \mathcal{R}(s, a).$

*The probability measure $\mathbb{P}_\pi^{\mathcal{M}}$ is actually the unique probability measure defined over the measurable infinite trajectories of the MC $\mathcal{M}^\pi$* (Baier & Katoen, 2008). *We omit the memory dimension $Q$ when $\pi$ is stationary (i.e., memoryless).*

**Limiting distributions.** We now formally define the distribution over states encountered at the limit when an agent operates in an MDP under a given policy, as well as the conditions of existence of such a distribution. To that aim, we distinctly separate the discrete and the continuous cases, where the conditions to reach the stationarity of the system and the properties that hold at the limit of the interaction are different. In the following, we write $\mathcal{M}[s]$ for the MDP where we change the initial state $s_I$ of $\mathcal{M}$ by $s \in \mathcal{S}$.

(a) *Finite state spaces.* Let $\mathcal{M}$ be an MDP with *finite* state space $\mathcal{S}$ and $\pi$ be a stationary policy for $\mathcal{M}$. The measure $\xi_\pi^t : \mathcal{S} \to \Delta(\mathcal{S})$ with $\xi_\pi^t(s' \mid s) = \mathbb{P}_\pi^{\mathcal{M}[s]}\big(\{ s_{0:\infty}, a_{0:\infty} \mid s_t = s' \}\big)$ is the distribution giving the probability for the agent of being in each state of $\mathcal{M}[s]$ after exactly $t$ steps.

**Definition D.5** (BSCC). *The subset $B \subseteq \mathcal{S}$ is a* strongly connected component *(SCC) of $\mathcal{M}^\pi$ if for any pair of states $s, s' \in B$, there is a $t \in \mathbb{N}$ so that $\xi_\pi^t(s' \mid s) > 0$. It is a* bottom SCC *(BSCC) if (i) $B$ is a maximal SCC, and (ii) for each $s \in B$, $\mathbf{P}_\pi(B \mid s) = 1$.*

The unique *stationary distribution* of $B$ is $\xi_\pi \in \Delta(B)$, defined as $\xi_\pi(s) = \mathbb{E}_{\dot{s} \sim \xi_\pi} \mathbf{P}_\pi(s \mid \dot{s}) = \lim_{T \to \infty} \frac{1}{T} \sum_{t=0}^T \xi_\pi^t(s \mid s_\perp)$ for any $s_\perp \in B$ (Baier & Katoen, 2008).

The *period* of a state $s \in \mathcal{S}$ is the greatest common divisor of the set $\{ t > 0 \mid \xi_\pi^t(s \mid s) > 0 \}$. A BSCC $B$ is *aperiodic* iff all states of $B$ have period one.

An MDP $\mathcal{M}$ is *ergodic* under the policy $\pi$ if the state space of $\mathcal{M}^\pi$ consists of a unique aperiodic BSCC. In that case, $\xi_\pi = \lim_{t \to \infty} \xi_\pi^t(\cdot \mid s)$ for all $s \in \mathcal{S}$.

(b) *General state spaces.* For convenience, we focus on a subset of well-behaved Markov chains, namely *Harris chains* (e.g., Athreya & Ney 1978; Revuz 1984), which are guaranteed to converge to a stationary distribution at the limit under some conditions. Let $\mathcal{M}$ be an MDP with a *complete, separable* state space $\mathcal{S}$. We denote by $\Sigma(\mathcal{S})$ the set of measurable events of $\mathcal{S}$. Let $\pi$ be a stationary policy for $\mathcal{M}$. The measure $\xi_\pi^t : \mathcal{S} \to \Delta(\mathcal{S})$ with $\xi_\pi^t(S' \mid s) = \mathbb{P}_\pi^{\mathcal{M}[s]}\big(\{ s_{0:\infty}, a_{0:\infty} \mid s_t \in S' \}\big)$ is the distribution giving the probability for the agent of being in a state $s' \in S$ after exactly $t$ steps, starting from $s \in \mathcal{S}$, for any $S' \in \Sigma(\mathcal{S})$. Denote the *reachability* event to $H \in \Sigma(\mathcal{S})$ as

$$\Diamond H = \{ \langle s_{0:\infty}, a_{0:\infty}\rangle \in \mathit{Traj}_{\mathcal{M}} \mid \text{there is a } t \in \mathbb{N} \text{ so that } s_t \in H \}.$$

**Definition D.6** (Recurrent Harris chain). *The induced MC $\mathcal{M}^\pi$ is a* recurrent Harris chain *if there exist a measurable set $H \in \Sigma(\mathcal{S})$, an $\epsilon \in (0, 1]$, and a probability measure $\rho \in \Delta(\mathcal{S})$ so that*

*(i) $H$ is* recurrent, *i.e., $\mathbb{P}_\pi^{\mathcal{M}[s]}(\Diamond H) = 1$ for all $s \in \mathcal{S}$; and*

*(ii) $H$ is a* small set, *i.e., for any $s \in H$ and $C \in \Sigma(\mathcal{S})$, $\mathbf{P}_\pi(C \mid s) \geqslant \epsilon \rho(C)$.*

Recurrent Harris chains are guaranteed to have a stationary measure $\xi_\pi \in \Delta(\mathcal{S})$ (Durrett, 2010, Theorem 5.8.9), so that $\xi_\pi(S) = \mathbb{E}_{\dot{s} \sim \xi_\pi} \mathbf{P}_\pi(S \mid \dot{s})$ for any $S \in \Sigma(\mathcal{S})$.

Similarly to the discrete case, we say that a Harris chain is *aperiodic* if and only if there is a moment from which the agent is able to return to the region $H$ *at any time*: for any $s \in H$, there exists a moment $i \in \mathbb{N}$ so that for any $j \geqslant i$, $\xi_\pi^j(H \mid s) > 0$.

Assume $\mathcal{M}_\pi$ is an aperiodic recurrent Harris chain with stationary distribution $\xi_\pi$, then $\xi_\pi^t$ converges to $\xi_\pi$ *in total variation* as $t$ goes to $\infty$ (Durrett, 2010, Theorem 5.8.12), i.e.,

$$\lim_{t \to \infty} d_{TV}\left(\xi_\pi^t(\cdot \mid s), \xi_\pi\right) = 0$$

for all $s \in \mathcal{S}$.

To provide such a stationary distribution over histories, we define a *history unfolding* MDP, where the state space keeps track of the current history of $\mathcal{P}$ during the interaction. We then show that this history MDP is *equivalent* to $\mathcal{P}$ under $\bar{\pi}$.

## D.2  HISTORY UNFOLDING

Let us define the *history unfolding* MDP $\mathcal{M}_\mathcal{H}$, which consists of the tuple $\langle \mathcal{S}_\mathcal{H}, \mathcal{A}, \mathbf{P}_\mathcal{H}, \mathcal{R}_\mathcal{H}, \star, \gamma \rangle$, where:

- the state space consists of the set of all the possible histories (i.e., sequence of actions and observations) that can be encountered in $\mathcal{P}$, i.e., $\mathcal{S}_\mathcal{H} = (\mathcal{A} \cdot \Omega)^* \cup \{h_{\text{reset}}\}$, where we particularly denote by $\epsilon$ the empty string with $\tau^*(\epsilon) = \delta_{s_I}$ (no action nor observation has been perceived yet), and $h_{\text{reset}}$ is a special reset state;

- the transition function maps the current history to the belief space to infer the distribution over the next possible observations, i.e.,

$$\mathbf{P}_\mathcal{H}\left(h' \mid h, a\right) = \mathop{\mathbb{E}}_{s \sim \tau^*(h)} \mathop{\mathbb{E}}_{s' \sim \mathbf{P}(\cdot \mid s, a)} \mathop{\mathbb{E}}_{o' \sim \mathcal{O}(\cdot \mid s', a)} \delta_{h \cdot a \cdot o'}\left(h'\right) \qquad \text{if } \tau^*(h) \neq \delta_{s_{\text{reset}}}, \text{ and}$$

$$\mathbf{P}_\mathcal{H}\left(h' \mid h, a\right) = \mathbf{P}_\mathcal{B}(\delta_{s_{\text{reset}}} \mid \delta_{s_{\text{reset}}}, a) \cdot \delta_{h_{\text{reset}}}\left(h'\right) + \mathbf{P}_\mathcal{B}(\delta_{s_I} \mid \delta_{s_{\text{reset}}}, a) \cdot \delta_\epsilon\left(h'\right) \qquad \text{otherwise,}$$

where $h \cdot a \cdot o'$ is the concatenation of $a, o'$ with the history $h = \langle a_{0:T-1}, o_{1:T} \rangle$, resulting in the history $\langle a_{0:T}, o_{1:T+1} \rangle$ so that $a_T = a$ and $o_{T+1} = o'$; and

- the reward function maps the history to the belief space as well, which enables to infer the expected rewards obtained in the states over the this belief, i.e., $\mathcal{R}_\mathcal{H}(h, a) = \mathbb{E}_{s \sim \tau^*(h)} \mathcal{R}(s, a)$.

We now aim at showing that, under the latent policy $\bar{\pi}$, the POMDP $\mathcal{P}$ and the MDP $\mathcal{M}_\mathcal{H}$ are *equivalent*. More formally, we are looking for an equivalence relation between two probabilistic models, so that the latter induce the same behaviors, or in other words, the same expected return. We formalize this equivalence relation as a *stochastic bisimulation* between $\mathcal{M}_\mathcal{B}$ (that we know being an MDP formulation of $\mathcal{P}$) and $\mathcal{M}_\mathcal{H}$.

**Definition D.7** (Bisimulation). *Let $\mathcal{M} = \langle \mathcal{S}, \mathcal{A}, \mathbf{P}, \mathcal{R}, s_I, \gamma \rangle$ be an MDP. A stochastic bisimulation $\equiv$ on $\mathcal{M}$ is a behavioral equivalence between states $s_1, s_2 \in \mathcal{S}$ so that, $s_1 \equiv s_2$ iff*

1. *$\mathcal{R}(s_1, a) = \mathcal{R}(s_2, a)$, and*

2. *$\mathbf{P}(T \mid s_1, a) = \mathbf{P}(T \mid s_2, a)$,*

*for each action $a \in \mathcal{A}$ and equivalence class $T \in \mathcal{S}/ \equiv$.*

Properties of bisimulation include trajectory equivalence and the equality of their optimal expected return (Larsen & Skou, 1989; Givan et al., 2003). The relation can be extended to compare two MDPs by considering the disjoint union of their state space.

**Lemma D.8.** *Let $\mathcal{P}$ be the POMDP of Lemma D.1, and $\bar{\pi}: \overline{\mathcal{B}} \to \Delta(\mathcal{A})$ be a latent policy conditioned on the beliefs of a latent space model of $\mathcal{P}$. Define the stationary policy $\bar{\pi}^\clubsuit: \mathcal{S}_\mathcal{H} \to \Delta(\mathcal{A})$ for $\mathcal{M}_\mathcal{H}$ as $\bar{\pi}^\clubsuit(\cdot \mid h) = \bar{\pi}(\cdot \mid \varphi^*(h))$, and the policy $\bar{\pi}^\diamondsuit$ for $\mathcal{M}_\mathcal{B}$ encoded by the Mealy machine detailed in Example 2. Then, $\mathcal{M}_\mathcal{H}^{\bar{\pi}^\clubsuit}$ and $\mathcal{M}_\mathcal{B}^{\bar{\pi}^\diamondsuit}$ are in stochastic bisimulation.*

*Proof.* First, note that the MC $\mathcal{M}_\mathcal{B}^{\bar{\pi}^\diamondsuit}$ is defined as the tuple $\langle \mathcal{B} \times \overline{\mathcal{B}}, \mathbf{P}_{\bar{\pi}^\diamondsuit}, \mathcal{R}_{\bar{\pi}^\diamondsuit}, \langle b_I, \bar{b}_I \rangle, \gamma \rangle$ so that

$$\mathbf{P}_{\bar{\pi}^\diamondsuit}\left(b', \bar{b}' \mid b, \bar{b}\right) = \mathop{\mathbb{E}}_{a \sim \bar{\pi}(\cdot \mid \bar{b})} \bar{\pi}_\mu\left(\bar{b}' \mid b, \bar{b}, a, b'\right) \cdot \mathbf{P}_\mathcal{B}\left(b' \mid b, a\right)$$

$$= \mathop{\mathbb{E}}_{a \sim \bar{\pi}(\cdot | \bar{b})} \mathop{\mathbb{E}}_{s \sim b} \mathop{\mathbb{E}}_{s' \sim \mathbf{P}(\cdot | s, a)} \mathop{\mathbb{E}}_{o' \sim \mathcal{O}(\cdot | s, a)} \delta_{\varphi(\bar{b}, o, a)}(\bar{b}') \cdot \delta_{\tau(b, o, a)}(b'), \text{ and}$$

$$\mathcal{R}_{\bar{\pi}^\diamond}(b, \bar{b}) = \mathop{\mathbb{E}}_{a \sim \bar{\pi}(\cdot | \bar{b})} \mathop{\mathbb{E}}_{s \sim b} \mathcal{R}(s, a). \qquad \text{(cf. Definition D.4)}$$

Define the relation $\equiv_\varphi^\tau$ as the set $\{\langle h, \langle b, \bar{b} \rangle \rangle \in \mathcal{S}_\mathcal{H} \times (\mathcal{B} \times \bar{\mathcal{B}}) \mid \tau^*(h) = b \text{ and } \varphi^*(h) = \bar{b}\}$. We show that $\equiv_\varphi^\tau$ is a bisimulation relation between the states of $\mathcal{M}_\mathcal{H}^{\bar{\pi}^\clubsuit}$ and $\mathcal{M}_\mathcal{B}^{\bar{\pi}^\diamond}$. Let $h = \langle a_{0:T-1}, o_{1:T} \rangle \in \mathcal{S}_\mathcal{H}$, $b \in \mathcal{B}$, and $\bar{b} \in \bar{\mathcal{B}}$ so that $o_T \neq o^\star$ and $h \equiv_\varphi^\tau \langle b, \bar{b} \rangle$:

1. $\mathcal{R}_{\bar{\pi}^\clubsuit}(h) = \mathbb{E}_{a \sim \bar{\pi}(\cdot | \varphi^*(h))} \mathbb{E}_{s \sim \tau^*(h)} \mathcal{R}(s, a) = \mathbb{E}_{a \sim \bar{\pi}(\cdot | \bar{b})} \mathbb{E}_{s \sim b} \mathcal{R}(s, a) = \mathcal{R}_{\bar{\pi}^\diamond}(b, \bar{b})$.

2. Note that histories are deterministically mapped to a *unique* belief (resp. latent belief) state through the belief update function $\tau$ (resp. $\varphi$). However, $\tau$ and $\varphi$ *are not injective*, as multiple histories can lead to the same belief (resp. latent belief) state. Then, each equivalence class $T$ can be associated to a single belief and latent belief pair, aggregating multiple histories that lead to the same pair, through $\tau$ and $\varphi$. Concretely, let $b' \in \mathcal{B}$, $\bar{b}' \in \bar{\mathcal{B}}$, an equivalence class of $\equiv_\varphi^\tau$ has the form $T = [\langle b', \bar{b}' \rangle]_{\equiv_\varphi^\tau}$ so that

   (a) the intersection of $[\langle b', \bar{b}' \rangle]_{\equiv_\varphi^\tau}$ with $\mathcal{S}_\mathcal{H}$ is the set $\{h' \in \mathcal{S}_\mathcal{H} \mid \tau^*(h') = b' \text{ and } \varphi^*(h') = \bar{b}'\}$, and

   (b) the intersection of $[\langle b', \bar{b}' \rangle]_{\equiv_\varphi^\tau}$ with the state space of $\mathcal{M}_\mathcal{B}^{\bar{\pi}^\diamond}$ merely consists of the pair $\langle b', \bar{b}' \rangle$.

   Therefore,

$$\mathbf{P}_{\bar{\pi}^\clubsuit}\left([\langle b', \bar{b}' \rangle]_{\equiv_\varphi^\tau} \mid h\right)$$

$$= \int_{[\langle b', \bar{b}' \rangle]_{\equiv_\varphi^\tau}} \mathop{\mathbb{E}}_{a \sim \bar{\pi}(\cdot | \varphi^*(h))} \mathop{\mathbb{E}}_{s \sim \tau^*} \mathop{\mathbb{E}}_{s' \sim \mathbf{P}(\cdot | s, a)} \mathop{\mathbb{E}}_{o' \sim \mathcal{O}(\cdot | s', a)} \delta_{h \cdot a \cdot o'}(h') \, dh'$$

$$= \int_{\mathcal{S}_\mathcal{H}} \mathop{\mathbb{E}}_{a \sim \bar{\pi}(\cdot | \varphi^*(h))} \mathop{\mathbb{E}}_{s \sim \tau^*(h)} \mathop{\mathbb{E}}_{s' \sim \mathbf{P}(\cdot | s, a)} \mathop{\mathbb{E}}_{o' \sim \mathcal{O}(\cdot | s', a)} \delta_{h \cdot a \cdot o'}(h') \cdot \delta_{\tau^*(h')}(b') \cdot \delta_{\varphi^*(h')}(\bar{b}') \, dh'$$
$$\text{(by definition of } [\langle b', \bar{b}' \rangle]_{\equiv_\varphi^\tau})$$

$$= \mathop{\mathbb{E}}_{a \sim \bar{\pi}(\cdot | \varphi^*(h))} \mathop{\mathbb{E}}_{s \sim \tau^*(h)} \mathop{\mathbb{E}}_{s' \sim \mathbf{P}(\cdot | s, a)} \mathop{\mathbb{E}}_{o' \sim \mathcal{O}(\cdot | s', a)} \delta_{\tau^*(h \cdot a \cdot o')}(b') \cdot \delta_{\varphi^*(h \cdot a \cdot o')}(\bar{b}')$$

$$= \mathop{\mathbb{E}}_{a \sim \bar{\pi}(\cdot | \bar{b})} \mathop{\mathbb{E}}_{s \sim b} \mathop{\mathbb{E}}_{s' \sim \mathbf{P}(\cdot | s, a)} \mathop{\mathbb{E}}_{o' \sim \mathcal{O}(\cdot | s', a)} \delta_{\tau(b, a, o')}(b') \cdot \delta_{\varphi(\bar{b}, a, o')}(\bar{b}') \qquad \text{(since } h \equiv_\varphi^\tau \langle b, \bar{b} \rangle)$$

$$= \mathbf{P}_{\bar{\pi}^\diamond}(b', \bar{b}' \mid b, \bar{b})$$

$$= \mathbf{P}_{\bar{\pi}^\diamond}\left([\langle b', \bar{b}' \rangle]_{\equiv_\varphi^\tau} \mid b, \bar{b}\right)$$

By 1 and 2, we have that $\mathcal{M}_\mathcal{H}$ and $\mathcal{M}_\mathcal{B}$ are in bisimulation under the equivalence relation $\equiv_\varphi^\tau$, when the policies $\bar{\pi}^\clubsuit$ and $\bar{\pi}^\diamond$ are respectively executed in the two models. $\qquad\square$

**Corollary D.9.** *The agent behaviors, formulated through the expected return, that are obtained by executing the policies respectively in the two models are the same:* $\mathbb{E}_{\bar{\pi}^\clubsuit}^{\mathcal{M}_\mathcal{H}}\left[\mathcal{R}(s_I, a_0) + \sum_{t=1}^\infty \gamma^t \cdot \mathcal{R}_\mathcal{H}(a_{0:t-1}, o_{1:t})\right] = \mathbb{E}_{\bar{\pi}^\diamond}^{\mathcal{M}_\mathcal{B}}\left[\sum_{t=0}^\infty \gamma^t \cdot \mathcal{R}_\mathcal{B}(b_t, a_t)\right].$

*Proof.* Follows directly from (Larsen & Skou, 1989; Givan et al., 2003): the bisimulation relation implies the equality of the maximum expected return in the two models, where the maximum is taken over the set of all stationary policies of the two models. Since we consider MCs and not MDPs, the models are purely stochastic, so there is no nondeterministic choice linked to the choice of action, and then there is only one possible expected return, which yields the result. $\qquad\square$

Note that we omitted the super script of $\bar{\pi}^\clubsuit$ in the main text; we directly considered $\bar{\pi}$ as a policy conditioned over histories, by using the exact same definition.

### D.3 Existence of a Stationary Distribution over Histories

Now that we have proven that the history unfolding is equivalent to the belief MDP, we now have all the ingredients to prove Lemma D.1.

*Proof.* By definition of $\mathcal{M}_{\mathcal{H}}$, the execution of $\bar{\pi}^{\clubsuit}$ is guaranteed to remain an episodic process.

(a) Assume that the number of branchings in the history space is finite when unrolling all possible episodes of the environment when $\bar{\pi}^{\clubsuit}$ is executed. Then, $\mathcal{M}_{\mathcal{H}}^{\bar{\pi}^{\clubsuit}}$ is finite as well. Every episodic process in MDP with countable state spaces is ergodic (Huang, 2020), there is thus a unique stationary distribution $\mathcal{H}_{\bar{\pi}^{\clubsuit}} = \lim_{t \to \infty} \xi_{\bar{\pi}^{\clubsuit}}^t(\cdot \mid \star)$ defined over the state space of $\mathcal{M}_{\mathcal{H}}$.

(b) More generally, assume now that $\Omega$ is a complete and separable space. We first show that $\mathcal{M}_{\mathcal{H}}^{\bar{\pi}^{\clubsuit}}$ defines a recurrent Harris chain. Take $H = \{h_{\text{reset}}\}$; intuitively the event of visiting the reset state is triggered infinitely often with probability one under any policy, which ensures the history unfolding to be Harris recurrent. To see why, notice first that by Assumption 2, $\mathcal{M}_{\mathcal{H}}$ is episodic, meaning that under any policy, the environment almost surely reaches $s_{\text{reset}}$ (and so, $h_{\text{reset}}$ by definition of $\mathcal{M}_{\mathcal{H}}$), thus condition (i) of Definition D.6 is fulfilled. Second, since $H$ merely consists of a singleton, we can take $\rho \in \Delta(\mathcal{S}_{\mathcal{H}})$ so that, for any measurable $C \in \Sigma(\mathcal{S}_{\mathcal{H}})$, $\rho(C) = \mathbb{E}_{a \sim \bar{\pi}^{\clubsuit}(\cdot \mid h_{\text{reset}})} \mathbf{P}_{\mathcal{H}}(C \mid h_{\text{reset}}, a)$. Then, condition (ii) of Definition D.6 is trivially satisfied. Therefore, $\mathcal{M}_{\mathcal{H}}^{\bar{\pi}^{\clubsuit}}$ is a recurrent Harris chain, so we know there exists a stationary measure $\mathcal{H}_{\bar{\pi}^{\clubsuit}}$ over the histories of $\mathcal{P}$ that are likely to be seen under $\bar{\pi}^{\clubsuit}$ (Durrett, 2010, Theorem 5.8.9). Then, the $\epsilon$-*perturbation trick* of Huang 2020 can be used as is to enforce the aperiodicity in $\mathcal{M}_{\mathcal{H}}^{\bar{\pi}^{\clubsuit}}$. So, we have that $\xi_{\bar{\pi}^{\clubsuit}}^t(\cdot \mid \star)$ converges to $\mathcal{H}_{\bar{\pi}^{\clubsuit}}$ in total variation as $t$ goes to $\infty$ (Durrett, 2010, Theorem 5.8.12).

The resulting stationary distribution is thus the limiting distribution over the histories of $\mathcal{P}$ when the latter operates under $\bar{\pi}$, or equivalently, the limiting distribution of the MC $\mathcal{M}_{\mathcal{B}}^{\bar{\pi}^{\diamond}}$ by Corollary D.9. $\square$

## E Discrete Latent Variables and Temperatures

As mentioned in Section 2.3, the optimization process of the WAE-MDP relies on a temperature parameter, $\lambda \in (0, 1)$. The latter controls the continuity of the latent space learned. The purpose of the parameter is primarily to learn a discrete latent space model: precisely, we use continuous relaxation of discrete random variables (Maddison et al., 2017). This is essentially the Bernoulli version of the Gumbel softmax trick (Jang et al., 2017). The technique yields re-parameterizable and convex densities, which is compliant with stochastic gradient descent. The zero-temperature limit (i.e., passing from the continuous to the discrete setting) is theoretically enabled via simple rounding of continuous random variables. Furthermore, the logits of discrete densities are guaranteed to be identical to those of their relaxed counterparts.

Alternatively, one could use the *straight-through gradients estimator* (Bengio et al., 2013), as used by van den Oord et al. (2017); Fajtl et al. (2020) and Hafner et al. (2021). This consists in using discrete variables and non-differentiable functions in the forward pass of the input through the neural networks, while continuous variables and surrogate functions are used in the backward pass, i.e., during the backpropagation of the gradients. This yields low-variance, but biased gradients. In contrast, continuous relaxations allow to interpolate between these phenomena: a higher temperature produces low-variance but biased gradients, while lowering the temperature to zero increases the variance but ends up producing unbiased gradients.

Therefore, at any time, the discrete densities (zero-temperature limit) of the WAE-MDP are used, except when the gradients of the objective are computed, where their relaxed counterpart are used ($\lambda > 0$). As such, the WBU training procedure follows the same principle as WAE-MDPs: using continuous relaxation of the discrete random variables (precisely, multivariate Bernoullis) when the belief loss (Eq. 8) is minimized while the actual discrete (autoregressive) density is used otherwise. We chose temperature values by following the guidelines from the original paper (Maddison et al., 2017). In the latter, it is mentioned that setting up annealing schemes (to the zero temperature limit)

is a good practice but is not necessary for obtaining good results, which is confirmed experimentally in the experimental evaluation of WAE-MDPs (Delgrange et al., 2023, Appendix B.8).

## F  VALUE DIFFERENCE BOUNDS

This section is dedicated to proving Theorems 3.2 and 3.3. Both Theorems bound the value difference of histories, in the original and latent space models via our local and belief losses, to provide model and representation quality guarantees. Before proving the Theorems, we first formally define the *value function* of any POMDP, and then illustrate intuitively the meaning of each loss used to bound the value differences.

### F.1  VALUE FUNCTIONS

We start by formally defining the value function of any MDP.

**Definition F.1** (Value function). *Let $\mathcal{M} = \langle \mathcal{S}, \mathcal{A}, \mathbf{P}, \mathcal{R}, s_I, \gamma \rangle$ be an MDP, and $\pi$ be a policy for $\mathcal{M}$. Write $\mathcal{M}[s]$ for the MDP obtained by replacing $s_I$ by $s \in \mathcal{S}$. Then, the value of the state $s \in \mathcal{S}$ is defined as the expected return obtained from that state by running $\pi$, i.e., $V_\pi(s) = \mathbb{E}_\pi^{\mathcal{M}[s]} \left[ \sum_{t=0}^\infty \gamma^t \cdot \mathcal{R}(s_t, a_t) \right]$. Let $\mathcal{M}^\pi = \langle \mathcal{S}_\pi, \mathbf{P}_\pi, \mathcal{R}_\pi, s_I, \gamma \rangle$ be the Markov Chain induced by $\pi$ (cf. Definition D.4). Then, the value function can be defined as the unique solution of the Bellman's equation (Puterman, 1994): $V_\pi(s) = \mathcal{R}_\pi(s) + \mathbb{E}_{s' \sim \mathbf{P}_\pi(s)} \left[ \gamma \cdot V_\pi(s') \right]$. The typical goal of an RL agent is to learn a policy $\pi^\star$ that maximizes the value of the initial state of $\mathcal{M}$: $\max_{\pi^\star} V_{\pi^\star}(s_I)$.*

**Property F.2** (POMDP values). *We obtain the value function of any POMDP $\mathcal{P} = \langle \mathcal{M}, \Omega, \mathcal{O} \rangle$ by considering the values obtained in its belief MDP $\mathcal{M}_\mathcal{B} = \langle \mathcal{B}, \mathcal{A}, \mathbf{P}_\mathcal{B}, \mathcal{R}_\mathcal{B}, b_I, \gamma \rangle$. Therefore, the value of any history $h \in (\mathcal{A} \cdot \Omega)^*$ is obtained by mapping $h$ to the belief space: let $\pi$ be a policy conditioned on the beliefs of $\mathcal{P}$, then we write $V_\pi(h)$ for $V_\pi(\tau^*(h))$. Therefore, we have in particular for any latent policy $\bar{\pi} \colon \overline{\mathcal{B}} \to \Delta(\mathcal{A})$:*

$$V_{\bar{\pi}}(h) = \mathbb{E}_{\bar{\pi}^\diamond}^{\mathcal{M}_\mathcal{B}[\tau^*(h)]} \left[ \sum_{t=0}^\infty \gamma^t \cdot \mathcal{R}_\mathcal{B}(b_t, a_t) \right] \qquad \text{(cf. Lemma D.8 for definitions of } \bar{\pi}^\diamond \text{ and } \bar{\pi}^\clubsuit \text{)}$$

$$= \mathbb{E}_{\bar{\pi}^\clubsuit}^{\mathcal{M}_\mathcal{H}[h]} \left[ \sum_{t=0}^\infty \gamma^t \cdot \mathcal{R}_\mathcal{H}(h_t, a_t) \right] \qquad \text{(cf. Corollary D.9)}$$

$$= \mathbb{E}_{a \sim \bar{\pi}^\clubsuit(\cdot|h)} \left[ \mathcal{R}_\mathcal{H}(h, a) + \mathbb{E}_{h' \sim \mathbf{P}_\mathcal{H}(\cdot|h,a)} \left[ \gamma \cdot V_{\bar{\pi}}(h') \right] \right] \qquad \text{(by Definition F.1)}$$

$$= \mathbb{E}_{a \sim \bar{\pi}(\cdot|\varphi^*(h))} \left[ \mathcal{R}_\mathcal{H}(h, a) + \mathbb{E}_{h' \sim \mathbf{P}_\mathcal{H}(\cdot|h,a)} \left[ \gamma \cdot V_{\bar{\pi}}(h') \right] \right] \qquad \text{(by definition of } \bar{\pi}^\clubsuit \text{)}$$

$$= \mathbb{E}_{a \sim \bar{\pi}(\cdot|\varphi^*(h))} \mathbb{E}_{s \sim \tau^*(h)} \left[ \mathcal{R}(s, a) + \mathbb{E}_{s' \sim \mathbf{P}(\cdot|s,a)} \mathbb{E}_{o' \sim \mathcal{O}(\cdot|s',a)} \left[ \gamma \cdot V_{\bar{\pi}}(h \cdot a \cdot o') \right] \right]. \qquad \text{(by definition of } \mathcal{M}_\mathcal{H} \text{)}$$

*Similarly, we write $\overline{V}_{\bar{\pi}}$ for the values of a latent POMDP $\overline{\mathcal{P}}$.*

### F.2  LOCAL AND BELIEF LOSSES

Theorems 3.2 and 3.3 involve the minimization of *local* ($L_\mathcal{R}$, $L_\mathbf{P}$, $L_\mathcal{O}$) and *belief* ($L_{\bar{\tau}}$, $L_{\overline{\mathbf{P}}}^\varphi$, $L_{\overline{\mathcal{R}}}^\varphi$) losses. We intuitively describe how these losses are minimized via the latent flows depicted in Fig. 6.

The procedure allowing to minimize the local losses is depicted in Fig. 6a. At each step, a state $s$, an observation $o$ of $s$, and an action $a$ are drawn from the distribution of experiences encountered $\mathcal{H}_{\bar{\pi}}$ while executing $\bar{\pi}$. First, $\langle s, o \rangle$ is mapped to the latent space via the state embedding function of the WAE-MDP: $\phi(s, o) = \bar{s}$. Then, the action $a$ is executed both in the original and latent space models (respectively from $\langle s, o \rangle$ and $\bar{s}$), which allows to quantify the distance between the next reward and transition produced in the two models. Finally, the original model transitions to the next

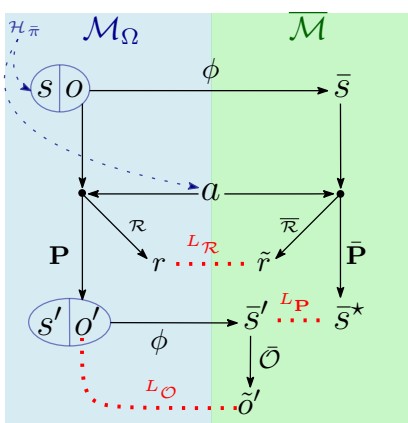

(a) Optimization of the latent space model parameters (i.e., $\overline{\mathcal{R}}$, $\overline{\mathbf{P}}$, and $\overline{\mathcal{O}}$) by minimizing local losses.

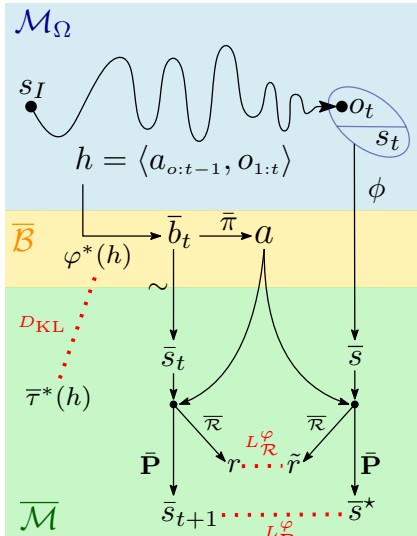

(b) Optimization of the belief encoder $\varphi$ by minimizing the (proxy) belief loss, as well as the reward and transition regularizers.

Figure 6: Latent flows used to compute the local and belief losses. Arrows represent (stochastic) mappings, the original state-observation (resp. latent state) space is spread along the blue (resp. green) area, and the latent belief space is spread along the yellow area. Distances (and discrepancies) are depicted in red. Notice that the blue area corresponds to the state-observation space $\mathcal{S}_\Omega$, which is accessible during training (Assumption 1).

state-observation pair $\langle s', o' \rangle$. Embedding this pair again to the latent space via $\phi(s', o') = \bar{s}'$ allows to quantify the distance between the original observation $o'$ and the one that is produced in the latent space model, from $\bar{s}'$ via $\tilde{o}' \sim \overline{\mathcal{O}}(\cdot \mid \bar{s}')$. Notice that in practice, $\overline{\mathcal{O}}$ is not necessarily Dirac (cf. Remark 3).

The procedure allowing to minimize the belief losses is depicted in Fig. 6b. This time, the optimization is performed *on-policy*, which means that it is performed while executing the policy in the original environment. At time step $t \geqslant 1$, the current history $h$ ends up in the observation $o_t$ of state $s_t$. First, the discrepancy between the latent belief obtained via our belief encoder $\bar{b}_t = \varphi^*(h)$ and the one obtained via the true latent belief update function $\bar{b}' = \bar{\tau}^*(h)$ is evaluated and can further be minimized from that point. Second, we compute the reward and transition regularizers by minimizing the distance between rewards and transitions produced from believed states $\bar{s}_t \sim \bar{b}_t$ (i.e., states witnessed by the current belief $\bar{b}_t$) and those produced by mapping the current state-observation pair into the latent space, via $\phi(s_t, o_t) = \bar{s}$, when the action $a \sim \bar{\pi}(\cdot \mid \bar{b}_t)$ is produced. Finally, $a$ is executed in the original environment and the process is repeated until the end of the episode.

## F.3 WARM UP: SOME WASSERSTEIN PROPERTIES

In the following, we elaborate on properties and definitions related to the Wasserstein metrics that will be useful to prove the main claims. In particular, Wasserstein can be reformulated as the maximum mean discrepancy of 1-*Lipschitz functions*. The claim holds when the temperature is at the zero-limit, i.e., when the discrete variables and densities of the models are used (cf. Appendix E). This further makes the distance metric $\bar{d}$ associated with the latent state space go to the discrete metric $\mathbf{1}_{\neq} \colon \mathcal{X} \to \{1, 0\}$ (Delgrange et al., 2023), formally defined as $\mathbf{1}_{\neq}(x1, x2) = 1$ iff $x_1 \neq x_2$.

**Definition F.3** (Lipschitz continuity). *Let $\mathcal{X}, \mathcal{Y}$ be two measurable set and $f \colon \mathcal{X} \to \mathcal{Y}$ be a function mapping elements from $\mathcal{X}$ to $\mathcal{Y}$. If otherwise specified, we consider that $f$ is real-valued function, i.e., $\mathcal{Y} = \mathbb{R}$. Assume that $\mathcal{X}$ is equipped with a metric $d \colon \mathcal{X} \to [0, \infty)$. Then, given a constant $K \geqslant 0$, we say that $f$ is $K$-Lipschitz iff, for any $x_1, x_2 \in \mathcal{X}$, $|f(x_1) - f(x_2)| \leqslant K \cdot d(x_1, x_2)$. We write $\mathcal{F}_d^K$ for the set of $K$-Lipschitz functions.*

**Definition F.4** (Wasserstein dual). *The Kantorovich-Rubinstein duality (Villani, 2009) allows formulating the Wasserstein distance between $P$ and $Q$ as $\mathcal{W}_d(P, Q) = \sup_{f \in \mathcal{F}_d^1} |\mathbb{E}_{x \sim P} f(x) - \mathbb{E}_{y \sim Q} f(y)|$.*

**Property F.5** (Lipschitz constant and total variation constants). *Let $f \colon \mathcal{X} \to \mathbb{R}$, so that $d$ is a metric on $\mathcal{X}$. Assume that $f$ is $K$-Lipschitz, i.e., $f \in \mathcal{F}_d^K$, then for any two distributions $P, Q \in \Delta(\mathcal{X})$, $|\mathbb{E}_{x_1 \sim P} f(x_1) - \mathbb{E}_{x_2 \sim Q} f(x_2)| \leqslant K \cdot \mathcal{W}_d(P, Q)$. In the case where $d = \mathbf{1}_{\neq}$, for any bounded function $g \colon \mathcal{X} \to Y$ with $Y \subseteq \mathbb{R}$, we have $|\mathbb{E}_{x_1 \sim P} g(x_1) - \mathbb{E}_{x_2 \sim Q} g(x_2)| \leqslant K_Y \cdot \mathcal{W}_{\mathbf{1}_{\neq}}(P, Q) = K_Y \cdot d_{TV}(P, Q)$, where $K_Y \geqslant \sup_{x \in \mathcal{X}} |g(x)|$ (Müller, 1997; Sriperumbudur et al., 2009).*

*In both cases, we call the smallest $K$ (resp. $K_Y$) the* integral probability metric (IPM) constant *of the function $f$ (resp. $g$).*

Property F.5 intuitively implies the emergence of the $K_{\overline{V}}$ constant in the Theorem's inequality: we know that the latent value function is bounded by $\sup_{\bar{s}, a} |\mathcal{R}(\bar{s}, a)| / 1 - \gamma$, so given two distributions $P, Q$ over $\overline{\mathcal{S}}$, the mean discrepancy of the latent value function is bounded by $K_{\overline{V}} \cdot \mathcal{W}_{\bar{d}}(P, Q)$ in the zero-temperature limit.

Finally, since the value difference is computed in expectation, we introduce the following useful property:

**Lemma F.6** (Wasserstein in expectation). *For any function $f \colon \mathcal{Y} \times \mathcal{X} \to \mathbb{R}$ so that $\mathcal{X}$ is equipped with the metric $d$, consider the function $g_y \colon \mathcal{X} \to \mathbb{R}$ defined as $g_y(x) = f(y, x)$. Assume that for any $y \in \mathcal{Y}$, $g_y$ is Lipschitz (or simply bounded if $d = \mathbf{1}_{\neq}$) and let $K$ be the IPM constant of $g_y$. Then, let $\mathcal{D} \in \Delta(\mathcal{Y})$ be a distribution over $\mathcal{Y}$ and $P, Q \in \Delta(\mathcal{X})$ be two distributions over $\mathcal{X}$, we have $\mathbb{E}_{y \sim \mathcal{D}} |\mathbb{E}_{x_1 \sim P} f(y, x_1) - \mathbb{E}_{x_2 \sim Q} f(y, x_2)| \leqslant K \cdot \mathcal{W}_d(P, Q)$.*

*Proof.* The proof is straightforward by construction of $g_y$:

$$
\begin{aligned}
& \mathbb{E}_{y \sim \mathcal{D}} \left| \mathbb{E}_{x_1 \sim P} f(y, x_1) - \mathbb{E}_{x_2 \sim Q} f(y, x_2) \right| \\
&= \mathbb{E}_{y \sim \mathcal{D}} \left| \mathbb{E}_{x_1 \sim P} g_y(x_1) - \mathbb{E}_{x_2 \sim Q} g_y(x_2) \right| \\
&\leqslant \mathbb{E}_{y \sim \mathcal{D}} \left[ K \cdot \mathcal{W}_d(P, Q) \right] \qquad \text{(by Property F.5)} \\
&= K \cdot \mathcal{W}_d(P, Q)
\end{aligned}
$$

$\square$

### F.4 MODEL QUALITY BOUND: TIME TO RAISE YOUR EXPECTATIONS

Let us restate Theorem 3.2:

**Theorem 3.2** (Model quality). *For any latent policy $\bar{\pi} \colon \overline{\mathcal{B}} \to \Delta(\mathcal{A})$, the values of $\mathcal{P}$ and $\overline{\mathcal{P}}$ are bounded by the local and belief losses in average when, in both models, the actions are produced by $\bar{\pi}$, which is conditioned on the latent belief induced by $\varphi$, i.e., $a \sim \bar{\pi}(\cdot \mid \varphi^*(h))$:*

$$
\mathbb{E}_{h \sim \mathcal{H}_{\bar{\pi}}} \left| V_{\bar{\pi}}(h) - \overline{V}_{\bar{\pi}}(h) \right| \leqslant \frac{\mathcal{L}}{1 - \gamma}. \tag{6}
$$

*Proof.* The plan of the proof is as follows:

1. We exploit the fact that the value function can be defined as the fixed point of the Bellman's equations;

2. We repeatedly apply the triangular and the Jensen's inequalities to end up with inequalities which reveal mean discrepancies for either rewards or value functions;

3. We exploit the fact that the WAE-MDP is in the zero-temperature limit to bound thos discrepancies by Wasserstein (see Porperty F.5 and the related discussion);

4. The last two points allow highlighting the $L_1$ norm and Wasserstein terms in the local and belief losses;

5. Finally, we set up the inequalities to obtain a discounted next value difference term, and we exploit the stationary property of $\mathcal{H}_{\bar{\pi}}$ to fall back on the original, discounted, absolute value difference term;

6. Putting all together, we end up with an inequality only composed of constants, multiplied by losses that we aim at minimizing.

Concretely, the absolute value difference can be bounded by:

$$
\mathop{\mathbb{E}}_{h\sim\mathcal{H}_{\bar{\pi}}} \left| V_{\bar{\pi}}(h) - \overline{V}_{\bar{\pi}}(h) \right|
$$

$$
= \mathop{\mathbb{E}}_{h\sim\mathcal{H}_{\bar{\pi}}} \left| \mathop{\mathbb{E}}_{s\sim\tau^*(h)} \mathop{\mathbb{E}}_{a\sim\bar{\pi}(\cdot|\varphi^*(h))} \left[ \mathcal{R}(s,a) + \gamma \mathop{\mathbb{E}}_{s'\sim\mathbf{P}(\cdot|s,a)} \mathop{\mathbb{E}}_{o'\sim\mathcal{O}(\cdot|s',a)} V_{\bar{\pi}}(h\cdot a\cdot o') \right] \right.
$$

$$
\left. - \mathop{\mathbb{E}}_{\bar{s}\sim\bar{\tau}^*(h)} \mathop{\mathbb{E}}_{a\sim\bar{\pi}(\cdot|\varphi^*(h))} \left[ \overline{\mathcal{R}}(\bar{s},a) + \gamma \mathop{\mathbb{E}}_{\bar{s}'\sim\overline{\mathbf{P}}(\cdot|\bar{s},a)} \mathop{\mathbb{E}}_{o'\sim\overline{\mathcal{O}}(\cdot|\bar{s}')} \overline{V}_{\bar{\pi}}(h\cdot a\cdot o') \right] \right|
$$

$$
\text{(see Property F.2)}
$$

$$
= \mathop{\mathbb{E}}_{h\sim\mathcal{H}_{\bar{\pi}}} \mathop{\mathbb{E}}_{a\sim\bar{\pi}(\cdot|\varphi^*(h))} \left| \mathop{\mathbb{E}}_{s\sim\tau^*(h)} \left[ \mathcal{R}(s,a) + \gamma \mathop{\mathbb{E}}_{s'\sim\mathbf{P}(\cdot|s,a)} \mathop{\mathbb{E}}_{o'\sim\mathcal{O}(\cdot|s',a)} V_{\bar{\pi}}(h\cdot a\cdot o') \right] \right.
$$

$$
\left. - \mathop{\mathbb{E}}_{\bar{s}\sim\bar{\tau}^*(h)} \left[ \overline{\mathcal{R}}(\bar{s},a) + \gamma \mathop{\mathbb{E}}_{\bar{s}'\sim\overline{\mathbf{P}}(\cdot|\bar{s},a)} \mathop{\mathbb{E}}_{o'\sim\overline{\mathcal{O}}(\cdot|\bar{s}')} \overline{V}_{\bar{\pi}}(h\cdot a\cdot o') \right] \right|
$$

$$
(\textcolor{blue}{\text{ Jensen's inequality}})
$$

$$
\leqslant \mathop{\mathbb{E}}_{h\sim\mathcal{H}_{\bar{\pi}}} \mathop{\mathbb{E}}_{a\sim\bar{\pi}(\cdot|\varphi^*(h))} \left[ \left| \mathop{\mathbb{E}}_{s\sim\tau^*(h)} \mathcal{R}(s,a) - \mathop{\mathbb{E}}_{\bar{s}\sim\bar{\tau}^*(h)} \overline{\mathcal{R}}(\bar{s},a) \right| \right.
$$

$$
\left. + \gamma \left| \mathop{\mathbb{E}}_{s\sim\tau^*(h)} \mathop{\mathbb{E}}_{s'\sim\mathbf{P}(\cdot|s,a)} \mathop{\mathbb{E}}_{o'\sim\mathcal{O}(\cdot|s',a)} V_{\pi}(h\cdot a\cdot o') - \mathop{\mathbb{E}}_{\bar{s}\sim\bar{\tau}^*(h)} \mathop{\mathbb{E}}_{\bar{s}'\sim\overline{\mathbf{P}}(\cdot|\bar{s},a)} \mathop{\mathbb{E}}_{o'\sim\overline{\mathcal{O}}(\cdot|\bar{s}')} \overline{V}_{\bar{\pi}}(h\cdot a\cdot o') \right| \right]
$$

$$
(\textcolor{blue}{\text{ Triangular inequality}})
$$

For the sake of clarity, we split the inequality into two parts.

**Part 1: Reward bounds**

$$
\mathop{\mathbb{E}}_{h\sim\mathcal{H}_{\bar{\pi}}} \mathop{\mathbb{E}}_{a\sim\bar{\pi}(\cdot|\varphi^*(h))} \left| \mathop{\mathbb{E}}_{s\sim\tau^*(h)} \mathcal{R}(s,a) - \mathop{\mathbb{E}}_{\bar{s}\sim\bar{\tau}^*(h)} \overline{\mathcal{R}}(\bar{s},a) \right|
$$

$$
= \mathop{\mathbb{E}}_{h,o\sim\mathcal{H}_{\bar{\pi}}} \mathop{\mathbb{E}}_{a\sim\bar{\pi}(\cdot|\varphi^*(h))} \left| \mathop{\mathbb{E}}_{s\sim\tau^*(h)} \left[ \mathcal{R}(s,a) - \overline{\mathcal{R}}(\phi(s,o),a) \right] + \mathop{\mathbb{E}}_{s\sim\tau^*(h)} \mathop{\mathbb{E}}_{\bar{s}\sim\bar{\tau}^*(h)} \left[ \overline{\mathcal{R}}(\phi(s,o),a) - \overline{\mathcal{R}}(\bar{s},a) \right] \right|
$$

($o$ is the last observation of $h$;

the state embedding function $\phi$ that links the original and latent state spaces comes into play)

$$
\leqslant \mathop{\mathbb{E}}_{h,o\sim\mathcal{H}_{\bar{\pi}}} \mathop{\mathbb{E}}_{a\sim\bar{\pi}(\cdot|\varphi^*(h))} \left[ \left| \mathop{\mathbb{E}}_{s\sim\tau^*(h)} \left[ \mathcal{R}(s,a) - \overline{\mathcal{R}}(\phi(s,o),a) \right] \right| + \left| \mathop{\mathbb{E}}_{s\sim\tau^*(h)} \mathop{\mathbb{E}}_{\bar{s}\sim\bar{\tau}^*(h)} \left[ \overline{\mathcal{R}}(\phi(s,o),a) - \overline{\mathcal{R}}(\bar{s},a) \right] \right| \right]
$$

$$
(\textcolor{blue}{\text{ Triangular inequality}})
$$

$$
\leqslant \mathop{\mathbb{E}}_{h,o\sim\mathcal{H}_{\bar{\pi}}} \mathop{\mathbb{E}}_{a\sim\bar{\pi}(\cdot|\varphi^*(h))} \left[ \mathop{\mathbb{E}}_{s\sim\tau^*(h)} \left| \mathcal{R}(s,a) - \overline{\mathcal{R}}(\phi(s,o),a) \right| + \left| \mathop{\mathbb{E}}_{s\sim\tau^*(h)} \mathop{\mathbb{E}}_{\bar{s}\sim\bar{\tau}^*(h)} \left[ \overline{\mathcal{R}}(\phi(s,o),a) - \overline{\mathcal{R}}(\bar{s},a) \right] \right| \right]
$$

$$
(\textcolor{blue}{\text{ Jensen's inequality}})
$$

$$
= \mathop{\mathbb{E}}_{h,o\sim\mathcal{H}_{\bar{\pi}}} \mathop{\mathbb{E}}_{a\sim\bar{\pi}(\cdot|\varphi^*(h))} \mathop{\mathbb{E}}_{s\sim\tau^*(h)} \left| \mathcal{R}(s,a) - \overline{\mathcal{R}}(\phi(s,o),a) \right|
$$

$$
+ \mathop{\mathbb{E}}_{h,o\sim\mathcal{H}_{\bar{\pi}}} \mathop{\mathbb{E}}_{a\sim\bar{\pi}(\cdot|\varphi^*(h))} \left| \mathop{\mathbb{E}}_{s\sim\tau^*(h)} \mathop{\mathbb{E}}_{\bar{s}\sim\bar{\tau}^*(h)} \left[ \overline{\mathcal{R}}(\phi(s,o),a) - \overline{\mathcal{R}}(\bar{s},a) \right] \right|
$$

$$=L_{\mathcal{R}} + \mathop{\mathbb{E}}_{h,o\sim\mathcal{H}_{\bar{\pi}}} \mathop{\mathbb{E}}_{a\sim\pi(\cdot|\varphi*(h))} \left| \mathop{\mathbb{E}}_{s\sim\tau*(h)} \mathop{\mathbb{E}}_{\bar{s}\sim\bar{\tau}*(h)} \left[\overline{\mathcal{R}}(\phi(s,o),a) - \overline{\mathcal{R}}(\bar{s},a)\right] \right| \quad (\text{ by definition of } L_{\mathcal{R}})$$

$$=L_{\mathcal{R}} + \mathop{\mathbb{E}}_{h,o\sim\mathcal{H}_{\bar{\pi}}} \mathop{\mathbb{E}}_{a\sim\pi(\cdot|\varphi*(h))} \left| \mathop{\mathbb{E}}_{s\sim\tau*(h)} \mathop{\mathbb{E}}_{\bar{s}\sim\bar{\tau}*(h)} \mathop{\mathbb{E}}_{\bar{s}_\perp\sim\varphi*(h)} \left[\left[\overline{\mathcal{R}}(\phi(s,o),a) - \overline{\mathcal{R}}(\bar{s}_\perp,a)\right] + \left[\overline{\mathcal{R}}(\bar{s}_\perp,a) - \overline{\mathcal{R}}(\bar{s},a)\right]\right] \right|$$
$$(\text{ the belief encoder } \varphi \text{ comes into play})$$

$$=L_{\mathcal{R}} + \mathop{\mathbb{E}}_{h,o\sim\mathcal{H}_{\bar{\pi}}} \mathop{\mathbb{E}}_{a\sim\pi(\cdot|\varphi*(h))} \left| \mathop{\mathbb{E}}_{s\sim\tau*(h)} \mathop{\mathbb{E}}_{\bar{s}\sim\varphi*(h)} \left[\overline{\mathcal{R}}(\phi(s,o),a) - \overline{\mathcal{R}}(\bar{s},a)\right] \right.$$
$$\left. + \mathop{\mathbb{E}}_{\bar{s}\sim\bar{\tau}*(h)} \mathop{\mathbb{E}}_{\bar{s}_\perp\sim\varphi*(h)} \left[\overline{\mathcal{R}}(\bar{s}_\perp,a) - \overline{\mathcal{R}}(\bar{s},a)\right] \right|$$

$$\leqslant L_{\mathcal{R}} + \mathop{\mathbb{E}}_{h,o\sim\mathcal{H}_{\bar{\pi}}} \mathop{\mathbb{E}}_{a\sim\pi(\cdot|\varphi*(h))} \left[ \left| \mathop{\mathbb{E}}_{s\sim\tau*(h)} \mathop{\mathbb{E}}_{\bar{s}\sim\varphi*(h)} \left[\overline{\mathcal{R}}(\phi(s,o),a) - \overline{\mathcal{R}}(\bar{s},a)\right] \right| \right.$$
$$\left. + \left| \mathop{\mathbb{E}}_{\bar{s}\sim\bar{\tau}*(h)} \mathop{\mathbb{E}}_{\bar{s}_\perp\sim\varphi*(h)} \left[\overline{\mathcal{R}}(\bar{s}_\perp,a) - \overline{\mathcal{R}}(\bar{s},a)\right] \right| \right]$$
$$(\text{ Triangular inequality})$$

$$\leqslant L_{\mathcal{R}} + \mathop{\mathbb{E}}_{h,o\sim\mathcal{H}_{\bar{\pi}}} \mathop{\mathbb{E}}_{a\sim\pi(\cdot|\varphi*(h))} \left[ \mathop{\mathbb{E}}_{s\sim\tau*(h)} \mathop{\mathbb{E}}_{\bar{s}\sim\varphi*(h)} \left|\overline{\mathcal{R}}(\phi(s,o),a) - \overline{\mathcal{R}}(\bar{s},a)\right| \right.$$
$$\left. + \left| \mathop{\mathbb{E}}_{\bar{s}\sim\bar{\tau}*(h)} \mathop{\mathbb{E}}_{\bar{s}_\perp\sim\varphi*(h)} \left[\overline{\mathcal{R}}(\bar{s}_\perp,a) - \overline{\mathcal{R}}(\bar{s},a)\right] \right| \right]$$
$$(\text{ Jensen's inequality})$$

$$=L_{\mathcal{R}} + \mathop{\mathbb{E}}_{h,o\sim\mathcal{H}_{\bar{\pi}}} \mathop{\mathbb{E}}_{a\sim\pi(\cdot|\varphi*(h))} \mathop{\mathbb{E}}_{s\sim\tau*(h)} \mathop{\mathbb{E}}_{\bar{s}\sim\varphi*(h)} \left|\overline{\mathcal{R}}(\phi(s,o),a) - \overline{\mathcal{R}}(\bar{s},a)\right|$$
$$+ \mathop{\mathbb{E}}_{h,o\sim\mathcal{H}_{\bar{\pi}}} \mathop{\mathbb{E}}_{a\sim\pi(\cdot|\varphi*(h))} \left| \mathop{\mathbb{E}}_{\bar{s}\sim\bar{\tau}*(h)} \mathop{\mathbb{E}}_{\bar{s}_\perp\sim\varphi*(h)} \overline{\mathcal{R}}(\bar{s}_\perp,a) - \overline{\mathcal{R}}(\bar{s},a) \right|$$

$$=L_{\mathcal{R}} + L_{\overline{\mathcal{R}}}^{\varphi} + \mathop{\mathbb{E}}_{h,o\sim\mathcal{H}_{\bar{\pi}}} \mathop{\mathbb{E}}_{a\sim\pi(\cdot|\varphi*(h))} \left| \mathop{\mathbb{E}}_{\bar{s}\sim\bar{\tau}*(h)} \mathop{\mathbb{E}}_{\bar{s}_\perp\sim\varphi*(h)} \left[\overline{\mathcal{R}}(\bar{s}_\perp,a) - \overline{\mathcal{R}}(\bar{s},a)\right] \right|$$
$$(\text{ by definition of } L_{\overline{\mathcal{R}}}^{\varphi}, \text{ Eq. 5})$$

$$\leqslant L_{\mathcal{R}} + L_{\overline{\mathcal{R}}}^{\varphi} + \mathop{\mathbb{E}}_{h\sim\mathcal{H}_{\bar{\pi}}} \overline{\mathcal{R}}^\star \mathcal{W}_{\bar{d}}\left(\bar{\tau}*(h), \varphi*(h)\right) \qquad (\text{ as } \lambda \to 0, \text{ by Lem. F.6 and Prop. F.5})$$
$$=L_{\mathcal{R}} + L_{\overline{\mathcal{R}}}^{\varphi} + \overline{\mathcal{R}}^\star L_{\bar{\tau}};$$

**Part 2: Next value bounds**

$$\gamma \cdot \mathop{\mathbb{E}}_{h\sim\mathcal{H}_{\bar{\pi}}} \mathop{\mathbb{E}}_{a\sim\pi(\cdot|\varphi*(h))} \left| \mathop{\mathbb{E}}_{s\sim\tau*(h)} \mathop{\mathbb{E}}_{s'\sim\mathbf{P}(\cdot|s,a)} \mathop{\mathbb{E}}_{o'\sim\mathcal{O}(\cdot|s',a)} V_{\bar{\pi}}\left(h\cdot a\cdot o'\right) \right.$$
$$\left. - \mathop{\mathbb{E}}_{\bar{s}\sim\bar{\tau}*(h)} \mathop{\mathbb{E}}_{\bar{s}'\sim\overline{\mathbf{P}}(\cdot|\bar{s},a)} \mathop{\mathbb{E}}_{o'\sim\overline{\mathcal{O}}(\cdot|\bar{s}')} \overline{V}_{\bar{\pi}}\left(h\cdot a\cdot o'\right) \right|$$

$$= \gamma \cdot \mathop{\mathbb{E}}_{h\sim\mathcal{H}_{\bar{\pi}}} \mathop{\mathbb{E}}_{a\sim\pi(\cdot|\varphi*(h))} \left| \mathop{\mathbb{E}}_{s\sim\tau*(h)} \mathop{\mathbb{E}}_{s'\sim\mathbf{P}(\cdot|s,a)} \mathop{\mathbb{E}}_{o'\sim\mathcal{O}(\cdot|s',a)} \left[ V_{\bar{\pi}}\left(h\cdot a\cdot o'\right) - \mathop{\mathbb{E}}_{\hat{o}'\sim\overline{\mathcal{O}}(\cdot|\phi(s',o'))} \overline{V}_{\bar{\pi}}\left(h\cdot a\cdot \hat{o}'\right) \right] \right.$$
$$+ \left[ \mathop{\mathbb{E}}_{s\sim\tau*(h)} \mathop{\mathbb{E}}_{s'\sim\mathbf{P}(\cdot|s,a)} \mathop{\mathbb{E}}_{o'\sim\mathcal{O}(\cdot|s',a)} \mathop{\mathbb{E}}_{\hat{o}'\sim\overline{\mathcal{O}}(\cdot|\phi(s',o'))} \overline{V}_{\bar{\pi}}\left(h\cdot a\cdot \hat{o}'\right) \right.$$
$$\left. \left. - \mathop{\mathbb{E}}_{\bar{s}\sim\bar{\tau}*(h)} \mathop{\mathbb{E}}_{\bar{s}'\sim\overline{\mathbf{P}}(\cdot|\bar{s},a)} \mathop{\mathbb{E}}_{o'\sim\overline{\mathcal{O}}(\cdot|\bar{s}')} \overline{V}_{\bar{\pi}}\left(h\cdot a\cdot o'\right) \right] \right|$$
$$(\text{the state embedding function } \phi \text{ comes into play, as well as the latent observation function } \overline{\mathcal{O}})$$

$$\leqslant \quad \gamma \cdot \mathop{\mathbb{E}}_{h\sim\mathcal{H}_{\bar\pi}} \mathop{\mathbb{E}}_{a\sim\pi(\cdot|\varphi^*(h))} \left| \mathop{\mathbb{E}}_{s\sim\tau^*(h)} \mathop{\mathbb{E}}_{s'\sim\mathbf{P}(\cdot|s,a)} \mathop{\mathbb{E}}_{o'\sim\mathcal{O}(\cdot|s',a)} \left[ V_{\bar\pi}(h\cdot a\cdot o') - \mathop{\mathbb{E}}_{\hat o'\sim\overline{\mathcal{O}}(\cdot|\phi(s',o'))} \overline{V}_{\bar\pi}(h\cdot a\cdot \hat o') \right] \right|$$

$$+ \gamma \cdot \mathop{\mathbb{E}}_{h\sim\mathcal{H}_{\bar\pi}} \mathop{\mathbb{E}}_{a\sim\bar\pi(\cdot|\varphi^*(h))} \left| \mathop{\mathbb{E}}_{s\sim\tau^*(h)} \mathop{\mathbb{E}}_{s'\sim\mathbf{P}(\cdot|s,a)} \mathop{\mathbb{E}}_{o'\sim\mathcal{O}(\cdot|s',a)} \mathop{\mathbb{E}}_{\hat o'\sim\overline{\mathcal{O}}(\cdot|\phi(s',o'))} \overline{V}_{\bar\pi}(h\cdot a\cdot\hat o') \right.$$

$$\left. - \mathop{\mathbb{E}}_{\bar s\sim\bar\tau^*(h)} \mathop{\mathbb{E}}_{\bar s'\sim\overline{\mathbf{P}}(\cdot|\bar s,a)} \mathop{\mathbb{E}}_{o'\sim\overline{\mathcal{O}}(\cdot|\bar s')} \overline{V}_{\bar\pi}(h\cdot a\cdot o') \right|$$

(Triangular inequality)

$$= \quad \gamma \cdot \mathop{\mathbb{E}}_{h\sim\mathcal{H}_{\bar\pi}} \mathop{\mathbb{E}}_{a\sim\pi(\cdot|\varphi^*(h))} \left| \mathop{\mathbb{E}}_{s\sim\tau^*(h)} \mathop{\mathbb{E}}_{s'\sim\mathbf{P}(\cdot|s,a)} \mathop{\mathbb{E}}_{o'\sim\mathcal{O}(\cdot|s',a)} \left[ V_{\bar\pi}(h\cdot a\cdot o') - \mathop{\mathbb{E}}_{\hat o'\sim\overline{\mathcal{O}}(\cdot|\phi(s',o'))} \overline{V}_{\bar\pi}(h\cdot a\cdot \hat o') \right] \right|$$

$$+ \gamma \cdot \mathop{\mathbb{E}}_{h,o\sim\mathcal{H}_{\bar\pi}} \mathop{\mathbb{E}}_{a\sim\bar\pi(\cdot|\varphi^*(h))} \left| \mathop{\mathbb{E}}_{s\sim\tau^*(h)} \left[ \mathop{\mathbb{E}}_{s',o'\sim\mathbf{P}_\Omega(\cdot|s,o,a)} \mathop{\mathbb{E}}_{\hat o'\sim\overline{\mathcal{O}}(\cdot|\phi(s',o'))} \overline{V}_{\bar\pi}(h\cdot a\cdot\hat o') \right.\right.$$

$$\left. - \mathop{\mathbb{E}}_{\bar s'\sim\overline{\mathbf{P}}(\cdot|\phi(s,o),a)} \mathop{\mathbb{E}}_{\hat o'\sim\overline{\mathcal{O}}(\cdot|\bar s')} \overline{V}_{\bar\pi}(h\cdot a\cdot\hat o') \right]$$

$$+ \left[ \mathop{\mathbb{E}}_{s\sim\tau^*(h)} \mathop{\mathbb{E}}_{\bar s'\sim\overline{\mathbf{P}}(\cdot|\phi(s,o),a)} \mathop{\mathbb{E}}_{o'\sim\overline{\mathcal{O}}(\cdot|\bar s')} \overline{V}_{\bar\pi}(h\cdot a\cdot o') \right.$$

$$\left.\left. - \mathop{\mathbb{E}}_{\bar s\sim\bar\tau^*(h)} \mathop{\mathbb{E}}_{\bar s'\sim\overline{\mathbf{P}}(\cdot|\bar s,a)} \mathop{\mathbb{E}}_{o'\sim\overline{\mathcal{O}}(\cdot|\bar s')} \overline{V}_{\bar\pi}(h\cdot a\cdot o') \right] \right|$$

(*o* is the last observation of *h*; the latent MDP dynamics, modeled by $\overline{\mathbf{P}}$, come into play)

$$\leqslant \quad \gamma \cdot \mathop{\mathbb{E}}_{h\sim\mathcal{H}_{\bar\pi}} \mathop{\mathbb{E}}_{a\sim\pi(\cdot|\varphi^*(h))} \left| \mathop{\mathbb{E}}_{s\sim\tau^*(h)} \mathop{\mathbb{E}}_{s'\sim\mathbf{P}(\cdot|s,a)} \mathop{\mathbb{E}}_{o'\sim\mathcal{O}(\cdot|s',a)} \left[ V_{\bar\pi}(h\cdot a\cdot o') - \mathop{\mathbb{E}}_{\hat o'\sim\overline{\mathcal{O}}(\cdot|\phi(s',o'))} \overline{V}_{\bar\pi}(h\cdot a\cdot \hat o') \right] \right|$$

$$+ \gamma \cdot \mathop{\mathbb{E}}_{h,o\sim\mathcal{H}_{\bar\pi}} \mathop{\mathbb{E}}_{a\sim\bar\pi(\cdot|\varphi^*(h))} \left| \mathop{\mathbb{E}}_{s\sim\tau^*(h)} \left[ \mathop{\mathbb{E}}_{s',o'\sim\mathbf{P}_\Omega(\cdot|s,o,a)} \mathop{\mathbb{E}}_{\hat o'\sim\overline{\mathcal{O}}(\cdot|\phi(s',o'))} \overline{V}_{\bar\pi}(h\cdot a\cdot\hat o') \right.\right.$$

$$\left.\left. - \mathop{\mathbb{E}}_{\bar s'\sim\overline{\mathbf{P}}(\cdot|\phi(s,o),a)} \mathop{\mathbb{E}}_{\hat o'\sim\overline{\mathcal{O}}(\cdot|\bar s')} \overline{V}_{\bar\pi}(h\cdot a\cdot\hat o') \right] \right|$$

$$+ \gamma \cdot \mathop{\mathbb{E}}_{h,o\sim\mathcal{H}_{\bar\pi}} \mathop{\mathbb{E}}_{a\sim\bar\pi(\cdot|\varphi^*(h))} \left| \mathop{\mathbb{E}}_{s\sim\tau^*(h)} \mathop{\mathbb{E}}_{\bar s'\sim\overline{\mathbf{P}}(\cdot|\phi(s,o),a)} \mathop{\mathbb{E}}_{o'\sim\overline{\mathcal{O}}(\cdot|\bar s')} \overline{V}_{\bar\pi}(h\cdot a\cdot o') \right.$$

$$\left. - \mathop{\mathbb{E}}_{\bar s\sim\bar\tau^*(h)} \mathop{\mathbb{E}}_{\bar s'\sim\overline{\mathbf{P}}(\cdot|\bar s,a)} \mathop{\mathbb{E}}_{o'\sim\overline{\mathcal{O}}(\cdot|\bar s')} \overline{V}_{\bar\pi}(h\cdot a\cdot o') \right|$$

(Triangular inequality)

$$\leqslant \quad \gamma \cdot \mathop{\mathbb{E}}_{h\sim\mathcal{H}_{\bar\pi}} \mathop{\mathbb{E}}_{a\sim\pi(\cdot|\varphi^*(h))} \left| \mathop{\mathbb{E}}_{s\sim\tau^*(h)} \mathop{\mathbb{E}}_{s'\sim\mathbf{P}(\cdot|s,a)} \mathop{\mathbb{E}}_{o'\sim\mathcal{O}(\cdot|s',a)} \left[ V_{\bar\pi}(h\cdot a\cdot o') - \mathop{\mathbb{E}}_{\hat o'\sim\overline{\mathcal{O}}(\cdot|\phi(s',o'))} \overline{V}_{\bar\pi}(h\cdot a\cdot \hat o') \right] \right|$$

$$+ \gamma \cdot \mathop{\mathbb{E}}_{h,o\sim\mathcal{H}_{\bar\pi}} \mathop{\mathbb{E}}_{a\sim\bar\pi(\cdot|\varphi^*(h))} \mathop{\mathbb{E}}_{s\sim\tau^*(h)} \left| \mathop{\mathbb{E}}_{\bar s'\sim\phi\mathbf{P}_\Omega(\cdot|s,o,a)} \mathop{\mathbb{E}}_{o'\sim\overline{\mathcal{O}}(\cdot|\bar s')} \overline{V}_{\bar\pi}(h\cdot a\cdot o') \right.$$

$$\left. - \mathop{\mathbb{E}}_{\bar s'\sim\overline{\mathbf{P}}(\cdot|\phi(s,o),a)} \mathop{\mathbb{E}}_{o'\sim\overline{\mathcal{O}}(\cdot|\bar s')} \overline{V}_{\bar\pi}(h\cdot a\cdot o') \right|$$

$$+ \gamma \cdot \mathop{\mathbb{E}}_{h,o\sim\mathcal{H}_{\bar\pi}} \mathop{\mathbb{E}}_{a\sim\bar\pi(\cdot|\varphi^*(h))} \left| \mathop{\mathbb{E}}_{s\sim\tau^*(h)} \mathop{\mathbb{E}}_{\bar s'\sim\overline{\mathbf{P}}(\cdot|\phi(s,o),a)} \mathop{\mathbb{E}}_{o'\sim\overline{\mathcal{O}}(\cdot|\bar s')} \overline{V}_{\bar\pi}(h\cdot a\cdot o') \right.$$

$$- \mathop{\mathbb{E}}_{\bar{s} \sim \bar{\tau}^*(h)} \mathop{\mathbb{E}}_{\bar{s}' \sim \overline{\mathbf{P}}(\cdot | \bar{s}, a)} \mathop{\mathbb{E}}_{o' \sim \overline{\mathcal{O}}(\cdot | \bar{s}')} \overline{V}_{\bar{\pi}}\big(h \cdot a \cdot o'\big) \Bigg|$$

$$\text{(by definition of } \phi \mathbf{P}_\Omega \text{ and the Jensen's inequality)}$$

$$\leqslant \gamma \cdot \mathop{\mathbb{E}}_{h \sim \mathcal{H}_{\bar{\pi}} \ a \sim \bar{\pi}(\cdot | \varphi^*(h))} \left| \mathop{\mathbb{E}}_{s \sim \tau^*(h)} \mathop{\mathbb{E}}_{s' \sim \mathbf{P}(\cdot | s, a)} \mathop{\mathbb{E}}_{o' \sim \mathcal{O}(\cdot | s', a)} \left[ V_{\bar{\pi}}\big(h \cdot a \cdot o'\big) - \mathop{\mathbb{E}}_{\hat{o}' \sim \overline{\mathcal{O}}(\cdot | \phi(s', o'))} \overline{V}_{\bar{\pi}}\big(h \cdot a \cdot \hat{o}'\big) \right] \right|$$

$$+ \gamma \cdot \mathop{\mathbb{E}}_{h, o \sim \mathcal{H}_{\bar{\pi}} \ a \sim \bar{\pi}(\cdot | \varphi^*(h)) \ s \sim \tau^*(h)} K_{\overline{V}} \cdot \mathcal{W}_{\bar{d}}\big( \phi \mathbf{P}_\Omega(\cdot \mid s, a), \overline{\mathbf{P}}(\cdot \mid \phi(s, o), a) \big)$$

$$+ \gamma \cdot \mathop{\mathbb{E}}_{h, o \sim \mathcal{H}_{\bar{\pi}} \ a \sim \bar{\pi}(\cdot | \varphi^*(h))} \left| \mathop{\mathbb{E}}_{s \sim \tau^*(h)} \mathop{\mathbb{E}}_{\bar{s}' \sim \overline{\mathbf{P}}(\cdot | \phi(s, o), a)} \mathop{\mathbb{E}}_{o' \sim \overline{\mathcal{O}}(\cdot | \bar{s}')} \overline{V}_{\bar{\pi}}\big(h \cdot a \cdot o'\big) \right.$$

$$\left. - \mathop{\mathbb{E}}_{\bar{s} \sim \bar{\tau}^*(h)} \mathop{\mathbb{E}}_{\bar{s}' \sim \overline{\mathbf{P}}(\cdot | \bar{s}, a)} \mathop{\mathbb{E}}_{o' \sim \overline{\mathcal{O}}(\cdot | \bar{s}')} \overline{V}_{\bar{\pi}}\big(h \cdot a \cdot o'\big) \right|$$

$$\text{(as } \lambda \to 0 \text{, by Lem. F.6)}$$

$$\leqslant \gamma \cdot \mathop{\mathbb{E}}_{h \sim \mathcal{H}_{\bar{\pi}} \ a \sim \bar{\pi}(\cdot | \varphi^*(h))} \left| \mathop{\mathbb{E}}_{s \sim \tau^*(h)} \mathop{\mathbb{E}}_{s' \sim \mathbf{P}(\cdot | s, a)} \mathop{\mathbb{E}}_{o' \sim \mathcal{O}(\cdot | s', a)} \left[ V_{\bar{\pi}}\big(h \cdot a \cdot o'\big) - \mathop{\mathbb{E}}_{\hat{o}' \sim \overline{\mathcal{O}}(\cdot | \phi(s', o'))} \overline{V}_{\bar{\pi}}\big(h \cdot a \cdot \hat{o}'\big) \right] \right|$$

$$+ \gamma K_{\overline{V}} L_{\mathbf{P}}$$

$$+ \gamma \cdot \mathop{\mathbb{E}}_{h, o \sim \mathcal{H}_{\bar{\pi}} \ a \sim \bar{\pi}(\cdot | \varphi^*(h))} \left| \mathop{\mathbb{E}}_{s \sim \tau^*(h)} \mathop{\mathbb{E}}_{\bar{s}' \sim \overline{\mathbf{P}}(\cdot | \phi(s, o), a)} \mathop{\mathbb{E}}_{o' \sim \overline{\mathcal{O}}(\cdot | \bar{s}')} \overline{V}_{\bar{\pi}}\big(h \cdot a \cdot o'\big) \right.$$

$$\left. - \mathop{\mathbb{E}}_{\bar{s} \sim \bar{\tau}^*(h)} \mathop{\mathbb{E}}_{\bar{s}' \sim \overline{\mathbf{P}}(\cdot | \bar{s}, a)} \mathop{\mathbb{E}}_{o' \sim \overline{\mathcal{O}}(\cdot | \bar{s}')} \overline{V}_{\bar{\pi}}\big(h \cdot a \cdot o'\big) \right|$$

$$\text{(by definition of } L_{\mathbf{P}} \text{)}$$

$$= \gamma \cdot \mathop{\mathbb{E}}_{h \sim \mathcal{H}_{\bar{\pi}} \ a \sim \bar{\pi}(\cdot | \varphi^*(h))} \left| \mathop{\mathbb{E}}_{s \sim \tau^*(h)} \mathop{\mathbb{E}}_{s' \sim \mathbf{P}(\cdot | s, a)} \mathop{\mathbb{E}}_{o' \sim \mathcal{O}(\cdot | s', a)} \left[ V_{\bar{\pi}}\big(h \cdot a \cdot o'\big) - \mathop{\mathbb{E}}_{\hat{o}' \sim \overline{\mathcal{O}}(\cdot | \phi(s', o'))} \overline{V}_{\bar{\pi}}\big(h \cdot a \cdot \hat{o}'\big) \right] \right|$$

$$+ \gamma K_{\overline{V}} L_{\mathbf{P}}$$

$$+ \gamma \cdot \mathop{\mathbb{E}}_{h, o \sim \mathcal{H}_{\bar{\pi}} \ a \sim \bar{\pi}(\cdot | \varphi^*(h))} \left| \left[ \mathop{\mathbb{E}}_{s \sim \tau^*(h)} \mathop{\mathbb{E}}_{\bar{s}' \sim \overline{\mathbf{P}}(\cdot | \phi(s, o), a)} \mathop{\mathbb{E}}_{o' \sim \overline{\mathcal{O}}(\cdot | \bar{s}')} \overline{V}_{\bar{\pi}}\big(h \cdot a \cdot o'\big) \right. \right.$$

$$\left. - \mathop{\mathbb{E}}_{\bar{s} \sim \varphi^*(h)} \mathop{\mathbb{E}}_{\bar{s}' \sim \overline{\mathbf{P}}(\cdot | \bar{s}, a)} \mathop{\mathbb{E}}_{o' \sim \overline{\mathcal{O}}(\cdot | \bar{s}')} \overline{V}_{\bar{\pi}}\big(h \cdot a \cdot o'\big) \right]$$

$$\left. + \left[ \mathop{\mathbb{E}}_{\bar{s} \sim \varphi^*(h)} \mathop{\mathbb{E}}_{\bar{s}' \sim \overline{\mathbf{P}}(\cdot | \bar{s}, a)} \mathop{\mathbb{E}}_{o' \sim \overline{\mathcal{O}}(\cdot | \bar{s}')} \overline{V}_{\bar{\pi}}\big(h \cdot a \cdot o'\big) - \mathop{\mathbb{E}}_{\bar{s} \sim \bar{\tau}^*(h)} \mathop{\mathbb{E}}_{\bar{s}' \sim \overline{\mathbf{P}}(\cdot | \bar{s}, a)} \mathop{\mathbb{E}}_{o' \sim \overline{\mathcal{O}}(\cdot | \bar{s}')} \overline{V}_{\bar{\pi}}\big(h \cdot a \cdot o'\big) \right] \right|$$

$$\text{(the belief encoder } \varphi \text{ comes into play)}$$

$$\leqslant \gamma \cdot \mathop{\mathbb{E}}_{h \sim \mathcal{H}_{\bar{\pi}} \ a \sim \bar{\pi}(\cdot | \varphi^*(h))} \left| \mathop{\mathbb{E}}_{s \sim \tau^*(h)} \mathop{\mathbb{E}}_{s' \sim \mathbf{P}(\cdot | s, a)} \mathop{\mathbb{E}}_{o' \sim \mathcal{O}(\cdot | s', a)} \left[ V_{\bar{\pi}}\big(h \cdot a \cdot o'\big) - \mathop{\mathbb{E}}_{\hat{o}' \sim \overline{\mathcal{O}}(\cdot | \phi(s', o'))} \overline{V}_{\bar{\pi}}\big(h \cdot a \cdot \hat{o}'\big) \right] \right|$$

$$+ \gamma K_{\overline{V}} L_{\mathbf{P}}$$

$$+ \gamma \cdot \mathop{\mathbb{E}}_{h, o \sim \mathcal{H}_{\bar{\pi}} \ a \sim \bar{\pi}(\cdot | \varphi^*(h))} \left| \mathop{\mathbb{E}}_{s \sim \tau^*(h)} \mathop{\mathbb{E}}_{\bar{s}' \sim \overline{\mathbf{P}}(\cdot | \phi(s, o), a)} \mathop{\mathbb{E}}_{o' \sim \overline{\mathcal{O}}(\cdot | \bar{s}')} \overline{V}_{\bar{\pi}}\big(h \cdot a \cdot o'\big) \right.$$

$$\left. - \mathop{\mathbb{E}}_{\bar{s} \sim \varphi^*(h)} \mathop{\mathbb{E}}_{\bar{s}' \sim \overline{\mathbf{P}}(\cdot | \bar{s}, a)} \mathop{\mathbb{E}}_{o' \sim \overline{\mathcal{O}}(\cdot | \bar{s}')} \overline{V}_{\bar{\pi}}\big(h \cdot a \cdot o'\big) \right|$$

$$+ \gamma \cdot \mathop{\mathbb{E}}_{h \sim \mathcal{H}_{\bar{\pi}} \ a \sim \bar{\pi}(\cdot | \varphi^*(h))} \left| \mathop{\mathbb{E}}_{\bar{s} \sim \varphi^*(h)} \mathop{\mathbb{E}}_{\bar{s}' \sim \overline{\mathbf{P}}(\cdot | \bar{s}, a)} \mathop{\mathbb{E}}_{o' \sim \overline{\mathcal{O}}(\cdot | \bar{s}')} \overline{V}_{\bar{\pi}}\big(h \cdot a \cdot o'\big) \right.$$

$$- \mathop{\mathbb{E}}_{\bar{s} \sim \bar{\tau}^*(h)} \mathop{\mathbb{E}}_{\bar{s}' \sim \overline{\mathbf{P}}(\cdot|\bar{s},a)} \mathop{\mathbb{E}}_{o' \sim \overline{\mathcal{O}}(\cdot|\bar{s}')} \overline{V}_{\overline{\pi}}(h \cdot a \cdot o') \Bigg|$$

$$\textcolor{blue}{(\text{ triangular inequality})}$$

$$\leqslant \gamma \cdot \mathop{\mathbb{E}}_{h \sim \mathcal{H}_{\overline{\pi}}} \mathop{\mathbb{E}}_{a \sim \overline{\pi}(\cdot|\varphi^*(h))} \Bigg| \mathop{\mathbb{E}}_{s \sim \tau^*(h)} \mathop{\mathbb{E}}_{s' \sim \mathbf{P}(\cdot|s,a)} \mathop{\mathbb{E}}_{o' \sim \mathcal{O}(\cdot|s',a)} \Bigg[ V_{\overline{\pi}}(h \cdot a \cdot o') - \mathop{\mathbb{E}}_{\hat{o}' \sim \overline{\mathcal{O}}(\cdot|\phi(s',o'))} \overline{V}_{\overline{\pi}}(h \cdot a \cdot \hat{o}') \Bigg] \Bigg|$$

$$+ \gamma K_{\overline{V}} L_{\mathbf{P}}$$

$$+ \gamma \cdot \mathop{\mathbb{E}}_{h,o \sim \mathcal{H}_{\overline{\pi}}} \mathop{\mathbb{E}}_{a \sim \overline{\pi}(\cdot|\varphi^*(h))} \mathop{\mathbb{E}}_{s \sim \tau^*(h)} \mathop{\mathbb{E}}_{\bar{s} \sim \varphi^*(h)} \Bigg| \mathop{\mathbb{E}}_{\bar{s}' \sim \overline{\mathbf{P}}(\cdot|\phi(s,o),a)} \Bigg[ \mathop{\mathbb{E}}_{o' \sim \overline{\mathcal{O}}(\cdot|\bar{s}')} \overline{V}_{\overline{\pi}}(h \cdot a \cdot o') \Bigg]$$

$$- \mathop{\mathbb{E}}_{\bar{s}' \sim \overline{\mathbf{P}}(\cdot|\bar{s},a)} \Bigg[ \mathop{\mathbb{E}}_{o' \sim \overline{\mathcal{O}}(\cdot|\bar{s}')} \overline{V}_{\overline{\pi}}(h \cdot a \cdot o') \Bigg] \Bigg|$$

$$+ \gamma \cdot \mathop{\mathbb{E}}_{h \sim \mathcal{H}_{\overline{\pi}}} \mathop{\mathbb{E}}_{a \sim \overline{\pi}(\cdot|\varphi^*(h))} \Bigg| \mathop{\mathbb{E}}_{\bar{s} \sim \varphi^*(h)} \mathop{\mathbb{E}}_{\bar{s}' \sim \overline{\mathbf{P}}(\cdot|\bar{s},a)} \mathop{\mathbb{E}}_{o' \sim \overline{\mathcal{O}}(\cdot|\bar{s}')} \overline{V}_{\overline{\pi}}(h \cdot a \cdot o')$$

$$- \mathop{\mathbb{E}}_{\bar{s} \sim \bar{\tau}^*(h)} \mathop{\mathbb{E}}_{\bar{s}' \sim \overline{\mathbf{P}}(\cdot|\bar{s},a)} \mathop{\mathbb{E}}_{o' \sim \overline{\mathcal{O}}(\cdot|\bar{s}')} \overline{V}_{\overline{\pi}}(h \cdot a \cdot o') \Bigg|$$

$$\textcolor{blue}{(\text{ Jensen's inequality})}$$

$$\leqslant \gamma \cdot \mathop{\mathbb{E}}_{h \sim \mathcal{H}_{\overline{\pi}}} \mathop{\mathbb{E}}_{a \sim \overline{\pi}(\cdot|\varphi^*(h))} \Bigg| \mathop{\mathbb{E}}_{s \sim \tau^*(h)} \mathop{\mathbb{E}}_{s' \sim \mathbf{P}(\cdot|s,a)} \mathop{\mathbb{E}}_{o' \sim \mathcal{O}(\cdot|s',a)} \Bigg[ V_{\overline{\pi}}(h \cdot a \cdot o') - \mathop{\mathbb{E}}_{\hat{o}' \sim \overline{\mathcal{O}}(\cdot|\phi(s',o'))} \overline{V}_{\overline{\pi}}(h \cdot a \cdot \hat{o}') \Bigg] \Bigg|$$

$$+ \gamma K_{\overline{V}} L_{\mathbf{P}}$$

$$+ \gamma \cdot \mathop{\mathbb{E}}_{h,o \sim \mathcal{H}_{\overline{\pi}}} \mathop{\mathbb{E}}_{a \sim \overline{\pi}(\cdot|\varphi^*(h))} \mathop{\mathbb{E}}_{s \sim \tau^*(h)} \mathop{\mathbb{E}}_{\bar{s} \sim \varphi^*(h)} K_{\overline{V}} \mathcal{W}_{\bar{d}} \big( \overline{\mathbf{P}}(\cdot \mid \phi(s,o),a), \overline{\mathbf{P}}(\cdot \mid \bar{s},a) \big)$$

$$+ \gamma \cdot \mathop{\mathbb{E}}_{h \sim \mathcal{H}_{\overline{\pi}}} \mathop{\mathbb{E}}_{a \sim \overline{\pi}(\cdot|\varphi^*(h))} \Bigg| \mathop{\mathbb{E}}_{\bar{s} \sim \varphi^*(h)} \mathop{\mathbb{E}}_{\bar{s}' \sim \overline{\mathbf{P}}(\cdot|\bar{s},a)} \mathop{\mathbb{E}}_{o' \sim \overline{\mathcal{O}}(\cdot|\bar{s}')} \overline{V}_{\overline{\pi}}(h \cdot a \cdot o')$$

$$- \mathop{\mathbb{E}}_{\bar{s} \sim \bar{\tau}^*(h)} \mathop{\mathbb{E}}_{\bar{s}' \sim \overline{\mathbf{P}}(\cdot|\bar{s},a)} \mathop{\mathbb{E}}_{o' \sim \overline{\mathcal{O}}(\cdot|\bar{s}')} \overline{V}_{\overline{\pi}}(h \cdot a \cdot o') \Bigg|$$

$$\textcolor{blue}{(\text{ as } \lambda \to 0, \text{ by Lem. F.6})}$$

$$= \gamma \cdot \mathop{\mathbb{E}}_{h \sim \mathcal{H}_{\overline{\pi}}} \mathop{\mathbb{E}}_{a \sim \overline{\pi}(\cdot|\varphi^*(h))} \Bigg| \mathop{\mathbb{E}}_{s \sim \tau^*(h)} \mathop{\mathbb{E}}_{s' \sim \mathbf{P}(\cdot|s,a)} \mathop{\mathbb{E}}_{o' \sim \mathcal{O}(\cdot|s',a)} \Bigg[ V_{\overline{\pi}}(h \cdot a \cdot o') - \mathop{\mathbb{E}}_{\hat{o}' \sim \overline{\mathcal{O}}(\cdot|\phi(s',o'))} \overline{V}_{\overline{\pi}}(h \cdot a \cdot \hat{o}') \Bigg] \Bigg|$$

$$+ \gamma K_{\overline{V}} \cdot \left( L_{\mathbf{P}} + \textcolor{blue}{L_{\overline{\mathbf{P}}}^{\varphi}} \right)$$

$$+ \gamma \cdot \mathop{\mathbb{E}}_{h \sim \mathcal{H}_{\overline{\pi}}} \mathop{\mathbb{E}}_{a \sim \overline{\pi}(\cdot|\varphi^*(h))} \Bigg| \mathop{\mathbb{E}}_{\bar{s} \sim \varphi^*(h)} \Bigg[ \mathop{\mathbb{E}}_{\bar{s}' \sim \overline{\mathbf{P}}(\cdot|\bar{s},a)} \mathop{\mathbb{E}}_{o' \sim \overline{\mathcal{O}}(\cdot|\bar{s}')} \overline{V}_{\overline{\pi}}(h \cdot a \cdot o') \Bigg]$$

$$- \mathop{\mathbb{E}}_{\bar{s} \sim \bar{\tau}^*(h)} \Bigg[ \mathop{\mathbb{E}}_{\bar{s}' \sim \overline{\mathbf{P}}(\cdot|\bar{s},a)} \mathop{\mathbb{E}}_{o' \sim \overline{\mathcal{O}}(\cdot|\bar{s}')} \overline{V}_{\overline{\pi}}(h \cdot a \cdot o') \Bigg] \Bigg|$$

$$\textcolor{blue}{(\text{ by definition of } L_{\overline{\mathbf{P}}}^{\varphi}, \text{ Eq. 5})}$$

$$\leqslant \gamma \cdot \mathop{\mathbb{E}}_{h \sim \mathcal{H}_{\overline{\pi}}} \mathop{\mathbb{E}}_{a \sim \overline{\pi}(\cdot|\varphi^*(h))} \Bigg| \mathop{\mathbb{E}}_{s \sim \tau^*(h)} \mathop{\mathbb{E}}_{s' \sim \mathbf{P}(\cdot|s,a)} \mathop{\mathbb{E}}_{o' \sim \mathcal{O}(\cdot|s',a)} \Bigg[ V_{\overline{\pi}}(h \cdot a \cdot o') - \mathop{\mathbb{E}}_{\hat{o}' \sim \overline{\mathcal{O}}(\cdot|\phi(s',o'))} \overline{V}_{\overline{\pi}}(h \cdot a \cdot \hat{o}') \Bigg] \Bigg|$$

$$+ \gamma K_{\overline{V}} \cdot \left( L_{\mathbf{P}} + L_{\overline{\mathbf{P}}}^{\varphi} \right)$$

$$+ \textcolor{blue}{\gamma \cdot \mathop{\mathbb{E}}_{h \sim \mathcal{H}_{\overline{\pi}}} \mathop{\mathbb{E}}_{a \sim \overline{\pi}(\cdot|\varphi^*(h))} K_{\overline{V}} \mathcal{W}_{\bar{d}} \left( \bar{\tau}^*(h), \varphi^*(h) \right)}$$

$$\textcolor{blue}{(\text{as } \lambda \to 0, \text{ by Lem. F.6; note that Wasserstein is symmetric since it is a distance metric (Villani, 2009))}}$$

$$
\begin{aligned}
\leqslant \quad & \gamma \cdot \underset{h \sim \mathcal{H}_{\overline{\pi}}}{\mathbb{E}} \underset{a \sim \pi(\cdot|\varphi^*(h))}{\mathbb{E}} \left| \underset{s \sim \tau^*(h)}{\mathbb{E}} \underset{s' \sim \mathbf{P}(\cdot|s,a)}{\mathbb{E}} \underset{o' \sim \mathcal{O}(\cdot|s',a)}{\mathbb{E}} \left[ V_{\overline{\pi}}(h \cdot a \cdot o') - \underset{\hat{o}' \sim \overline{\mathcal{O}}(\cdot|\phi(s',o'))}{\mathbb{E}} \overline{V}_{\overline{\pi}}(h \cdot a \cdot \hat{o}') \right] \right| \\
& + \gamma K_{\overline{V}} \cdot \left( L_{\mathbf{P}} + L_{\overline{\mathbf{P}}}^{\varphi} + {\color{blue}L_{\overline{\tau}}} \right) \hspace{3cm} \text{{\color{blue}(by definition of $L_{\overline{\tau}}$, Eq. 5)}} \\[4pt]
= \quad & \gamma \cdot \underset{h,o \sim \mathcal{H}_{\overline{\pi}}}{\mathbb{E}} \underset{a \sim \pi(\cdot|\varphi^*(h))}{\mathbb{E}} \left| \underset{s \sim \tau^*(h)}{\mathbb{E}} \underset{s',o' \sim \mathbf{P}_{\Omega}(\cdot|s,o,a)}{\mathbb{E}} \left[ \left( V_{\overline{\pi}}(h \cdot a \cdot o') - \overline{V}_{\overline{\pi}}(h \cdot a \cdot o') \right) \right.\right. \\
& \hspace{5cm} \left.\left. + \left( \overline{V}_{\overline{\pi}}(h \cdot a \cdot o') - \underset{\hat{o}' \sim \overline{\mathcal{O}}(\cdot|\phi(s',o'))}{\mathbb{E}} \overline{V}_{\overline{\pi}}(h \cdot a \cdot \hat{o}') \right) \right] \right| \\
& + \gamma K_{\overline{V}} \cdot \left( L_{\mathbf{P}} + L_{\overline{\mathbf{P}}}^{\varphi} + L_{\overline{\tau}} \right) \\[4pt]
\leqslant \quad & \gamma \cdot \underset{h,o \sim \mathcal{H}_{\overline{\pi}}}{\mathbb{E}} \underset{a \sim \pi(\cdot|\varphi^*(h))}{\mathbb{E}} \left| \underset{s \sim \tau^*(h)}{\mathbb{E}} \underset{s',o' \sim \mathbf{P}_{\Omega}(\cdot|s,o,a)}{\mathbb{E}} \left[ V_{\overline{\pi}}(h \cdot a \cdot o') - \overline{V}_{\overline{\pi}}(h \cdot a \cdot o') \right] \right| \\
& + \gamma \cdot \underset{h,o \sim \mathcal{H}_{\overline{\pi}}}{\mathbb{E}} \underset{a \sim \pi(\cdot|\varphi^*(h))}{\mathbb{E}} \left| \underset{s \sim \tau^*(h)}{\mathbb{E}} \underset{s',o' \sim \mathbf{P}_{\Omega}(\cdot|s,o,a)}{\mathbb{E}} \left[ \overline{V}_{\overline{\pi}}(h \cdot a \cdot o') - \underset{\hat{o}' \sim \overline{\mathcal{O}}(\cdot|\phi(s',o'))}{\mathbb{E}} \overline{V}_{\overline{\pi}}(h \cdot a \cdot \hat{o}') \right] \right| \\
& + \gamma K_{\overline{V}} \cdot \left( L_{\mathbf{P}} + L_{\overline{\mathbf{P}}}^{\varphi} + L_{\overline{\tau}} \right) \hspace{3cm} \text{{\color{blue}(triangular inequality)}} \\[4pt]
\leqslant \quad & \gamma \cdot \underset{h,o \sim \mathcal{H}_{\overline{\pi}}}{\mathbb{E}} \underset{a \sim \pi(\cdot|\varphi^*(h))}{\mathbb{E}} \left| \underset{s \sim \tau^*(h)}{\mathbb{E}} \underset{s',o' \sim \mathbf{P}_{\Omega}(\cdot|s,o,a)}{\mathbb{E}} \left[ V_{\overline{\pi}}(h \cdot a \cdot o') - \overline{V}_{\overline{\pi}}(h \cdot a \cdot o') \right] \right| \\
& + \gamma \cdot \underset{h,o \sim \mathcal{H}_{\overline{\pi}}}{\mathbb{E}} \underset{a \sim \pi(\cdot|\varphi^*(h))}{\mathbb{E}} \underset{s \sim \tau^*(h)}{\mathbb{E}} \underset{s' \sim \mathbf{P}(\cdot|s,a)}{\mathbb{E}} \left| \underset{o' \sim \mathcal{O}(\cdot|s',a)}{\mathbb{E}} \left[ \overline{V}_{\overline{\pi}}(h \cdot a \cdot o') - \underset{\hat{o}' \sim \overline{\mathcal{O}}(\cdot|\phi(s',o'))}{\mathbb{E}} \overline{V}_{\overline{\pi}}(h \cdot a \cdot \hat{o}') \right] \right| \\
& + \gamma K_{\overline{V}} \cdot \left( L_{\mathbf{P}} + L_{\overline{\mathbf{P}}}^{\varphi} + L_{\overline{\tau}} \right) \hspace{2cm} \text{{\color{blue}(by definition of $\mathbf{P}_{\Omega}$ and the Jensen's inequality)}} \\[4pt]
\leqslant \quad & \gamma \cdot \underset{h,o \sim \mathcal{H}_{\overline{\pi}}}{\mathbb{E}} \underset{a \sim \pi(\cdot|\varphi^*(h))}{\mathbb{E}} \left| \underset{s \sim \tau^*(h)}{\mathbb{E}} \underset{s',o' \sim \mathbf{P}_{\Omega}(\cdot|s,o,a)}{\mathbb{E}} \left[ V_{\overline{\pi}}(h \cdot a \cdot o') - \overline{V}_{\overline{\pi}}(h \cdot a \cdot o') \right] \right| \\
& + \gamma \cdot \underset{h,o \sim \mathcal{H}_{\overline{\pi}}}{\mathbb{E}} \underset{a \sim \pi(\cdot|\varphi^*(h))}{\mathbb{E}} \underset{s \sim \tau^*(h)}{\mathbb{E}} \underset{s' \sim \mathbf{P}(\cdot|s,a)}{\mathbb{E}} K_{\overline{V}} d_{TV}\left( \mathcal{O}(\cdot \mid s',a), \underset{o' \sim s',a}{\mathbb{E}} \overline{\mathcal{O}}(\cdot \mid \phi(s',o')) \right) \\
& + \gamma K_{\overline{V}} \cdot \left( L_{\mathbf{P}} + L_{\overline{\mathbf{P}}}^{\varphi} + L_{\overline{\tau}} \right) \hspace{3cm} \text{{\color{blue}(cf. Prop. F.5 and Lem F.6)}} \\[4pt]
= \quad & \gamma \cdot \underset{h,o \sim \mathcal{H}_{\overline{\pi}}}{\mathbb{E}} \underset{a \sim \pi(\cdot|\varphi^*(h))}{\mathbb{E}} \left| \underset{s \sim \tau^*(h)}{\mathbb{E}} \underset{s',o' \sim \mathbf{P}_{\Omega}(\cdot|s,o,a)}{\mathbb{E}} \left[ V_{\overline{\pi}}(h \cdot a \cdot o') - \overline{V}_{\overline{\pi}}(h \cdot a \cdot o') \right] \right| \\
& + \gamma K_{\overline{V}} \cdot \left( L_{\mathbf{P}} + L_{\overline{\mathbf{P}}}^{\varphi} + L_{\overline{\tau}} + {\color{blue}L_{\mathcal{O}}} \right) \hspace{3cm} \text{{\color{blue}(by definition of $L_{\mathcal{O}}$, Eq. 4)}} \\[4pt]
\leqslant \quad & \gamma \cdot \underset{h,o \sim \mathcal{H}_{\overline{\pi}}}{\mathbb{E}} \underset{a \sim \pi(\cdot|\varphi^*(h))}{\mathbb{E}} \underset{s \sim \tau^*(h)}{\mathbb{E}} \underset{s',o' \sim \mathbf{P}_{\Omega}(\cdot|s,o,a)}{\mathbb{E}} \left| V_{\overline{\pi}}(h \cdot a \cdot o') - \overline{V}_{\overline{\pi}}(h \cdot a \cdot o') \right| \\
& + \gamma K_{\overline{V}} \cdot \left( L_{\mathbf{P}} + L_{\overline{\mathbf{P}}}^{\varphi} + L_{\overline{\tau}} + L_{\mathcal{O}} \right) \hspace{3cm} \text{{\color{blue}(Jensen's inequality)}} \\[4pt]
= \quad & \gamma \cdot \underset{h,o \sim \mathcal{H}_{\overline{\pi}}}{\mathbb{E}} \left| V_{\overline{\pi}}(h) - \overline{V}_{\overline{\pi}}(h) \right| + \gamma K_{\overline{V}} \cdot \left( L_{\mathbf{P}} + L_{\overline{\mathbf{P}}}^{\varphi} + L_{\overline{\tau}} + L_{\mathcal{O}} \right) \\
& \hspace{1cm} \text{{\color{blue}($\mathcal{H}_{\overline{\pi}}$ is a stationary distribution (Lem. D.1) which allows us to apply the stationary property)}}
\end{aligned}
$$

**Putting all together.** To recap, by Part 1 and 2, we have:

$$
\begin{aligned}
\underset{h \sim \mathcal{H}_{\overline{\pi}}}{\mathbb{E}} \left| V_{\overline{\pi}}(h) - \overline{V}_{\overline{\pi}}(h) \right| &\leqslant L_{\mathcal{R}} + L_{\overline{\mathcal{R}}}^{\varphi} + \overline{\mathcal{R}}^{\star} L_{\overline{\tau}} + \gamma \cdot \underset{h \sim \mathcal{H}_{\overline{\pi}}}{\mathbb{E}} \left| V_{\overline{\pi}}(h) - \overline{V}_{\overline{\pi}}(h) \right| \\
& \quad + \gamma K_{\overline{V}} \cdot \left( L_{\mathbf{P}} + L_{\overline{\mathbf{P}}}^{\varphi} + L_{\overline{\tau}} + L_{\mathcal{O}} \right) \\[4pt]
\underset{h \sim \mathcal{H}_{\overline{\pi}}}{\mathbb{E}} \left| V_{\overline{\pi}}(h) - \overline{V}_{\overline{\pi}}(h) \right| \cdot (1 - \gamma) &\leqslant L_{\mathcal{R}} + L_{\overline{\mathcal{R}}}^{\varphi} + \overline{\mathcal{R}}^{\star} L_{\overline{\tau}} + \gamma K_{\overline{V}} \cdot \left( L_{\mathbf{P}} + L_{\overline{\mathbf{P}}}^{\varphi} + L_{\overline{\tau}} + L_{\mathcal{O}} \right) \\[4pt]
\underset{h \sim \mathcal{H}_{\overline{\pi}}}{\mathbb{E}} \left| V_{\overline{\pi}}(h) - \overline{V}_{\overline{\pi}}(h) \right| &\leqslant \frac{L_{\mathcal{R}} + L_{\overline{\mathcal{R}}}^{\varphi} + \overline{\mathcal{R}}^{\star} L_{\overline{\tau}} + \gamma K_{\overline{V}} \cdot \left( L_{\mathbf{P}} + L_{\overline{\mathbf{P}}}^{\varphi} + L_{\overline{\tau}} + L_{\mathcal{O}} \right)}{1 - \gamma}
\end{aligned}
$$

which finally concludes the proof. □

## F.5 Representation Quality Bound

We start by showing that the optimal *latent* value function is *almost* Lipschitz continuous in the latent belief space. Coupled with Theorem 3.2, this result allows to show that whenever *any histories are encoded to close representations, their values (i.e., the return obtained from that history points) are guaranteed to be close as well* whenever the losses introduced in Sec. 3.2 are minimized and go to zero. Phrased differently, this Theorem ensures that the representation induced by our encoder is suitable to optimize the value function since the distance between beliefs in the latent space characterizes the distance of behaviors of the agent in the original environment. The latent belief space thus captures the necessary information to learn a policy that optimizes the expected return.

**Definition F.7** (Almost Lipschitzness). *Let $\mathcal{X}$ be a measurable set equipped with a metric $d \colon \mathcal{X} \to [0, \infty)$ and $f \colon \mathcal{X} \to \mathbb{R}$. We say that $f$ is* almost *Lipschitz continuous (e.g., Vanderbei 1991) iff for all $\epsilon > 0$, there is a constant $K \geqslant 0$ so that $|f(x_1) - f(x_2)| \leqslant K d(x_1, x_2) + \epsilon$ for any $x_1, x_2 \in \mathcal{X}$.*

*Notation* 1 (Optimal value function). For any MDP $\mathcal{M}$, let $\pi^\star$ be an optimal policy of $\mathcal{M}$, then we write $V^\star$ for $V_{\pi^\star}$ (see Property F.2 for the values of a POMDP).

**Lemma F.8.** *Let $\mathcal{P} = \langle \mathcal{M}, \Omega, \mathcal{O} \rangle$ be a POMDP with underlying MDP $\mathcal{M} = \langle \mathcal{S}, \mathcal{A}, \mathbf{P}, \mathcal{R}, s_I, \gamma \rangle$. Assume that $\mathcal{P}$ is finite, i.e., $\mathcal{S}$, $\mathcal{A}$, and $\Omega$ are finite sets. Then, $V^\star$ is almost Lipschitz continuous.*

*Proof.* Define $\mathcal{V}$ as the set of real-valued bounded functions $V \colon \mathcal{B} \to \mathbb{R}$ and $U \colon \mathcal{B} \times \mathcal{A} \times \mathcal{V} \to \mathcal{V}$ as

$$U(b, a, V) = \mathcal{R}_\mathcal{B}(b, a) + \underset{b' \sim \mathbf{P}_\mathcal{B}(\cdot \mid b, a)}{\mathbb{E}} \left[ \gamma V(b') \right].$$

The Bellman update operator is defined as $\mathcal{U} \colon \mathcal{V} \to \mathcal{V}$ as $(\mathcal{U}V)(b) = \max_{a \in \mathcal{A}} U(b, a, V)$ and is an isotone mapping that is a contraction under the supremum norm with fixed point $V^\star$, i.e., $V^\star = \mathcal{U}V^\star$ (Puterman, 1994; Hauskrecht, 2000; Sutton & Barto, 1998). Furthermore, for any initial value function $V_0 \in \mathcal{V}$, the sequence resulting from value iteration (VI), $V_{i+1} = \mathcal{U}V_i$, converges to $V^\star$ (with linear convergence rate $\gamma$ Puterman (1994)): for any $\epsilon' > 0$, there is a $i \in \mathbb{N}$ so that for all $j \geqslant i$, $\left\| V_j - V^\star \right\|_\infty \leqslant \epsilon'$. Now, let $\epsilon > 0$; in particular, the latter statement holds for $\epsilon' = \epsilon/2$. Since the convergence of VI holds for any initial value, we assume that $V_0 \in \mathcal{V}$ has been chosen as a *piecewise linear convex* (PWLC) function. Then, for all $i \geqslant 0$, $V_i$ is also PWLC (Sondik, 1978; Smallwood & Sondik, 1973; Hauskrecht, 2000). This means that $V_i$ can be defined a finite collection of linear functions. Thus, $V_i$ is also $2K$-Lipschitz: one just need take $2K$ as the higher slope of these functions (in absolute value). In consequence, for any pair of beliefs $b_1, b_2 \in \mathcal{B}$,

$$\begin{aligned}
&|V^\star(b_1) - V^\star(b_2)| \\
=\ & |V^\star(b_1) - V_i(b_1) + V_i(b_1) - V_i(b_2) + V_i(b_2) - V^\star(b_2)| \\
\leqslant\ & |V^\star(b_1) - V_i(b_1)| + |V_i(b_1) - V_i(b_2)| + |V_i(b_2) - V^\star(b_2)| && \text{(Triangular inequality)} \\
\leqslant\ & 2\epsilon' + |V_i(b_1) - V_i(b_2)| && \text{(by the convergence of VI)} \\
=\ & \epsilon + |V_i(b_1) - V_i(b_2)| \\
\leqslant\ & \epsilon + K \cdot d_{TV}(b_1, b_2), && \text{(since } d_{TV}(b_1, b_2) = 1/2 \left\| b_1 - b_2 \right\|_1 \text{)}
\end{aligned}$$

which means that $V^\star$ is almost Lipschitz, by definition. □

Lemma F.8 can be applied to $\overline{\mathcal{P}}$. To see why, notice first the state space of $\overline{\mathcal{P}}$ is finite. Second, $\overline{\mathcal{O}}$ is deterministic by definition (it consists of the Dirac centered on the observation produced by $\overline{\mathcal{O}}_\mu$), so the observations of $\overline{\mathcal{P}}$ can be limited to the union of the supports of $\overline{\mathcal{O}}(\cdot \mid \bar{s})$, ranging over all latent states $\bar{s} \in \overline{\mathcal{S}}$. The resulting set is also finite. Note that the same reasoning holds for any latent observation function with finite support. Therefore, we obtain the following corollary.

**Corollary F.9.** *When the WAE-MDP is in the zero-temperature limit (see Appendix E), the optimal latent value function of $\overline{\mathcal{P}}$ is almost Lipschitz-continuous.*

**Theorem 3.3** (Representation quality). *Let $\bar{\pi}^\star$ be an optimal policy of $\overline{\mathcal{P}}$, then for any $\epsilon > 0$ and $n \geqslant 0$, there is a $K \geqslant 0$ so that for any histories $h_1, h_2$ of length at most $n$ that are measurable*

*under $\mathcal{P}$ and $\overline{\mathcal{P}}$ with $\varphi^*(h_1) = \bar{b}_1$ and $\varphi^*(h_2) = \bar{b}_2$, the representation induced by $\varphi$ yields:*

$$|V_{\bar{\pi}^\star}(h_1) - V_{\bar{\pi}^\star}(h_2)| \quad \leqslant \quad K\mathcal{W}_{\bar{d}}(\bar{b}_1, \bar{b}_2) \;+\; \epsilon \;+\; \frac{KL_{\bar{\tau}} + \mathcal{L}}{1 - \gamma}\left(\frac{1}{\mathcal{H}_{\bar{\pi}^\star}(h_1)} + \frac{1}{\mathcal{H}_{\bar{\pi}^\star}(h_2)}\right). \quad (7)$$

*Proof.* First, observe that for any measurable history $h$, $|V_{\bar{\pi}^\star}(h) - \overline{V}_{\bar{\pi}^\star}(h)| \leqslant \mathcal{H}_{\bar{\pi}^\star}(h)^{-1} \cdot \mathbb{E}_{h' \sim \mathcal{H}_{\bar{\pi}^\star}} |V_{\bar{\pi}^\star}(h') - \overline{V}_{\bar{\pi}^\star}(h')|$ (see, e.g., Gelada et al. 2019).

Let $\epsilon > 0$, $n \geqslant 0$, and $h_1, h_2$ be two measurable histories so that $h_i = \langle a_{0:T-1}, o_{1:T} \rangle$ with $T \leqslant n$ and $o_t \in \text{supp}(\overline{\mathcal{O}}(\cdot \mid \bar{s}))$ for some $\bar{s} \in \overline{\mathcal{S}}$ and for all $t < T$, $i \in \{1, 2\}$ (the two histories are compliant with $\mathcal{P}$ and $\overline{\mathcal{P}}$). Then, we have:

$$|V_{\bar{\pi}^\star}(h_1) - V_{\bar{\pi}^\star}(h_2)|$$
$$= |V_{\bar{\pi}^\star}(h_1) - \overline{V}_{\bar{\pi}^\star}(h_1) + \overline{V}_{\bar{\pi}^\star}(h_1) - \overline{V}_{\bar{\pi}^\star}(h_2) + \overline{V}_{\bar{\pi}^\star}(h_2) - V_{\bar{\pi}^\star}(h_2)|$$
$$\leqslant |V_{\bar{\pi}^\star}(h_1) - \overline{V}_{\bar{\pi}^\star}(h_1)| + |\overline{V}_{\bar{\pi}^\star}(h_1) - \overline{V}_{\bar{\pi}^\star}(h_2)| + |\overline{V}_{\bar{\pi}^\star}(h_2) - V_{\bar{\pi}^\star}(h_2)| \quad \text{(Triangular inequality)}$$
$$\leqslant \mathcal{H}_{\bar{\pi}^\star}(h_1)^{-1} \mathop{\mathbb{E}}_{h \sim \mathcal{H}_{\bar{\pi}^\star}} |V_{\bar{\pi}^\star}(h) - \overline{V}_{\bar{\pi}^\star}(h)| + |\overline{V}_{\bar{\pi}^\star}(h_1) - \overline{V}_{\bar{\pi}^\star}(h_2)|$$
$$\quad + \mathcal{H}_{\bar{\pi}^\star}(h_2)^{-1} \mathop{\mathbb{E}}_{h \sim \mathcal{H}_{\bar{\pi}^\star}} |V_{\bar{\pi}^\star}(h) - \overline{V}_{\bar{\pi}^\star}(h)|$$
$$\leqslant |\overline{V}_{\bar{\pi}^\star}(h_1) - \overline{V}_{\bar{\pi}^\star}(h_2)| + \frac{\mathcal{L}}{1 - \gamma}\left(\mathcal{H}_{\bar{\pi}^\star}(h_1)^{-1} + \mathcal{H}_{\bar{\pi}^\star}(h_2)^{-1}\right) \quad \text{(Thm. 3.2)}$$

Consider $h_\perp = \arg\min \{\mathcal{H}_{\bar{\pi}*}(h) : h \in \text{supp}(\mathcal{H}_{\bar{\pi}*}) \cap \mathbf{H}(\mathcal{P}, \overline{\mathcal{P}}) \text{ and } |h| \leqslant n\}$, where $\text{supp}(\mathcal{H}_{\bar{\pi}^\star})$ denotes the support of $\mathcal{H}_{\bar{\pi}^\star}$, $|h|$ the length of the history $h$, and $\mathbf{H}(\mathcal{P}, \overline{\mathcal{P}})$ the set of histories compliant with both $\mathcal{P}$ and $\overline{\mathcal{P}}$, i.e., histories whose observations belong to $\bigcup_{\bar{s} \in \overline{\mathcal{S}}} \text{supp}(\overline{\mathcal{O}}(\cdot \mid \bar{s}))$. Notice that $h_\perp$ exists: we consider histories which are measurable in the two models (i.e., the original and latent POMDPs) and the observation space of the latent POMDP is finite (cf. the premise of Corollary F.9). So, the set of histories from which we extract $h_\perp$ consists of a finite number of action-observation branchings.

Recall that the latent value function (defined over the latent belief space) is almost Lipschitz continuous (Corollary F.9). In particular, for $\delta = \frac{\epsilon}{(1 + 2\mathcal{H}_{\bar{\pi}^\star}(h_\perp)^{-1})}$, there is a $K \geqslant 0$ so that for any $\bar{b}, \bar{b}' \in \overline{\mathcal{B}}$, $|\overline{V}^\star(b) - \overline{V}^\star(b')| \leqslant K\mathcal{W}_{\bar{d}}(b, b') + \delta$. Then:

$$|\overline{V}^\star(h_1) - \overline{V}^\star(h_2)|$$
$$= |\overline{V}^\star(\bar{\tau}^*(h_1)) - \overline{V}^\star(\bar{\tau}^*(h_2))|$$
$$= |\overline{V}^\star(\bar{\tau}^*(h_1)) - \overline{V}^\star(\varphi^*(h_1)) + \overline{V}^\star(\varphi^*(h_1)) - \overline{V}^\star(\varphi^*(h_2)) + \overline{V}^\star(\varphi^*(h_2)) - \overline{V}^\star(\bar{\tau}^*(h_2))|$$
$$\leqslant |\overline{V}^\star(\bar{\tau}^*(h_1)) - \overline{V}^\star(\varphi^*(h_1))| + |\overline{V}^\star(\varphi^*(h_1)) - \overline{V}^\star(\varphi^*(h_2))| + |\overline{V}^\star(\varphi^*(h_2)) - \overline{V}^\star(\bar{\tau}^*(h_2))|$$
$$\text{(Triangular inequality)}$$
$$\leqslant \mathcal{H}_{\bar{\pi}^\star}(h_1)^{-1} \mathop{\mathbb{E}}_{h \sim \mathcal{H}_{\bar{\pi}^\star}} |\overline{V}^\star(\bar{\tau}^*(h)) - \overline{V}^\star(\varphi^*(h))| + |\overline{V}^\star(b_1) - \overline{V}^\star(b_2)|$$
$$\quad + \mathcal{H}_{\bar{\pi}^\star}(h_2)^{-1} \mathop{\mathbb{E}}_{h \sim \mathcal{H}_{\bar{\pi}^\star}} |\overline{V}^\star(\bar{\tau}^*(h)) - \overline{V}^\star(\varphi^*(h))|$$
$$= \left(\mathcal{H}_{\bar{\pi}^\star}(h_1)^{-1} + \mathcal{H}_{\bar{\pi}^\star}(h_2)^{-1}\right) \mathop{\mathbb{E}}_{h \sim \mathcal{H}_{\bar{\pi}^\star}} |\overline{V}^\star(\bar{\tau}^*(h)) - \overline{V}^\star(\varphi^*(h))| + |\overline{V}^\star(b_1) - \overline{V}^\star(b_2)|$$
$$\leqslant \left(\mathcal{H}_{\bar{\pi}^\star}(h_1)^{-1} + \mathcal{H}_{\bar{\pi}^\star}(h_2)^{-1}\right) \mathop{\mathbb{E}}_{h \sim \mathcal{H}_{\bar{\pi}^\star}} [K\mathcal{W}_{\bar{d}}(\bar{\tau}^*(h), \varphi^*(h)) + \delta] + K\mathcal{W}_{\bar{d}}(\bar{b}_1, \bar{b}_2) + \delta$$
$$\text{($\overline{V}^\star$ is almost Lipschitz)}$$
$$= \left(\mathcal{H}_{\bar{\pi}^\star}(h_1)^{-1} + \mathcal{H}_{\bar{\pi}^\star}(h_2)^{-1}\right)(KL_{\bar{\tau}} + \delta) + K\mathcal{W}_{\bar{d}}(\bar{b}_1, \bar{b}_2) + \delta \quad \text{(by definition of } L_{\bar{\tau}})$$
$$= \left(\mathcal{H}_{\bar{\pi}^\star}(h_1)^{-1} + \mathcal{H}_{\bar{\pi}^\star}(h_2)^{-1}\right)KL_{\bar{\tau}} + K\mathcal{W}_{\bar{d}}(\bar{b}_1, \bar{b}_2) + \delta\left(1 + \mathcal{H}_{\bar{\pi}^\star}(h_1)^{-1} + \mathcal{H}_{\bar{\pi}^\star}(h_2)^{-1}\right)$$
$$\leqslant \left(\mathcal{H}_{\bar{\pi}^\star}(h_1)^{-1} + \mathcal{H}_{\bar{\pi}^\star}(h_2)^{-1}\right)KL_{\bar{\tau}} + K\mathcal{W}_{\bar{d}}(\bar{b}_1, \bar{b}_2) + \delta\left(1 + 2\mathcal{H}_{\bar{\pi}^\star}(h_\perp)^{-1}\right)$$

$$= \Big( \mathcal{H}_{\overline{\pi}^\star}(h_1)^{-1} + \mathcal{H}_{\overline{\pi}^\star}(h_2)^{-1} \Big) K L_{\overline{\tau}} + K \mathcal{W}_{\overline{d}} \left( \overline{b}_1, \overline{b}_2 \right) + \epsilon$$

Putting all together, we have that for any $\epsilon > 0$, $n \geq 0$, one can find a constant $K \geq 0$ so that:

$$\left| V_{\overline{\pi}^\star}(h_1) - V_{\overline{\pi}^\star}(h_2) \right| \quad \leq \quad K \mathcal{W}_{\overline{d}} \left( \overline{b}_1, \overline{b}_2 \right) \; + \; \varepsilon \; + \; \frac{KL_{\overline{\tau}} + \mathcal{L}}{1 - \gamma} \Big( \mathcal{H}_{\overline{\pi}*}(h_1)^{-1} + \mathcal{H}_{\overline{\pi}*}(h_2)^{-1} \Big).$$

$\square$

*Remark* 4 (Representation quality for latent policies). The bound of Theorem 3.3 also holds for any latent policy $\overline{\pi}$ which is executed in both models by processing histories through $\varphi$. To see why, notice that such policies can be encoded through a Mealy machine (Definition D.3) with observations as inputs (instead of states): one just need to use as memory states the reachable fragment of the latent belief space. Then, each time an observation $o_{t+1}$ is perceived subsequent to an action $a_t$, the memory $\overline{b}_t$ of the machine is updated to $\overline{b}_{t+1} = \varphi \big( \overline{b}_t, a_t, o_{t+1} \big)$.

Remarkably, when a horizon is fixed, the set of latent beliefs that can be reached in the Mealy machine is finite since (i) $\overline{\mathcal{P}}$ is finite, (ii) the initial latent belief is fixed, and (iii) each belief has at most $|\mathcal{A}| \cdot \left| \bigcup_{\overline{s} \in \overline{\mathcal{S}}} \overline{\mathcal{O}}(\cdot \mid \overline{s}) \right|$ successors.

Each function $V_i$ obtained from a sequence $(V_i)_{i \in \mathbb{N}}$ of value functions generated by VI to evaluate any policy *encoded through a finite state controller* is PWLC (Hansen, 1998). Since VI converges, one can find a value function $V_i$ from this sequence (for a finite $i \geq 0$) which ensures an error of at most $\varepsilon > 0$ w.r.t. the value of this particular policy. Consider $i$ as being the horizon used in the memory of the Mealy machine. Then, all the arguments from the proofs of Lemma F.8 and Theorem 3.3 can be applied to the latent policy encoded via this particular Mealy machine.

## G  ADDITIONAL DETAILS ON BACK PROPAGATION THROUGH TIME

In this section, we elaborate on why BPTT is not required, nor recommended, to learn a representation of the belief when using our algorithm.

In RNN-A2C (Figure 2, right) the hidden state is computed recursively, which is *in the same spirit* as how the belief is computed through our WBU (Figure 2, left). The motivation behind activating BPTT for RNN-A2C and disabling it for WBU lies in the difference between beliefs and hidden states and what they represent.

On the one hand, beliefs are state distributions which can be inferred from the previous belief and be themselves seen as Markovian states (in the belief MDP $\mathcal{M}_{\mathcal{B}}$). Therefore, one can see the belief update function as the transition function of an MDP, where an observation is additionally provided. Disabling BPTT is thus consistent with the literature of model based RL for learning Markovian transition functions (e.g., Gelada et al., 2019; François-Lavet et al., 2019; Delgrange et al., 2022). Notably, our goal is not targeted towards learning directly a history representation (from sequences) but rather on learning to reproduce the belief update rule, based on the latent stochastic transition function $\overline{\mathbf{P}}$, and observation function $\overline{\mathcal{O}}$ of the latent model. The latter are learned separately and considered fixed when learning the belief update (Section 4). Therefore, stopping the gradients does not produce approximate gradients. Furthermore, the error linked to this recursive update can be fully determined through the error of our latent model (Theorem 3.2). Then, enabling BPTT increases the risk of accumulating errors across time.

On the other hand, RNN hidden states are latent representations of the history and it is up to the RL algorithm to learn a useful representation of the history for the control task. This is in stark contrast to our approach: we separately learn a representation which is guaranteed to be well suited for the RL agent, and we use it as input of the policy. In R-A2C the main objective is to find a policy that optimizes the return conditioned on the full history, making it sound to use BPTT in that case. By contrast, in WBU, the RL agent uses the belief as a given state-signal to condition the value function and the policy, and therefore, the RL objective is not backpropagated through the belief encoder.

Finally, it is worth mentioning that as beliefs are computed with a forward recurrence, errors may accumulate over time. This means that if the belief $b_{t+k}$ diverges significantly from its correct value, then $b_{t+k+1}$ will diverge even more. Then, allowing gradients from the learning of $b_{t+k}$ to be computed based on an incorrect $b_{t+k+1}$ and further flow to $b_t$ may hinders the learning procedure.

## H  ALGORITHM

We describe the final WBU learning procedure in Algorithm 1. Note that they keyword **Update** means that we compute the gradients of the input loss, and update the parameters of the neural networks of the pointed function/model accordingly.

**Normalizing term.** Given the set of parameters $\iota$ of $\varphi$, we minimize the KL divergence $D_{\mathrm{KL}}$ by gradient descent on the Monte-Carlo estimate of the divergence:

$$\nabla_\iota D_{\mathrm{KL}}\big(\varphi\big(\bar{b}_t, a_t, o_{t+1}; \iota\big) \,\|\, \bar{\tau}\big(\bar{b}_t, a_t, o_{t+1}\big)\big) =$$

$$\nabla_\iota \mathop{\mathbb{E}}_{\bar{s}_{t+1}\sim\varphi\big(\bar{b}_t, a_t, o_{t+1}; \iota\big)} \left[\log\varphi\big(\bar{s}_{t+1} \mid \bar{b}_t, a_t, o_{t+1}; \iota\big) - \log \mathop{\mathbb{E}}_{\bar{s}\sim\bar{b}_t}\overline{\mathbf{P}}(\bar{s}_{t+1} \mid \bar{s}, a_t) - \log\overline{\mathcal{O}}(o_{t+1} \mid \bar{s}_{t+1})\right]$$

Notice that the first term of the divergence (the belief normalization term) of Eq. 8 does not depend on $\varphi$ and thus yields zero gradient. Nevertheless, we observed during our experiments that adding the normalizing term allows to stabilize and reduce the variance of the belief loss.

**Optimizing Wasserstein.** To optimize the Wasserstein term of the belief losses, we follow the same learning procedure than (Delgrange et al., 2023, Appenix A.5): we introduce neural networks $\mathcal{F}_\spadesuit$ (for $\spadesuit \in \{\overline{\mathbf{P}}, \Omega\}$) that are trained to attain the supremum of the dual formulation of the Wasserstein distance. To do so, we need enforce the Lipschitzness of $\mathcal{F}_\spadesuit$ and, as in WAE-MDPs (Delgrange et al., 2023), we do so via the gradient penalty approach of Gulrajani et al. (2017), leveraging that any differentiable function is 1-Lipschitz iff it has gradients with norm at most 1 everywhere. Finally, notice that we do not directly optimize the total variation distance of $L_\mathcal{O}$, but rather the Wasserstein; we take the usual Euclidean distance as metric over $\Omega$ which is proven to be Lipschitz equivalent to a distance converging to the discrete metric as the temperature of the WAE-MDP goes to zero (Delgrange et al., 2023, Appendix A.6) to finally recover $d_{TV}$.

## I  HYPERPARAMETERS

Table 1 provides the range of hyperparameters used in the search, along with the selected values for each environment. The hyperparameter search was performed using OPTUNA (Akiba et al., 2019). Pre-training of the WAE-MDP involved collecting 10240 transitions with a random policy and performing 200 training steps. These pre-training transitions are taken into account in the reported results.

Additionally, Table 2 (for R-A2C) and Table 3 (for DVRL) present the specific hyperparameters used for each algorithm. A grid search was conducted over all possible combinations for both baselines. The hidden size of all neural networks was set to 128 neurons and two hidden layers (except for the sub-belief encoder, and the policy which uses one) without further tuning. The experiments were carried out using 16 parallel environments. The original implementation of DVRL and their version of R-A2C were used in this study.

We ran the experiments on a cluster composed of Intel Xeon Gold 6148 CPU.

In the following tables the enviromnents are represented as follows: STATELESSCARTPOLE $\heartsuit$, NOISYSTATELESSCARTPOLE $\clubsuit$, REPEATPREVIOUS $\diamondsuit$, SPACEINVADERS $\spadesuit$, NOISYSPACEINVADERS $\star$.

The t-SNE (van der Maaten & Hinton, 2008) presented in the main text, in Fig. 4, was obtained using cuML (Raschka et al., 2020). Table 4 reports the hyperparameters that were used. We used 50,000 sub-belief-value pairs, sampled from a dataset of 1 million timesteps collected at the end of the training.

---

**Algorithm 1:** WASSERSTEIN BELIEF UPDATER

---

**Input:** Batch sizes $B_{\text{WBU}}$, $B_{\text{WAE}}$, $B_{\text{NEXT}}$; global learning steps $N$; no. of model updates per
   iteration $N_{\text{MODEL}}$; your favorite collect strategy $\pi_{\text{init}}$; replay buffer (RB) $\mathcal{D}$; Lipschitz
   networks $\mathcal{F}_{\overline{\mathbf{P}}} \colon \overline{\mathcal{S}} \to \mathbb{R}$, $\mathcal{F}_{\Omega} \colon \Omega \to \mathbb{R}$; observation variance network $\overline{\mathcal{O}}_{\sigma}$; and loss
   weights $w_{\overline{\mathcal{R}}}$, $w_{\overline{\mathbf{P}}}$

**collect** $\theta = \{\, s_i, o_i, a_i, r_i, s_i', o_i' \,\}_{i=1}^{N_{\text{init}}}$ by executing $\pi_{\text{init}}$ for $N_{\text{init}}$ steps; **store** $\theta$ in $\mathcal{D}$
         $\triangleright$ *Use the exploration policy $\pi_{\text{init}}$ to collect transitions and initialize the RB*

**repeat** $N$ **times**
 **repeat** $N_{\text{MODEL}}$ **times**
  $\triangleright$ *Update the WAE-MDP model for $N_{\text{MODEL}}$ consecutive training steps*
  **for** $i \leftarrow 1$ *to* $B_{\text{WAE}}$ **do**
   $\langle s_i, o_i, a_i, r_i, s_i', o_i' \rangle \sim \mathcal{D}$       $\triangleright$ *Sample a transition from the RB*
   $\bar{s}' \leftarrow \phi(s_i', o_i')$         $\triangleright$ *Embed $\langle s_i', o_i' \rangle$ to the latent space*
   $\tilde{o}_i' \sim \overline{\mathcal{O}}(\cdot \mid \bar{s}')$        $\triangleright$ *Observe the resulting latent state via $\overline{\mathcal{O}}$*

  $\mathcal{L}_{\text{WAE}} \leftarrow$ **compute the WAE-MDP loss** on transition batch $\{\, s_i, o_i, a_i, r_i, s_i', o_i' \,\}_{i=1}^{B_{\text{WAE}}}$
  **Update** the WAE-MDP components (*in particular, those of Eq. 2*) by minimizing $\mathcal{L}_{\text{WAE}}$
  $\mathcal{L}_{\mathcal{O}} \leftarrow {}^{1}\!/_{B_{\text{WAE}}} \cdot \sum_{i=1}^{B_{\text{WAE}}} \left[ \mathcal{F}_{\Omega}(o_i') - \mathcal{F}_{\Omega}(\tilde{o}_i') \right]$   $\triangleright$ *Observation loss*
  **Update** $\mathcal{F}_{\Omega}$ by maximizing $\mathcal{L}_{\mathcal{O}}$ and enforcing its 1-Lipschizness w.r.t. metric $d_{\Omega}$
  **Update** $\overline{\mathcal{O}}_{\sigma}$ by minimizing $\mathcal{L}_{\mathcal{O}}$

 **for** $i \leftarrow 1$ *to* $B_{\text{WBU}}$ **do**
  $s_0 \leftarrow s_I$; $\bar{s}_0 \leftarrow \bar{s}_I$; $\bar{b}_0 \leftarrow \delta_{\bar{s}_0}$; $\beta_0 \leftarrow \beta_I$    $\triangleright$ *$\beta_I$ is arbitrary, e.g., zeroes*
  **for** $t \leftarrow 0$ *to* $T$ **do**
   $a_t \sim \bar{\pi}(\cdot \mid \beta_t)$    $\triangleright$ *Produce the action $a_t$ according to the sub-belief $\beta_t$*
   **execute** $a$ in the environment, **receive reward** $r_t$, and **perceive** the next
    state-observation $\langle s_{t+1}, o_{t+1} \rangle$
   **store** the transition $\langle s_t, o_t, a_t, r_t, s_{t+1}, o_{t+1} \rangle$ into $\mathcal{D}$
   $\beta_{t+1} \leftarrow \varphi^{sub}(\text{sg}(\beta_t), a_t, o_{t+1})$   $\triangleright$ *Update the sub-belief; sg is stop gradients*
   $\bar{b}_{t+1} \leftarrow \mathbb{M}(\beta_{t+1})$    $\triangleright$ *Retrieve the belief distribution $b_{t+1}$ via the MAF $\mathbb{M}$*
   $\bar{s}_{t+1} \sim \bar{b}_{t+1}$        $\triangleright$ **Believe** *the next latent state*
   $\triangleright$ *Marginalize the next latent state distribution w.r.t. the current belief*
   **for** $j \leftarrow 1$ *to* $B_{\text{NEXT}}$ **do**
    $\bar{s} \sim \bar{b}_t$; $\mathcal{L}_{\log \overline{\mathbf{P}}}^{j} \leftarrow \left[ \log \overline{\mathbf{P}}(\bar{s}_{t+1} \mid \bar{s}, a_t) - \log B_{\text{NEXT}} \right]$
   $\mathcal{L}_{\text{KL}}^{i,t} \leftarrow \log \bar{b}_{t+1}(\bar{s}_{t+1}) - \text{LSE}(\{\, \mathcal{L}_{\log \overline{\mathbf{P}}}^{j} \,\}_{j=1}^{B_{\text{NEXT}}}) - \log \overline{\mathcal{O}}(o_{t+1} \mid \bar{s}_{t+1})$
      $\triangleright$ *Pointwise decomposition of Eq. 8: divergence with the belief update rule*
   $\mathcal{L}_{\overline{\mathcal{R}}}^{i,t} \leftarrow \left| \overline{\mathcal{R}}(\phi(s_t, o_t), a_t) - \overline{\mathcal{R}}(\bar{s}_t, a_t) \right|$    $\triangleright$ *Latent reward regularizer*
   $\bar{s}' \sim \overline{\mathbf{P}}(\cdot \mid \bar{s}_t, a_t)$
     $\triangleright$ *Transition to the next latent state from the current believed latent state*
   $\mathcal{L}_{\overline{\mathbf{P}}}^{i,t} \leftarrow \left[ \mathcal{F}_{\overline{\mathbf{P}}}(\phi(s_{t+1}, o_{t+1})) - \mathcal{F}_{\overline{\mathbf{P}}}(\bar{s}') \right]$    $\triangleright$ *Latent transition regularizer*

 **Update** $\mathcal{F}_{\overline{\mathbf{P}}}$ by maximizing $\sum_{i=1}^{B_{\text{WBU}}} \sum_{t=0}^{T-1} \mathcal{L}_{\overline{\mathbf{P}}}^{i,t}$ and enforcing its 1-Lipschitzness w.r.t.
  latent metric $\bar{d}$
 **Update** $\varphi^{sub}$ and $\mathbb{M}$ by minimizing $1/(B_{\text{WBU}}+T) \sum_{i=1}^{B_{\text{WBU}}} \sum_{t=0}^{T-1} (\mathcal{L}_{\text{KL}}^{i,t} + w_{\overline{\mathcal{R}}} \cdot \mathcal{L}_{\overline{\mathcal{R}}}^{i,t} + w_{\overline{\mathbf{P}}} \cdot \mathcal{L}_{\overline{\mathbf{P}}}^{i,t})$
 **Update** $\bar{\pi}$ by minimizing the A2C loss on the batch $\{\, \beta_{0:T}^{i}, a_{0:T-1}^{i}, r_{0:T-1}^{i} \,\}_{i=1}^{B_{\text{WBU}}}$

**function** $\overline{\mathcal{O}}(\cdot \mid \bar{s})$            $\triangleright$ *(smooth) Observation Filter*
 $\mu \leftarrow \overline{\mathcal{O}}_{\mu}(\bar{s})$; $\sigma \leftarrow \overline{\mathcal{O}}_{\sigma}(\bar{s})$   $\triangleright$ *Decode $\bar{s}$; get the standard deviation of the reconstruction*
 **return** $\mathcal{N}(\mu, \sigma^2)$

---

Table 1: Range of hyperparameter search and selection per environment.

| | Range | ♡ | ♣ | ◇ | ♠ | ★ |
|---|---|---|---|---|---|---|
| WAE Updates per Belief Update | 1-2 | 1 | 1 | 2 | 2 | 2 |
| Activation function | leaky relu, elu | leaky relu | leaky relu | elu | elu | relu |
| Activation function lipshitz | leaky relu, tanh | tanh | tanh | tanh | tanh | leaky relu |
| Activation CNN | leaky relu, elu | | | | elu | elu |
| **Policy config** | | | | | | |
| Learning rate | 1.e-4, 3.e-4, 5.e-4 | 3.e-4 | 1.e-4 | 1.e-4 | 5.e-4 | 5.e-4 |
| Clip norm | 1, 10 | 10 | 10 | 10 | 10 | 10 |
| **Belief config** | | | | | | |
| Same opt as policy | Yes, No | Yes | Yes | Yes | No | No |
| Learning rate | 1.e-4, 3.e-4, 5.e-4 | | | | 1.e-4 | 3.e-4 |
| Loss factor | 1.e-1, 1.e-2, 1.e-3, 1.e-4, 1.e-5 | 1.e-1 | 1.e-4 | 1.e-5 | | |
| Filter variance min | 1.e-2, 1.e-3 | 1.e-2 | 1.e-2 | 1.e-3 | 1.e-2 | 1.e-2 |
| Normalize log obs filter | True, False | False | False | True | True | True |
| Sub belief prior temperature | .5, 0.66, .75, .9, .99 | 0.66 | 0.9 | 0.66 | 0.99 | 0.99 |
| Reward loss scale factor | 0.1, 1., 10., 20., 50., 100. | 0.1 | 20 | 100 | 100 | 100 |
| Transition loss scale factor | 0.1, 1., 10., 20., 50., 100. | 50 | 100 | 50 | 50 | 100 |
| Buffer size | 4096, 8192, 16384 | 4096 | 4096 | 16384 | 4096 | 4096 |
| N Critic Update | 5, 10 | 10 | 10 | 5 | 5 | 5 |
| N State samples | 16, 32, 64 | 32 | 64 | 32 | 32 | 32 |
| N next state samples | 16, 32, 64 | 32 | 64 | 64 | 16 | 32 |
| **WAE config** | | | | | | |
| Latent state size | ♡,♣ : 5 → 10, Others: 18 → 25 | 5 | 6 | 20 | 18 | 19 |
| Minimizer learning rate | 1.e-3, 3.e-4,1.e-4, 5.e-5, 1.e-5 | 5.e-5 | 5.e-5 | 3.e-4 | 1.e-5 | 1.e-5 |
| Maximizer learning rate | 1.e-3, 3.e-4,1.e-4, 5.e-5, 1.e-5 | 5.e-5 | 5.e-5 | 1.e-3 | 3.e-4 | 1.e-4 |
| State encoder temperature | 0.33, .5, 0.66, .75, .9, .99 | 0.33 | 0.75 | 0.66 | 0.5 | 0.5 |
| State prior temperature | 0.33, .5, 0.66, .75, .9, .99 | 0.75 | 0.75 | 0.5 | 0.75 | 0.75 |
| Local transition loss scaling | 10., 25., 50., 75., 80. | 50 | 80 | 80 | 10 | 10 |
| Steady state scaling | 10., 25., 50., 75., 80. | 80 | 80 | 100 | 25 | 25 |
| N critic update | 5, 10 | 10 | 10 | 5 | 5 | 10 |
| Batch size | 128, 256 | 128 | 128 | 256 | 128 | 128 |
| Clip grad | 1, 10. 100. | 100 | 100 | 100 | 10 | 10 |
| State vs Obs reconstruction weight | 1, 2 | 2 | 2 | 2 | 1 | 1 |
| Obs reg. min learning rate | 1.e-3, 3.e-4,1.e-4, 5.e-5, 1.e-5 | 5.e-5 | 1.e-4 | 1.e-3 | 5.e-5 | 5.e-5 |
| Obs reg. max learning rate | 1.e-3, 3.e-4,1.e-4, 5.e-5, 1.e-5 | 5.e-5 | 3.e-4 | 5.e-5 | 1.e-5 | 3.e-4 |
| Obs reg. gradient penalty | 50, 100, 500, 1000 | 50 | 50 | 1000 | 100 | 1000 |

Table 2: R-A2C hyperparameters

| | Range | ♡ | ♣ | ◇ | ♠ | ★ |
|---|---|---|---|---|---|---|
| Opitmizer | Adam, RMSProp | RMSProp | RMSProp | Adam | RMSProp | RMSProp |
| Clip grad norm | 0.5, 1., 10. | 1.0 | 10 | 10. | 10 | 10 |
| Learning rate | 3.e-5, 1.e-4, 3.e-4, 5.e-4, 1.e-3 | 5.e-4 | 1.e-4 | 5.e-4 | 1.e-3 | 1.e-3 |

Table 3: DVRL hyperparameters

| | Range | ♡ | ♣ | ◇ | ♠ | ★ |
|---|---|---|---|---|---|---|
| Optimizer | Adam, RMSProp | Adam | Adam | RMSProp | RMSProp | RMSProp |
| Clip grad norm | 0.5, 1., 10. | 10.0 | 1.0 | 1.0 | 10 | 0.5 |
| Learning rate | 3.e-5, 1.e-4, 3.e-4, 5.e-4, 1.e-3 | 5.e-4 | 1.e-3 | 3.e-4 | 1.e-3 | 1.e-3 |
| | | | | | | |
| Encoding loss factor | 1,.1,.5,.05 | 0.5 | 1.0 | 1.0 | 1.0 | 0.05 |
| Number of particles | 5,10,15 | 10 | 3 | 10 | 5 | 5 |

Table 4: t-SNE hyperparameters

| | |
|---|---|
| Method | FFT |
| Perplexity | 10 |
| N Neighbors | 50 |
| Metric | Manhattan |
| Learning rate | 2400 |
| Early exaggeration | 50 |
| Late exaggeration | 3 |
| Exaggeration iter | 300 |

## J  INDEX OF NOTATIONS

**Divergences**

$D_{\mathrm{KL}}$    KL divergence

$d_{TV}$    Total variation distance

$\mathcal{W}_d$    Wasserstein distance

**Extended POMDP**

$\mathcal{O}^\uparrow \colon \mathcal{S}_\Omega \to \Omega$  Extended observation deterministic function (projection on the observation space)

$\mathcal{P}^\uparrow$    Extended POMDP

$\mathcal{M}_\Omega$    Extended MDP

$\mathbf{P}_\Omega \colon \mathcal{S}_\Omega \times \mathcal{A} \to \Delta(\mathcal{S}_\Omega)$  Extended probability transition

$\mathcal{R}_\Omega \colon \mathcal{S}_\Omega \times \mathcal{A} \to \mathbb{R}$  Extended reward function

$\mathcal{S}_\Omega = \mathcal{S} \times \Omega$  Extended state space

**Losses**

$L_{\bar{\tau}}$    Belief loss

$L_{\mathcal{R}}$    Local reward loss

$L_{\mathbf{P}}$    Local transition loss

$\mathcal{L}$    Combined local loss: $\mathcal{L} = L_{\mathcal{R}} + L_{\overline{\mathcal{R}}}^\varphi + \overline{\mathcal{R}}^\star L_{\bar{\tau}} + \gamma K_{\overline{V}} \cdot \left( L_{\mathbf{P}} + L_{\overline{\mathbf{P}}}^\varphi + L_{\bar{\tau}} + L_{\mathcal{O}} \right)$

$L_{\mathcal{O}}$    Observation loss

$L_{\overline{\mathcal{R}}}^\varphi$    On-policy reward loss

$L_{\overline{\mathbf{P}}}^\varphi$    On-policy transition loss

**MDP**

$a, \mathcal{A}$    Action, action space

$\gamma \in [0, 1)$   Discount factor

$\mathcal{M}$    MDP

$\mathbf{P} \colon \mathcal{S} \times \mathcal{A} \to \Delta(\mathcal{S})$   Transition function

$r, \mathcal{R}$    Reward, reward function

$s, \mathcal{S}$    State, state space

**POMDP**

$b, \mathcal{B}$    Belief, belief space

$\tau \colon \mathcal{B} \times \mathcal{A} \times \Omega \to \mathcal{B}$   Belief update function

$\tau^* \colon (\mathcal{A} \cdot \Omega)^* \to \mathcal{B}$   Recursive application of the belief update rule along histories

$h, (\mathcal{A} \cdot \Omega)^*$   History, history space

$\mathcal{H}_\pi$    Stationary distribution over histories based on a policy $\pi$

$o, \Omega$    Observation, observation space

$\mathcal{O} \colon \mathcal{S} \times \mathcal{A} \to \Delta(\Omega)$   Observation stochastic function

$\pi \colon \mathcal{B} \to \Delta(\mathcal{A})$   POMDP policy conditioned on beliefs

$\mathcal{P}$    POMDP

$V_\pi$    Value function for the policy $\pi$

**Latent (PO)MDP**

$\varphi \colon \overline{\mathcal{B}} \times \mathcal{A} \times \Omega \to \overline{\mathcal{B}}$  Belief encoder

$\varphi^* \colon (\mathcal{A} \cdot \Omega)^* \to \overline{\mathcal{B}}$  Recursive application of the belief encoder along histories

$\varphi^{\text{sub}}$      Sub-belief encoder, so that $\mathbb{M} \circ \varphi^{\text{sub}}(\beta, a, o) = \varphi(\mathbb{M}(\beta), a, o)$

$\beta$      sub-belief, so that $\mathbb{M}(\beta) = \bar{b} \in \bar{\mathcal{B}}$

$K_{\overline{V}}$      $\frac{\overline{\mathcal{R}}^\star}{1-\gamma}$

$\bar{b}, \bar{\mathcal{B}}$      Latent belief, latent belief space

$\bar{\tau} \colon \bar{\mathcal{B}} \times \mathcal{A} \times \Omega \to \bar{\mathcal{B}}$   Latent belief update

$\bar{\tau}^* \colon (\mathcal{A} \cdot \Omega)^* \to \bar{\mathcal{B}}$   Recursive application of the latent belief update rule along histories

$\overline{\mathcal{M}}$      Latent MDP

$\overline{\mathcal{O}} \colon \bar{\mathcal{S}} \to \Delta(\Omega)$   Latent observation function

$\bar{\pi} \colon \bar{\mathcal{B}} \to \Delta(\mathcal{A})$   POMDP latent policy conditioned on latent beliefs

$\overline{\mathcal{P}}$      Latent POMDP

$\overline{\mathbf{P}} \colon \bar{\mathcal{S}} \times \mathcal{A} \to \Delta(\bar{\mathcal{S}})$   Latent probability transition

$\bar{s}, \bar{\mathcal{S}}$      Latent state, latent state space

$\overline{V}_{\bar{\pi}}$      Latent value function for the policy $\bar{\pi}$

$\mathbb{M}$      MAF; allows to pass from sub-beliefs to latent beliefs $\bar{b}$ via $\mathbb{M}(\beta) = \bar{b} \in \bar{\mathcal{B}}$

$\bar{r}, \overline{\mathcal{R}}$      Latent reward, latent reward function

$\overline{\mathcal{R}}^\star$      $\left\| \overline{\mathcal{R}} \right\|_\infty = \sup_{\bar{s} \in \bar{\mathcal{S}}, a \in \mathcal{A}} \left| \overline{\mathcal{R}}(\bar{s}, a) \right|$

**WAE-MDP Components**

$\psi \colon \bar{\mathcal{S}} \to \mathcal{S} \times \Omega$   Extended state decoder, defined as $\psi(\bar{s}) = \langle \psi^{\mathcal{S}}(\bar{s}), \overline{\mathcal{O}}_\mu(\bar{s}) \rangle$

$\psi^{\mathcal{S}} \colon \bar{\mathcal{S}} \to \mathcal{S}$   State decoder

$\phi \colon \mathcal{S}_\Omega \to \bar{\mathcal{S}}$   State embedding function

$\overline{\mathcal{O}}_\mu \colon \bar{\mathcal{S}} \to \Omega$   Deterministic observation decoder

## ADDITIONAL REFERENCES

Takuya Akiba, Shotaro Sano, Toshihiko Yanase, Takeru Ohta, and Masanori Koyama. Optuna: A next-generation hyperparameter optimization framework. *CoRR*, abs/1907.10902, 2019. URL `http://arxiv.org/abs/1907.10902`.

K. B. Athreya and P. Ney. A new approach to the limit theory of recurrent markov chains. *Transactions of the American Mathematical Society*, 245:493–501, 1978. ISSN 00029947. URL `http://www.jstor.org/stable/1998882`.

Christel Baier and Joost-Pieter Katoen. *Principles of model checking*. MIT Press, 2008. ISBN 978-0-262-02649-9.

Yoshua Bengio, Nicholas Léonard, and Aaron C. Courville. Estimating or propagating gradients through stochastic neurons for conditional computation. *CoRR*, abs/1308.3432, 2013. URL `http://arxiv.org/abs/1308.3432`.

Luca de Alfaro, Thomas A. Henzinger, and Rupak Majumdar. Discounting the future in systems theory. In Jos C. M. Baeten, Jan Karel Lenstra, Joachim Parrow, and Gerhard J. Woeginger (eds.), *Automata, Languages and Programming, 30th International Colloquium, ICALP 2003, Eindhoven, The Netherlands, June 30 - July 4, 2003. Proceedings*, volume 2719 of *Lecture Notes in Computer Science*, pp. 1022–1037. Springer, 2003. doi: 10.1007/3-540-45061-0\_79. URL `https://doi.org/10.1007/3-540-45061-0_79`.

Josée Desharnais, Vineet Gupta, Radha Jagadeesan, and Prakash Panangaden. Metrics for labelled markov processes. *Theor. Comput. Sci.*, 318(3):323–354, 2004. doi: 10.1016/j.tcs.2003.09.013. URL `https://doi.org/10.1016/j.tcs.2003.09.013`.

Rick Durrett. *Probability: Theory and Examples, 4th Edition*. Cambridge University Press, 2010. ISBN 9780511779398. doi: 10.1017/CBO9780511779398. URL `https://doi.org/10.1017/CBO9780511779398`.

Jiri Fajtl, Vasileios Argyriou, Dorothy Monekosso, and Paolo Remagnino. Latent bernoulli autoencoder. In *Proceedings of the 37th International Conference on Machine Learning, ICML 2020, 13-18 July 2020, Virtual Event*, volume 119 of *Proceedings of Machine Learning Research*, pp. 2964–2974. PMLR, 2020. URL `http://proceedings.mlr.press/v119/fajtl20a.html`.

Robert Givan, Thomas L. Dean, and Matthew Greig. Equivalence notions and model minimization in markov decision processes. *Artif. Intell.*, 147(1-2):163–223, 2003. doi: 10.1016/S0004-3702(02) 00376-4. URL `https://doi.org/10.1016/S0004-3702(02)00376-4`.

Ishaan Gulrajani, Faruk Ahmed, Martín Arjovsky, Vincent Dumoulin, and Aaron C. Courville. Improved training of wasserstein gans. In Isabelle Guyon, Ulrike von Luxburg, Samy Bengio, Hanna M. Wallach, Rob Fergus, S. V. N. Vishwanathan, and Roman Garnett (eds.), *Advances in Neural Information Processing Systems 30: Annual Conference on Neural Information Processing Systems 2017, December 4-9, 2017, Long Beach, CA, USA*, pp. 5767–5777, 2017. URL `https://proceedings.neurips.cc/paper/2017/hash/892c3b1c6dccd52936e27cbd0ff683d6-Abstract.html`.

Ruiquan Huang, Yingbin Liang, and Jing Yang. Provably efficient UCB-type algorithms for learning predictive state representations. In *The Twelfth International Conference on Learning Representations*, 2024. URL `https://openreview.net/forum?id=jId5PXbBbX`.

Eric Jang, Shixiang Gu, and Ben Poole. Categorical reparameterization with gumbel-softmax. In *5th International Conference on Learning Representations, ICLR 2017 - Conference Track Proceedings*, 11 2017. URL `https://arxiv.org/abs/1611.01144`.

Kim Guldstrand Larsen and Arne Skou. Bisimulation through probabilistic testing. *Inf. Comput.*, 94 (1):1–28, 1991. doi: 10.1016/0890-5401(91)90030-6. URL `https://doi.org/10.1016/0890-5401(91)90030-6`.

Qinghua Liu, Alan Chung, Csaba Szepesvári, and Chi Jin. When is partially observable reinforcement learning not scary? In Po-Ling Loh and Maxim Raginsky (eds.), *Conference on Learning Theory, 2-5 July 2022, London, UK*, volume 178 of *Proceedings of Machine Learning Research*, pp. 5175–5220. PMLR, 2022. URL `https://proceedings.mlr.press/v178/liu22f.html`.

Chris J. Maddison, Andriy Mnih, and Yee Whye Teh. The concrete distribution: A continuous relaxation of discrete random variables. In *5th International Conference on Learning Representations, ICLR 2017, Toulon, France, April 24-26, 2017, Conference Track Proceedings*. OpenReview.net, 2017. URL `https://openreview.net/forum?id=S1jE5L5gl`.

Alfred Müller. Integral probability metrics and their generating classes of functions. *Advances in Applied Probability*, 29(2):429–443, 1997. ISSN 00018678. URL `http://www.jstor.org/stable/1428011`.

D. Revuz. *Markov Chains*. North-Holland mathematical library. Elsevier Science Publishers B.V., second (revised) edition, 1984. ISBN 9780444864000. URL `https://books.google.be/books?id=zIDWlAEACAAJ`.

Erfan SeyedSalehi, Nima Akbarzadeh, Amit Sinha, and Aditya Mahajan. Approximate information state based convergence analysis of recurrent q-learning. In *Sixteenth European Workshop on Reinforcement Learning*, 2023. URL `https://openreview.net/forum?id=co8W7pvSrS`.

Bharath K Sriperumbudur, Kenji Fukumizu, Arthur Gretton, Bernhard Schölkopf, and Gert RG Lanckriet. On integral probability metrics,\phi-divergences and binary classification. *arXiv preprint arXiv:0901.2698*, 2009.

Aäron van den Oord, Oriol Vinyals, and Koray Kavukcuoglu. Neural discrete representation learning. In Isabelle Guyon, Ulrike von Luxburg, Samy Bengio, Hanna M. Wallach, Rob Fergus, S. V. N. Vishwanathan, and Roman Garnett (eds.), *Advances in Neural Information Processing Systems 30: Annual Conference on Neural Information Processing Systems 2017, December 4-9, 2017, Long Beach, CA, USA*, pp. 6306–6315, 2017. URL https://proceedings.neurips.cc/paper/2017/hash/7a98af17e63a0ac09ce2e96d03992fbc-Abstract.html.

Robert J. Vanderbei. Uniform continuity is almost lipschitz continuity. Technical Report SOR-91–11, Statistics and Operations Research Series, Princeton University, 1991.

