# OpenReview forum: "The Wasserstein Believer: Learning Belief Updates for Partially Observable Environments through Reliable Latent Space Models"
_ICLR.cc/2024/Conference — ICLR 2024 poster_

### Official Review · Reviewer_Zb4B · 2023-10-27

**Soundness:** 2 fair
**Presentation:** 3 good
**Contribution:** 3 good
**Rating:** 6
**Confidence:** 3

**Summary:**

The paper proposes Wasserstein Belief Updater which is based on the Wasserstein Auto-encoded MDP (WAE-MDP) framework to learn a latent model to update beliefs for partially observable problems. Due to partial observability, in addition to WAE-MDP components, the proposed method learns a latent observation function and a latent belief encoder which recursively updates the beliefs. Experiments show that the RL agent using the updated belief outperforms previous RNN based algorithms.

**Strengths:**

- Every step of the proposed approach is explained well, and their connections to prior work are clearly discussed.

- The WBU approach nicely extends WAE-MDP to the partially observable settings, and the paper also provide theoretical guarantees that extends bisimulation bounds in WAE-MDP to POMDPs.

- The architecture of the latent belief encoder seems to be a novel by having the Masked Auto-Regressive Flows as the output layer on top of a sub-belief encoder. The training of the sub-belief is then done without back-propagation through time.

- Experimental results suggest that the beliefs updated by the latent belief encoder are useful information for decision-making with partial observations. The WBU agent achieves strong performance in several POMDP environments, and it especially outperforms existing algorithms by a significant margin in the environment requiring long-term memorization.

**Weaknesses:**

- Even though the method is called Wasserstein Belief Updater and the theoretical analysis is for the Wasserstein belief loss, the actual algorithm uses a KL loss for the belief updater part. Hence, the guarantees do not necessarily hold for the actual algorithm.

- The algorithm doesn't have any step for updating the MAF. Is there something missing or is the MAF pre-trained by some other way?

- In Section 4, the authors claim that the training of the sub-belief encoder does not require the challenging BPTT as in RNN. But given the recursive structure of the sub-belief encoder, it is not clear why the claim is true. Even when the sub-belief encoder is updated only by the belief loss, the gradient flow might still need to propagate back if the previous sub-belief is also generated by the same sub-belief encoder. Back-propagation might stop if the previous sub-belief is actually generated by other networks, like using (3), but this part is a bit unclear.

**Questions:**

- Can you provide more details on MAF training?

- Can you provide more details on how the sub-belief encoder is trained and why BPTT is not needed with some gradient analysis?

- Some possible typos and missing definitions:
  - Is $\varphi^*$ defined somewhere?
  - In equation (5), should the Wassertein distance be given by $D$ or?

---

> ### Author Response · Authors · 2023-11-16
>
> We thank the reviewer for their insightful comments, and answer their questions in the following.
>
> **Masked autoregressive flow.** The MAF is trained with the sub-belief encoder to minimize Equation 8. This is mentioned in the training paragraph of Section 4 and in the algorithm pseudo code in Appendix F. Details concerning the sampling and learning procedure of the particular MAF we use are given in Delgrange et. al, 2023, Section 3.2 (last paragraph).
>
> **Backpropagation through time (BPTT)** is a powerful tool for learning value functions due to their backward recurrence nature; an update in the value of a history $h_t$ ​ under policy $\pi$ can propagate changes back to previous histories. This approach is efficient as it reduces the number of necessary updates by allowing changes to flow through past timesteps. However, in the context of belief learning, the situation is different. Beliefs progress through forward recurrence, implying that a modification in belief $b_t$ affects subsequent beliefs but not preceding ones. Consequently, BPTT does not offer the same utility for belief learning as it does for value functions. To align with this forward recurrence nature and avoid the unnecessary complications of BPTT, we apply a stop gradient in our model, effectively preventing the backward flow of updates. This approach is illustrated in Figure 2 and detailed in the algorithm presented in Appendix F, where the use of the stop gradient __sg__ is explicitly implemented and explained.
>
> **Typos and missing definitions.** We did not explicitly define $\varphi^*$ which corresponds to the recursive application of $\varphi$ ( analogously to $\tau^*$). We modified the paper to make it explicit. The distance $D$, defined in the box page 5 as the distance between the real belief updater and our approximation, is set to the Wasserstein distance in the paragraph Belief Losses above Equation 5.
>
> **Theoretical Justification for KL as a Proxy.** The KL divergence, while more tractable to optimize, also offers theoretical backing. As per the Pinsker's inequality, in the zero-temperature limit, the KL divergence bounds the Wasserstein distance. This relationship allows us to leverage the easier optimization landscape of the KL divergence while still maintaining a connection to the Wasserstein framework. Essentially, optimizing the KL divergence serves as a practical approach to approximate the Wasserstein distance, especially given the challenges associated with the latter in our specific learning context. We refer to the answer to Reviewer Gzjw (paragraph *Wasserstein proxy*) for the practical challenges of Wasserstein optimization.
>
> References:
> - Florent Delgrange, Ann Nowe, and Guillermo Perez. Wasserstein auto-encoded MDPs: Formal verification of efficiently distilled RL policies with many-sided guarantees. (ICLR 2023)

---

> > ### Comment · Reviewer_Zb4B · 2023-11-21
> >
> > Thank you for the reply. I still don't agree with the argument for the unnecessity of BPTT. Beliefs do progress forward, but the same forward recurrence argument would also apply for the latent state process which was used in the previous RNN approach. One can always stop gradient in the implementation which results in truncated BPTT, but truncated BPTT only provides approximate gradients in general. Since gradient stop is used in training the model, does it mean that the gradients computed by the proposed method are only approximations? If that's the case, what is the justification of using approximate gradients and why would they be better than the actual gradients (computed by BPTT without gradient stop) in the belief case?

---

> ### Author Response · Authors · 2023-11-22
>
> We thank the reviewer for their reactivity and allowing us to expand on why BPTT is not needed in the context of Belief Learning.
>
> The reviewer noted that in the RNN-A2C the hidden state $z_t$ is also computed recursively, which is in the same spirit as how the belief $b_t$ is computed. The main difference between beliefs and hidden states lies in what they represent.
>
> On the one hand, beliefs are state distributions which can be inferred from the previous belief and be themselves seen as Markovian states (cf. the belief MDP construction). Therefore, one can see the belief update function as the transition function of an MDP where an observation is further provided. Disabling BPTT is therefore consistent with the literature of model based RL for learning Markovian transition functions (e.g., Gelada et al. 2019, Vincent François Lavet et al. 2019, Delgrange et al. 2022). Notably, our goal is not targeted towards learning directly a history representation (from sequences) *but rather on learning to reproduce the belief update rule*, based on the latent stochastic transition function $\bar{P}$, and observation function $\bar{O}$ of the latent model, which are learned separately and considered fixed in this phase. Therefore, stopping the gradients does not produce approximate gradients, as the error linked to this recursive update can be fully determined through the error of our latent model (Theorem 3.2, Equation 6), and using BPTT would increase the risk of accumulating errors across time.
>
> On the other hand, hidden states are latent representations of the history and it is up to the RL algorithm to learn a useful representation of the history for the control task. This is in stark contrast to our approach: we separately learn a representation which is guaranteed to be well suited for the RL agent, and we use it as input of the policy. In RNN-A2C the main objective is to find a policy that optimizes the return conditioned on the full history, making it sound to do BPTT in that case. In our case the RL agent uses the beliefs as a given state-signal to condition the value function and the policy, and therefore the RL objective is not backpropagated through the belief encoder.
>
> Finally, it is worth mentioning that as beliefs are computed with a forward recurrence, errors may accumulate over time. This means that if the belief $b_{t+k}$ diverges significantly from its correct value, then $b_{t+k+1}$ will diverge even more. Then, allowing gradients from the learning of $b_{t+k+1}$ to be computed based on an incorrect $b_{t+k}$ and further flow to $b_t$ may hinders the learning procedure.
>
> **References**
> - C. Gelada et al. "DeepMDP: Learning Continuous Latent Space Models for Representation Learning". ICML 2019.
> - V. François-Lavet et al. "Combined reinforcement learning via abstract representations". AAAI 2019.
> - F. Delgrange et al. "Distillation of RL policies with formal guarantees via variational abstraction of Markov decision processes". AAAI 2022.

---

### Official Review · Reviewer_Gzjw · 2023-10-30

**Soundness:** 3 good
**Presentation:** 3 good
**Contribution:** 3 good
**Rating:** 6
**Confidence:** 4

**Summary:**

This paper proposes an algorithm that learns the belief in POMDP.

## Learning the Environment
The algorithm first learns an estimation of the true MDP via the WAE-MDP method, under assumption that the underlying model in POMDP is accessible in training (*Assumption 1*). The WAE-MDP learns a latent MDP model which encodes not only the true state transition of the POMDP but also the observation mechanism, thus it induces all information of the original POMDP, if it is exactly learned (but WAE-MDP learns an estimation instead).

## Theoretical results
Under the episodic assumption of the POMDP (*Assumption 2*),
the estimation of the difference between values from the underlying (true) MDP and from the latent MDP is upper bounded theoretically by certain linear combination of local losses and Belief losses, which are expected Wasserstein distances or reward differences between the underlying (true) MDP / belief system and latent MDP / belief system. And in the variance zero limit, choosing a different learning history would result in an upper-bounded difference in optimal Value functions.

## Learning the Belief
Once the POMDP is known via the latent MDP, one can use the first-order belief evolution formula derived from Bayesian method to calculate the belief from timestep 0 to any timestep. But Bayesian method requires intractable integration over current belief, so the final method is to train another model to learn the belief.

Training belief follows online optimization of the objective function which is the expected Wasserstein distance between $\overline{\tau}$ and $\phi$. And due to computational difficulty, expected KL divergence is used as objective instead.

## Experiments
The paper tested the algorithm in several environments such as Space Invaders, Stateless Cartpole, etc., compare with R-A2C and DVRL.

**Strengths:**

- The paper provides a solid math ground (theoretical analysis of the POMDP dynamics) by
    - proving the existence of a distribution on training history set.
    - uses the distribution above to estimate certain bounds of learning effectiveness (Thm 3.2) and learning variance (Thm 3.3).
The results guarantees the learning algorithm would converge to a good approximation.

- Proposed learning methods (objectives) are from theoretical results.

- Outperforms R-A2C and DVRL in certain POMDP environments.

**Weaknesses:**

- Wasserstein Belief Updater finally chooses the objective function to be KL divergence as a Wasserstein proxy. I agree with the difficulty of optimizing Wasserstein (which is quite singular). But probably the authors can thing about the entropy-regularized version of Wasserstein distance? Hope the work after the following paper could help.
[ Marco Cuturi, Sinkhorn Distances: Lightspeed Computation of Optimal Transportation Distances ]
[ Marco Cuturi, et. al., Fast Computation of Wasserstein Barycenters ]

- I did not see what are the requirements of the sets we discuss? I guess they are at least discrete to make certain Markov chain theoretical results hold. But as the paper talked about Cartpole, I guess the theory would meant to be possibly applied to continuous state spaces (real manifold at least).

**Questions:**

Questions:
- In the theoretical part, the existence of $\mathcal{H}_{\overline{\pi}}$ is relying on the ergodicity of an episodic MDP (Huang Bojun, 2020), where the state space $\mathcal{S}$ is assumed to be discrete (finite or countably infinite). I want to know the topology (and geometry) of the history space.

- Is it the collection of all $(a,o)$ sequences such that $(\mathcal{A}\cdot \Omega)^*=Map(\mathbb{N}, \mathcal{A}\times \Omega)$, or it is the collection of all finite-length tuples that $(\mathcal{A}\cdot \Omega)^*=\bigsqcup_{n\in\mathbb{N}}(\mathcal{A}\times \Omega)^n$? Categorically speaking, is it a direct product (product) or direct sum (coproduct)? See https://en.wikipedia.org/wiki/Product_(category_theory).

- Is there any theoretical result on the continuous spaces such as CartPole? The definition of ergodicity is slightly different there.


Typos
- I could hardly analyse the second sentence of the abstract, .
- Page 4, first line is hard to read. (But the meaning is clear)

---

> ### Author Response · Authors · 2023-11-16
>
> **General state space POMDPs.** We thank the reviewer for pointing out that in general state spaces, ergodicity notions are different. We are aware of the issue and, while there is a precedent of allowing this small disconnection between theory and practice, we have (for completeness) confirmed the guarantees do hold in general state spaces under reasonable assumptions. Below we elaborate on these two points.
>
> As the reviewer mentioned, the theory and results developed by Huang Bojun (2020) hold for finite or countable infinite state spaces. Nevertheless, for their experiments, they evaluated their approach on uncountable infinite state space MDPs, on RobotSchoolBullet environments. Arguably, their evaluation agrees with the theory developed since everything is discrete in practice: the computer simulates the environment whose episodes are derived from a finite branching of histories unrolled through a machine-induced discretization of the observations.
>
> Assuming that uncountable infinite state spaces are endowed with a usual Borel sigma-algebra, we can still show that all our results hold in any episodic RL setting. We have updated the Appendix accordingly. Intuitively, the policies considered in our work all induce a *recurrent Harris chain* (Definition C.6 in Appendix), where the interaction is guaranteed to eventually result in a stationary distribution. Therefore, the approach of Huang 2020 (which facilitates rapid convergence to the stationary distribution) is still valid in our setting. We added a complete, formal discussion about this setting in Appendix C.3 (part (b) of the proof).
>
> **History space.** We admit that we are not experts in category theory, but relying on the sigma-algebra defined by Revuz (1984) and Putterman (1994), the history space is $(\mathcal{A}\cdot \Omega)^*=\bigsqcup_{n\in\mathbb{N}}(\mathcal{A}\times \Omega)^n$. This should clarify what topology we are working in.
>
> **Wasserstein proxy.** We thank the reviewer for the pointers to entropy-regularized versions of the Wasserstein distance. As pinpointed in the main text, Wasserstein optimization is known to be challenging, and one can indeed avoid the optimal coupling (primal) or Lipschitz (dual) constraints by considering entropy-regularized or relaxed distances. Still, the major challenge in learning to minimize a Wasserstein-like discrepancies between our latent belief encoder and the true latent belief update function is that *one need be able to sample from both distributions*, even via an entropy-regularized version of the distance. As mentioned in the main text, sampling from our belief encoder $\varphi$ is done via a masked autoregressive flow, but it is non-trivial to sample batches of states from the true latent belief yielded by $\overline{\tau}$: indeed, in that case, we need to infer the distribution from the closed form of Equation 3. To that aim, one could consider Monte Carlo Markov Chain sampling techniques such as Metropolis-Hastings, but this complicates the learning process as such procedures are computationally expensive, and further brings technical challenges. In contrast, Kullback Leilbler (KL) is easier to optimize since we only need to sample from our belief encoder to minimize the divergence (see Equation 8), while still offering guarantees (KL bounds Wasserstein by the Pinsker’s inequality in the zero-temperature limit, as mentioned in the main text). We defer as future work the study of alternative Wasserstein-like discrepancy to our KL loss.
>
> *References* :
> - Bojun Huang. Steady state analysis of episodic reinforcement learning. (NeurIPS 2020)
> - D. Revuz. Markov Chains. North-Holland mathematical library. Elsevier Science Publishers B.V.,
> second (revised) edition, 1984.
> - Martin L. Puterman. Markov Decision Processes: Discrete Stochastic Dynamic Programming

---

### Official Review · Reviewer_Rz6s · 2023-10-31

**Soundness:** 2 fair
**Presentation:** 2 fair
**Contribution:** 2 fair
**Rating:** 5
**Confidence:** 3

**Summary:**

The authors introduce a model-based procedure, the Wasserstein Belief Updater (WBU), for learning/optimizing POMDPs. The components are a latent model learner, a belief update learner, and a policy optimizer. The authors make an assumption that the POMDP latent states can be accessed during training time. They provide some theoretical approximation error bounds which bound the model quality and the representation quality resulting from their losses. The authors provide details on how to practically implement their proposal (Section 4), and provide experimental evaluations for their algorithm (Section 5).

**Strengths:**

- The paper presents a few interesting ideas for learning bisimulation-type metrics for POMDPs, which is a reasonable extension from the MDP case due to the additional structure of needing to approximate the belief states. Some of the ideas are certainly interesting and potentially worth further development.
- The problem of representation learning in POMDPs is difficult and interesting.
- The theoretical results do provide some amount of theoretical backing, and at a glance the proofs for the approximation error bounds seem non-trivial due to the partially observable problem setting. I have not checked the proofs very carefully.
- The ideas and theory are supported by experiments which show some improvement over baselines.

**Weaknesses:**

- The biggest drawback to this work is the latent observability condition. The methodology being presented is fairly depend on this condition, since it attempts to learn the latent transition model, observation model, and belief states. This simplifies the problem setting quite a bit by making the POMDPs more or less a complicated involved MDP. It is not clear how much applicability this method has beyond the settings where this assumption holds.
- In terms of related work, there is a fair amount of literature on provable guarantees for exploration in POMDPs under latent observability (e.g. Lee et al. ICML 2023 and related/followup work) or under structural conditions for POMDPs (e.g. Liu et al. 2022 and related/followup work) which is not mentioned here.
- The current theoretical guarantees are similar to the other papers on learning with bisimulation-like metrics, in that the extent of the theoretical results is a bound on the approximation error incurred via the population loss functions. I would argue that this is not an "end-to-end" theoretical guarantee that is useful, since there are many questions about how one can minimize such a loss when only given access to samples from the distribution (i.e. how close are the empirical losses and the population losses, are there any double-sampling issues, is one sampling from distributions one does not have access to, etc.) and if we can minimize it on policies that actually matter for downstream performance rather than only on the on-policy distributions (currently it is simply an "on-policy" bound which holds only for the distribution under which the data was collected). As an example of the weakness of such a bound, the usual bisimulation metric in tabular MDPs satisfies a similar bound (Theorem 5.1, Ferns et al. 2004), but there is still an exponential lower bound for learning any bisimulation-like abstraction (Appendix B of Modi et al., 2020). Thus, the argument that this paper presents theory that can be used to give guarantees is either incomplete or inaccurate, depending on one's perspective. The authors could perhaps support their theory by experimentally verifying that the (population) losses are indeed minimized and verifying that the quantities on the LHS of their bounds are controlled by their procedure.
- In terms of writing and presentation, I feel that more effort should be put into streamlining the notations and explanations. For example, even after several re-reads, it is still difficult to keep track of the notation for the different models MDPs/POMDPs being passed between, to type-check things (e.g. the state embedding functions depend on different things at different points), and to grok which ideas are essential/novel and which are simply details.
- Similarly, there is a fair amount of statements which are "fake rigorous" i.e. do not check out formally. Here is a list:
 1. "Unlike in MDPs, stationary policies that are based solely on the current observation of P do not induce any probability space on trajectories of M". This is probably not what the authors meant to say, since any policy (stationary or not) induces a distribution over trajectories.
 2. "This procedure guarantees \bar{M} to be probably approximately bisimlarly close to M as \lambda \rightarrow 0". Is this a formal claim? There is no PAC bound given here nor in the cited prior work on WAE-MDPs.
 3. Lemma 3.1: "There is a well defined probability distribution [...] over histories likely to be perceived at the limit by the agent when it executes \bar{\pi} in P". What does this mean? I can try to guess what this mean, but as a formal statement this does not make sense.
 4. Theorem 3.3: there is a "for any \epsilon > 0" in the statement and in the bound but no other quantities in the bound depend on $\varepsilon$
- The experimental results mild improvement over the baselines in some environments, and no improvement over the baselines in other environments. What is the source of this varying degree of improvement? Can more experiments be performed to understand when this specific approach yields benefits?

References:
Ferns, N., Panangaden, P., & Precup, D. (2004). Metrics for finite Markov decision processes. UAI 2004.
Modi A, Jiang N, Tewari A, Singh S. Sample complexity of reinforcement learning using linearly combined model ensembles. AISTATS 2020.
Lee, J., Agarwal, A., Dann, C. and Zhang, T., 2023, July. Learning in pomdps is sample-efficient with hindsight observability. ICML 2023.
Liu, Q., Chung, A., Szepesvári, C., & Jin, C. (2022, June). When is partially observable reinforcement learning not scary?. In Conference on Learning Theory (pp. 5175-5220). PMLR.

**Questions:**

There are several technical questions which I have:
1. What does "we assume that we have access to the latent model learned by the WAE-MDP" mean?
2. What does Lemma 3.1 mean?
3. In paragraph "Local losses", how is one expected to sample from $\tau^\star(h)$, the true belief function for history $h$.
4. What does $\varepsilon$ represent in Theorem 3.3?
5. It is interesting that Theorem 3.3. only holds on the optimal policy $\bar{\pi}^\star$. What is the source of this, and can this not be extended to work for other policies of interest?
6. Figure 3: Belief loss. How can the loss be negative?

---

> ### Author Response · Authors · 2023-11-16
>
> We appreciate the reviewer's insightful comments, and in the subsequent discussion, we aim to answer their engaging technical questions and address their concerns.
>
> **Responses to specific questions:**
>
> **Access to the latent model vs Access to states during training.**
> 1. **Access to the latent space model:** In Section 3, we present how to learn the latent space model, which is then used later on, in Section 4. Specifically, this includes using the learned latent transition and latent observation functions to train our belief encoder to replicate latent belief update function. *In practice, the latent model and belief encoder are learned in a round robin fashion* (see Figure 1).
> 2. **Access to the states during training:** For a detailed discussion on the applicability of Assumption 1, we kindly refer the reviewer to our Introduction and Remark 1, in the main text. Notice that despite the observability of the state during learning, the problem remains formalized as a POMDP, since the resulting policy is defined over past observations and actions, and not on the MDP states. We note that in the field of planning in POMDPs, the access to the model of the environment is typically assumed, which is a much stronger assumption, while keeping a POMDP formulation. Therefore, we believe our approach is not a simplification of the problem and does not reduce the problem into a complicated involved MDP.
>
> **Distribution over histories.** Lemma 3.1 ensures the existence of a (stationary) distribution over histories occurring under the latent policy, when the latter is executed in the original POMDP. We have modified the main text of the paper to improve the readability of the Lemma. Please, further notice that Appendix C is dedicated to the proof of the statement, where an extended and detailed version of this Lemma is formalized (Lemma C.1).
>
> **Sampling states from the true belief.** In the section related to local losses, sampling a state from the true belief function w.r.t. a history is achieved in practice via a replay buffer, where states perceived during the interaction are stored. Sampling a state from the replay buffer is therefore equivalent to sampling (i) a history, and then (ii) a state from the resulting belief computed through the true belief update function $\tau^*$.
>
> **Almost Lipschitzness.** In Theorem 3.3, the $\epsilon$ represents an error factor. Intuitively, when the losses go to zero, given $\epsilon > 0$, there always exists a constant $K < \infty$ so that the value function is almost surely $K$-Lipschitz w.r.t. the latent belief space, up to the error factor $\epsilon$.  The error $\epsilon$ is tied to $K$ as the proof relies on the _almost-Lipschitzness_ of the value function (see Definition E.8 and Lemma E.9 in Appendix). The epsilon-K notation is standard when referring to almost Lipschitzness (cf . Vanderbei 1991).
>
> **Representation quality relies on optimality.** The reviewer rightly noticed that the applicability of the Theorem is limited to optimal policies. At the moment, we conjecture that the Theorem may not apply to arbitrary policies. Technically, this is due to the fact that we rely on almost Lipschitzness of  the optimal value function of the POMDP (see Lemma E.9 in Appendix for additional details), which does not necessarily hold in general, for arbitrary policies. Nevertheless, in the context of RL we are particularly interested in optimal policies, so it is natural to focus on them.
>
> **Negative belief loss.** The negative values in Figure 3 result from the Monte Carlo approximation of the belief loss. This actually may occur when minimizing KL for two distributions $P, Q$  with densities $p, q$ via Monte Carlo sampling, since the log-ratio $\log \frac{p(x)}{q(x)}$ of the two distributions is averaged from samples produced by $x \sim P$ and not integrated over the whole space. We made clearer in the caption of Figure 3 that the belief loss is estimated.

---

> > ### Author Response · Authors · 2023-11-16
> > **On the weaknesses pointed out by the reviewer**
> >
> > **Related work.** We thank the reviewer for the pointers to relevant works. We have included references to Lee et al. in our revised related  work section. We would like to highlight the fact that *Lee et al.'s assumption of hindsight observability aligns closely with our Assumption 1.*
> >
> > **Theoretical guarantees.** Our focus is on developing deep RL theories for POMDPs, which is not necessarily oriented towards sample complexity results. We indeed intentionally targeted on-policy metrics, which explains the expectation operation over histories perceived during the interaction in our bounds. While WAE-MDPs are linked to bisimulation (pseudo-)metrics with guarantees, our belief encoder learning does not directly relate to bisimulation metrics (and definitely not to strong bisimulation concepts). However, we acknowledge the potential to strengthen our guarantees by incorporating bisimulation metrics in future work, as mentioned in our Conclusion, and we agree that this represents a considerable challenge.
> >
> > **Notations and formalities.** To assist the reader with the notations, we added an Index of notations in Appendix. Note that for the sake of presentation, we needed sometimes to find the right balance between formalities and intuitions in the paper. However, all the statements and rigorous proofs are clearly detailed and formalized in the Appendix. Furthermore, we also kept into consideration the remarks of the other reviewers to improve on this balance and the overall presentation of the results.
> >
> > Concerning those formalities,  we revised the confusing statements mentioned by the reviewer. Additionally, we would like to kindly point out to the reviewer that the statement made in Section 2.3 regarding the PAC bisimulation bounds does not reference to WAE-MDPs (Delgrange et al 2023), but rather the work of Delgrange et al, 2022, where PAC bounds are indeed formalized (and further used as the foundations of WAE-MDPs).
> >
> > **Experimental performance.** The source of the varying degree of improvement compared to the baseline is detailed in the experimental section. In short, whenever only short memorization is required, RNNs tend to perform well and learn fast, although they do so without any guarantees. Their learning capability and stability decrease as the size of required memory increases or when observations are noisy.
> >
> > *References*:
> > - Florent Delgrange, Ann Nowe, and Guillermo A. Perez. Distillation of rl policies with formal guarantees via variational abstraction of markov decision processes. (AAAI 2022)
> > - Florent Delgrange, Ann Nowe, and Guillermo Perez. Wasserstein auto-encoded MDPs: Formal verification of efficiently distilled RL policies with many-sided guarantees. (ICLR 2023)
> > - Robert J. Vanderbei. Uniform continuity is almost lipschitz continuity. Technical Report SOR-91–11, Statistics and Operations Research Series, Princeton University, 1991.

---

> > > ### Author Response · Authors · 2023-11-20
> > >
> > > As per request from Reviewer yC5B we expanded on our answer to several of your points (refinement of the related work, details about bisimulation metrics, and the nature of our guarantees) in a general answer.

---

### Official Review · Reviewer_yC5B · 2023-11-01

**Soundness:** 4 excellent
**Presentation:** 4 excellent
**Contribution:** 4 excellent
**Rating:** 10
**Confidence:** 5

**Summary:**

The paper studies latent space models for POMDPs, where the authors present theoretical guarantees on the quality of the learned state representation, i.e., capturing the dynamics of the environment, and its suitability for policy optimization. A key ingredient of the proposed formulation is the assumed access to the true state of the environment during the training phase, e.g., available through simulators or additional high fidelity sensors. This assumption enables a latent space model consisting of an augmented MDP with an associated POMDP amenable for learning, where the original POMDP is recovered as a variance parameter defining the latent POMDP observations goes to zero. The authors proceed to establish bounds on the difference in value functions upon conditioning the policy on the latent representation, i.e., the value function of the latent space model vs the true value function, in addition to a Lipschitz-like bound on the value difference in terms of the divergence between the embedded belief states. To evaluate the proposed method, the authors consider three types of POMDPs requiring long-term memory, inferring hidden states, and robustness to noise, where the proposed belief updater shows significant gains.

**Strengths:**

- Proposes the utilization of extra information available during training to guide the learning.
- Makes an important contribution to learning of POMDPs, grounded in a clear connection to an underlying MDP, with an effective learning strategy that comes with theoretical guarantees.
- Excellent exposition with detailed derivations and illustrations.

**Weaknesses:**

Some assumptions need to be highlighted early on with a clear account of where they may break, or how to workaround them in practice; see below.

**Questions:**

**Missing technical discussion:**
- Assumption(2) requiring an episodic RL process appears (out of nowhere) in S3.2.
  - It would help to anticipate the need for this assumption earlier in the introduction
  - It would also help to discuss how restrictive this assumption is, as was done for Assumption(1), and whether it is mainly a technicality for some of the theoretical derivations.
  - Perhaps related to this additional assumption, the future work section mentioned adaptations under relaxed assumptions, with a citation to (Lambrechts et al. 2022). It would help to elaborate on some of those potential relaxations.

**Presentation:**
- Abstract:
  - Assumptions regarding access to the true state during training should be mentioned in the abstract.
- Section 3:
  - It would help to introduce a simple diagram showing the relationship between the different (PO)MDPs, perhaps including the symbols for key new constructs. (I see the authors already have illustrations under Fig.6 on P.20. It appears the authors preferred to include Fig.1, so perhaps readers like me could benefit from at least a pointer to Fig.6 somewhere on P.5.)
  - (Follow up): I see now I missed how "bar" symbols refer to the latent MDP. The definition of $\bar{V}_{\bar{\pi}}$ appears right before Thm.3.2. It would help to establish this notation early in S3.1.
  - Let me also suggest to break up the paragraph before Eq.3.
  - Value difference bounds:
      - The big fraction on Eq.6 reappears on that of Eq.7, with only one extra term in the numerator. Since Thm.3.2 applies to the optimal policy, perhaps it would be possible to make the connection more explicit. (Indeed, this seems to appear halfway through the proof on P.29.)
      - The variance assumption in Thm.3.3 appears to be a technicality, which I worry it hinders the actual message. Looking at the proof, I see it's only needed for the value function to be almost Lipschitz. Following the pointer to Remark-2 (which only appears *after* Thm.3.3, I see that smoothing the observation in the first place was only introduced to enable training (also alluded to earlier in S3.1). I suggest to amend the statement of Thm.3.3 to clarify how the variance will be made to vanish by design, e.g., *as learning converges to the deterministic mapping ensured by ...*. (I see Fig.4 is obatiend *at the late stage of training*).
      - (Follow up) Based on the above, I don't see why Remark-2 appears where it is right now.

**Nitpicking:**
- Abstract: is can be used

---

> ### Author Response · Authors · 2023-11-16
>
> We would like to thank the reviewer for their comments, having attentively checked the Appendix, and their very positive appreciation of our work.
>
> **Episodic assumption.** Assumption 2 holds in a wide range of RL scenarios: intuitively this means that the agent is trained in an episodic manner, i.e., the environment is reset when the agent succeeds, fails, or after a finite number of time steps.This actually holds in the majority of RL environments relying on a simulator (see, e.g., Brockman et al., 2016; Pardo et al., 2018). Continual RL is one of the few settings where this Assumption does not hold. However, in our setting, the episodic assumption is used to prove the existence of a stationary distribution over the histories perceived under the learned policy (Lemma 3.1). We note that we can relax the episodic assumption if we can assume the existence of a stationary distribution, which is common in continual RL (see, e.g., Huang 2020 for a discussion). We added a remark (Remark 2) in the paper related to this episodic assumption, aiming to explore the constraints of this assumption and identify scenarios in which it can be relaxed.
>
> **Comments on the presentation.** We have updated the paper to implement the presentation suggestions. As the reviewer rightfully noticed, the discussion about the observation variance is technical and only required to learn the belief updater, in Section 4. Therefore, we removed this discussion from Section 3, we modified Theorem 3.3, and refined Remark 3 in Section 4. Furthermore, to help the reader to keep track of the notations, we added an index of notations in Appendix.
>
> *References*:
> - Bojun Huang. Steady state analysis of episodic reinforcement learning. (NeurIPS 2020)
> - Greg Brockman, Vicki Cheung, Ludwig Pettersson, Jonas Schneider, John Schulman, Jie Tang, and
> Wojciech Zaremba. Openai gym (2016)
> - Fabio Pardo, Arash Tavakoli, Vitaly Levdik, and Petar Kormushev. Time limits in reinforcement
> learning. (ICML 2018)

---

> > ### Comment · Reviewer_yC5B · 2023-11-17
> > **Thanks for the updates + follow up**
> >
> > I appreciate the author's response to my own comments.
> >
> > However, I'd like to see a more compelling response addressing reviewer's Rz6s comments regarding:
> > - Related works with related formulations/results.
> > Quoting reviewer Rz6s:
> > > [..] there is a fair amount of literature on provable guarantees for exploration in POMDPs under latent observability (e.g. Lee et al. ICML 2023 and related/followup work) or under structural conditions for POMDPs (e.g. Liu et al. 2022 and related/followup work) which is not mentioned here
> > - The nature of the theoretical guarantees:
> >   - How it may or may not serve as a useful end-to-end guarantee.
> >   - A more detailed comparison to known results using bisumulation metrics, with a more through review of this literature clarifying how the present contribution fits there.

---

> > > ### Author Response · Authors · 2023-11-20
> > >
> > > We thank the reviewer for their additional feedback. We addressed the different points in the general answer.

---

### Author Response · Authors · 2023-11-16
**General answer**

We express our gratitude to all the reviewers for their valuable feedback. We especially appreciate the attention given to the details presented in the main text, but also in the Appendix. Our paper, along with the Appendix, has been updated based on the comments provided by the reviewers.

---

### Author Response · Authors · 2023-11-20
**Related Work, Bisimulation and Nature of our guarantees**

We are grateful for all the useful recommendations on how to improve our related work Section. We will expand the current one for the final version of the paper. In the following, we elaborate on the concerns of Reviewer Rz6s, as requested by Reviewer yC5B17, as we believe our response may be relevant to other reviewers as well.

## Related Work
In the same spirit as in our work, the observability assumption of Lee et al. 2023 aligns perfectly with ours, but is leveraged in a different way, to provide results oriented towards sample complexity in the tabular case, as well as regret bounds when particular types of function approximators are used. Several complexity results and regrets bounds were developed for sub-classes of POMDPS in finite horizon, such as when the observability is well-behaved (weakly revealed POMDPs, Liu et al. 2022), or when the sequential decision making problem admits low rank structure (Liu et al. 2022, Huang et al. 2023). On the contrary, our results hold for general POMDPs; precisely, our constraints rather rely on the training procedure and not on the nature of the POMDP. Finally, we note that RQL-AIS (Seyedsalehi et al. 2023) presents an analysis of recurrent Q-learning for the tabular setting which also includes value difference bounds where their latent state is the (discrete) hidden state of an RNN. While theoretically interesting, their bounds are generally intractable as they rely on global metrics requiring them to be minimized over the whole history and action space. This limitation is made clear in their algorithm as they use the local version of those metrics as regularization losses. In our case, because we focus on on-policy metrics (local losses) our theoretical guarantees hold in practice.

## Bisimulation
We stress some important differences between bisimulation relations and bisimulation (pseudo-)metrics. Bisimulation is a behavioral equivalence relation between states and models whose properties include trajectory and value equivalence (Larsen & Skou, 1989; Givan et al., 2003). However, due to their all or nothing nature, bisimulation relations are often too restrictive; intuitively, states that share close rewards and dynamics but exhibit slight numerical differences are treated as being completely different. Bisimulation pseudometrics (Alfaro et al., 2003, Desharnais et al., 2004) generalize bisimulation by assigning a distance between states and quantifying their bisimilarity instead of yielding binary signals indicating whether they are bisimilar or not. Specifically, real-valued quantitative properties (Alfaro et al., 2003, Chatterjee et al., 2010) as well as expected returns (Ferns et al., 2012) are guaranteed to be Lipschitz continuous w.r.t. the related bisimulation distance.

Works on representation learning for RL (e.g., Gelada et al., 2019; van der Pol et al., 2020, Castro et al., 2021; Zhang et al., 2021; Zang et al., 2022, Delgrange et al., 2023) use bisimulation metrics to optimize the representation quality offered by a latent space, and aim at learning representations which capture the bisimulation distance, so that two states close in the representation provide close and relevant information for the policy. Precisely, such works seek the Lipschitz continuity of the optimal value function with regards to the distance of pairs of points in the latent space.

We stress that while our work builds upon the WAE-MDP framework (Delgrange et al., 2023), which learns a latent model based on bisimulation metrics, **our work does not learn any bisimulation-like metric in the underlying latent space of the POMDP.** Instead, we prioritize what truly matters in the context of representation for RL: the Lipschitz-continuity of the optimal value function. In particular, the attentive reader may notice that, in contrast to usual representation bounds in MDPs (e.g., Gelada et al. 2019, Delgrange et al. 2022), our bound involves the *almost* Lipschitzness the value function, as well as various additional losses linked to the belief update and observation functions, which underlines the challenge of providing such guarantees. Indeed, bisimulation in the context of POMDPs represents a major challenge (see e.g., Castro et al., 2009 for a discussion): intuitively, in this context, the bisimulation equivalence is not defined on the state space but rather on the belief space induced by the different components of the POMDP. Notably, even if the POMDP is finite, i.e., state, action, and observation spaces are finite, the belief space can be represented as the usual (uncountable space) simplex, which prevents the application of bisimulation algorithms developed in the context of countable MDPs. We are not aware of works concerning bisimulation pseudometrics in the context of POMDPs.

---

> ### Author Response · Authors · 2023-11-20
>
> ## Nature of our guarantees
>
> *We argue that our algorithm does enable an “end-to-end” learning in the sense that, at any time of the optimization process, the current measures of the losses of our algorithm serve as indicators of the proximity between the latent and original models*. In essence, this is exactly what we present in our main text; and in addition to end-to-end learning it allows a posteriori model checking, if desired. Our work is not oriented towards sample complexity or regrets, but rather on the representation guarantees that are induced by our algorithm based on the optimization of *tractable* losses. This tractability is related to their measurability, and the feasibility of their optimization and estimation at any time of the interaction (a.k.a. “on-policy” learning).
>
> Precisely, we focus on representation quality guarantees when function approximators are used to generate a suitable representation of the space over which decisions are based. To do so, we provide tractable losses, whose minimizations are compliant with stochastic gradient descent algorithms used by neural networks. These losses are guaranteed to lead to (i) a latent space model where the agent behaves closely to the way it behaves in the original environment, and (ii) a latent representation of the history space (over which decisions are based) that is guaranteed to be almost-Lipschitz continuous in the value function of the optimal policy. While (i) allows the agent to maintain a belief on the latent state space and take its decisions accordingly, (ii) guarantees that the representation induced by our approach is suitable to optimize the expected return, while ensuring that collapsing issues does not occur under optimality (any histories having close representation never lead to divergent returns).
>
> We stress that we designed the losses to be locally tractable, i.e., measurable and estimable under latent policies, which is in stark contrast to losses whose optimality relies on the whole state, action, and observation spaces, which is infeasible to optimize in practice. This means that at any time of the training process, the current measures of the losses are informative on the closeness of the latent and original models (Theorem 3.2).
>
> We agree with Reviewer Rz6s that the representation quality (Theorem 3.3) holding specifically for the optimal latent policy can be seen as a limitation since it could be interesting to assess the representation quality offered by our representation for any policy, but (i) we specifically focus on RL, where we seek for optimality, and (ii) we currently conjecture that such a bound does not hold for arbitrary policies, under the current assumptions. We defer as future work the refinement of the quality of the representation under relaxed assumptions.
>
> **References**
> - Lee, J., Agarwal, A., Dann, C. and Zhang, T., 2023, July. Learning in pomdps is sample-efficient with hindsight observability. ICML 2023
> - Liu, Q. et al. (2022). When is partially observable reinforcement learning not scary?. In Conference on Learning Theory. PMLR.
> - Huang, R. et al. Provably efficient ucb-type algorithms for learning predictive state representations. (2023).
> - SeyedSalehi E. el al., Approximate information state based convergence analysis of recurrent Q-learning, EWRL 2023
> - Larsen and Skou, Bisimulation through probabilistic testing. Proceedings of the 16th ACM SIGPLAN-SIGACT symposium on Principles of programming languages. 1989.
> - Givan, R. et al. Equivalence notions and model minimization in Markov decision processes. Artificial Intelligence (2003)
> - De Alfaro, L., et al.. Discounting the future in systems theory. International Colloquium on Automata, Languages, and Programming. 2003.
> - Desharnais, J., et al. Metrics for labelled Markov processes. Theoretical computer science (2004)
> - Chatterjee, et al. Algorithms for game metrics. Logical Methods in Computer Science  (2010).
> - Ferns, N.,et al.. Metrics for finite Markov decision processes. (2012).
> - Gelada, et al. Deepmdp: Learning continuous latent space models for representation learning. ICML. PMLR, 2019.
> - Van der Pol E, et al.. Plannable Approximations to MDP Homomorphisms: Equivariance under Actions, AAMAS 2020.
> - Castro, PS, et al. MICo: Improved representations via sampling-based state similarity for Markov decision processes. Advances in Neural Information Processing Systems 34 (2021):
> - Zhang, A., et al.(2021). Learning invariant representations for reinforcement learning without reconstruction.
> - Zang, H., et al.. SimSR: Simple distance-based state representations for deep reinforcement learning. AAAI 2022.
> - F. Delgrange, et al.. Wasserstein auto-encoded MDPs: Formal verification of efficiently distilled RL policies with many-sided guarantees. (ICLR 2023)
> - Castro, PS, et al. Equivalence Relations in Fully and Partially Observable Markov Decision Processes. IJCAI. 2009.

---

### Author Response · Authors · 2023-11-23

We would like to thank the reviewers for their constructive comments, which helped to improve the quality of our paper. We hope that the discussion phase successfully addressed all the concerns raised.

---

### Meta-Review · Area_Chair_TJuL · 2023-12-08

**Metareview:**

The authors propose a method, called Wasserstein Belief Updater (WBU), for solving POMDPs. The method is model-based and consists of 1) learning a latent model, 2) learning a belief update, and 3) optimizing policy. The method is based on an assumption that the latent states of the POMDP can be accessed during training. Given this assumption, the authors prove error bounds for model and representation quality. Finally, they show how their method can be implemented in practice and empirically evaluate it.

Representation learning in POMDPs and solving these models are important and challenging problems. The authors present interesting ideas for learning bisimulation metrics in POMDPs (extension from MDPs is non-trivial due to the need for approximating the belief state). The paper has a good balance of theoretical analysis, algorithmic implementation, and empirical evaluation.

The main limitation is the assumption that the state of the environment is observable during training, which plays an important role in the proposed method. This assumption simplifies the POMDP problem (getting closer to the MDP setting) and restricts the applicability of the method. There are also room for improvement in terms of 1) literature study and referring to related results, 2) writing and presentation, 3) claims against BPTT, and 4) better explaining the limitations of the theoretical results, which have been pointed out by the reviewers. It would be great if the authors take these points into account in preparing the next revision of their work.

**Justification For Why Not Higher Score:**

The main limitation is the assumption that the state of the environment is observable during training, which plays an important role in the proposed method. This assumption simplifies the POMDP problem (getting closer to the MDP setting) and restricts the applicability of the method. There are also room for improvement in terms of 1) literature study and referring to related results, 2) writing and presentation, 3) claims against BPTT, and 4) better explaining the limitations of the theoretical results, which have been pointed out by the reviewers.

**Justification For Why Not Lower Score:**

Representation learning in POMDPs and solving these models are important and challenging problems. The authors present interesting ideas for learning bisimulation metrics in POMDPs (extension from MDPs is non-trivial due to the need for approximating the belief state). The paper has a good balance of theoretical analysis, algorithmic implementation, and empirical evaluation.

---

### Decision · Program_Chairs · 2024-01-16

Accept (poster)